# Blue-shift photoconversion of near-infrared fluorescent proteins for labeling and tracking in living cells and organisms

Francesca Pennacchietti [1] ✉, Jonatan Alvelid [1], Rodrigo A. Morales [2,3], Martina Damenti [1], Dirk Ollech [1], Olena S. Oliinyk[4], Daria M. Shcherbakova[5], Eduardo J. Villablanca [2,3], Vladislav V. Verkhusha[4,5] & Ilaria Testa [1] ✉

Photolabeling of intracellular molecules is an invaluable approach to studying various dynamic processes in living cells with high spatiotemporal precision. Among fluorescent proteins, photoconvertible mechanisms and their products are in the visible spectrum (400–650 nm), limiting their in vivo and multiplexed applications. Here we report the phenomenon of near-infrared to far-red photoconversion in the miRFP family of near infrared fluorescent proteins engineered from bacterial phytochromes. This photoconversion is induced by near-infrared light through a non-linear process, further allowing optical sectioning. Photoconverted miRFP species emit fluorescence at 650 nm enabling photolabeling entirely performed in the near-infrared range. We use miRFPs as photoconvertible fluorescent probes to track organelles in live cells and in vivo, both with conventional and super-resolution microscopy. The spectral properties of miRFPs complement those of GFP-like photoconvertible proteins, allowing strategies for photoconversion and spectral multiplexed applications.

Fluorescent proteins (FPs) have been engineered to reversibly and irreversibly shift their spectral properties upon illumination at specific wavelength and energy[1,2]. Some FPs exhibit transitions from a dark to a bright state (photoactivation, PA) while others change color (photoconversion, PC)[3]. Irreversible spectral shift, also called photolabeling, enables a diverse type of imaging experiments where a subset of molecules or cells can be highlighted and monitored over time using conventional or super-resolution fluorescence microscopy[4–7]. Such a tool provides unique spatial and temporal information compared to conventional time-lapse imaging where the entire labeled population is monitored at the same time i.e., the dynamics of single cells or a subset of a protein species located in a specific compartment can be discerned from the ensemble and averaged behavior down to μs–ms temporal resolution[8].

Photoconvertible fluorescent proteins (PCFPs) can fluoresce in both states, which adds the option to monitor both photo-converted and unconverted population simultaneously. PCFPs have been engineered from many GFP-like proteins with a green-to-red spectral shift (like Dendra[9], mEos[10], mKikGR[11]) and further screened for brightness, minimal cross-talk and photostability in both states[12]. FPs capable of a cyan-to-green (PSCFP2[13]) and an orange-to-far-red (PSmOrange[14] and PSmOrange2[15]) spectral shift are also available. A common feature of these PCFPs is the need for ultraviolet (UV) or blue-green illumination to drive photoconversion. To bypass the use of highly energetic UV light, two-photon photoconversion[16,17] or primed conversion[18], i.e., the combined illumination of blue and red/near-infrared (NIR) light, have been used. However, the former approach suffers from low yield whereas the latter still requires

[1]Department of Applied Physics and SciLifeLab, KTH Royal Institute of Technology, Stockholm 17165, Sweden. [2]Division of Immunology and Allergy, Department of Medicine Solna, Karolinska Institutet and University Hospital, Stockholm 17176, Sweden. [3]Center for Molecular Medicine (CMM), Stockholm 17176, Sweden. [4]Medicum, University of Helsinki, Helsinki 00290, Finland. [5]Department of Genetics, and Gruss-Lipper Biophotonics Center, Albert Einstein College of Medicine, Bronx, NY 10461, USA. ✉e-mail: frapen@kth.se; testa@kth.se

visible light illumination and relatively high red-light intensities due to the photoconversion efficiency[18].

PCFPs with spectra and photoconversion light at longer wavelengths such as the NIR region are still missing despite being desirable for bioimaging applications, as they would reduce autofluorescence, light scattering, and phototoxicity[19,20]. Bacterial phytochrome photoreceptors (BPhPs) carry the most NIR-shifted known natural chromophore, a linear tetrapyrrole biliverdin IXα (BV). As an enzymatic product of heme catabolism, BV has the advantage of being naturally produced in mammalian cells[21]. BPhPs consist of at least four domains, including PAS (Per-ARNT-Sim), GAF (cGMP phosphodiesterase/adenylate cyclase/FhlA), PHY (phytochrome-specific), and various effector domains. The BV chromophore is located in the GAF domain and has a covalent thioether bond between the $C3^2$ atom of ring A and a Cys at

the N-terminal part of the PAS domain[22]. Photoswitching via light-induced rotation of ring D is an inherent feature of BPhPs whose primary function in host bacteria is light-driven signal transduction[23]. BPhPs reversibly switch between a red (Pr, $\lambda_{abs}$ ~700 nm) and a far-red/NIR (Pfr, $\lambda_{abs}$ ~ 750 nm) absorbing state. Canonical BPhPs adopt the Pr state in the dark and photoconvert to the Pfr state upon illumination with far-red light (peaked at 680–710 nm). From the Pfr state, NIR illumination (peaked at 740–760 nm) and/or thermal relaxation drives the BPhP back to the Pr state[23]. Two notable exceptions to this photoconversion behavior are the "bathy"-BPhP[24] and the RpBphP3 from *R. palustris*[25]. For the former, Pfr is the dark-adapted state which can photoconvert to Pr and for RpBphP3, the photoproduct absorbs at shorter wavelengths than the Pr state, in the near-red (Pnr, $\lambda_{abs}$ ~ 645 nm)[25,26].

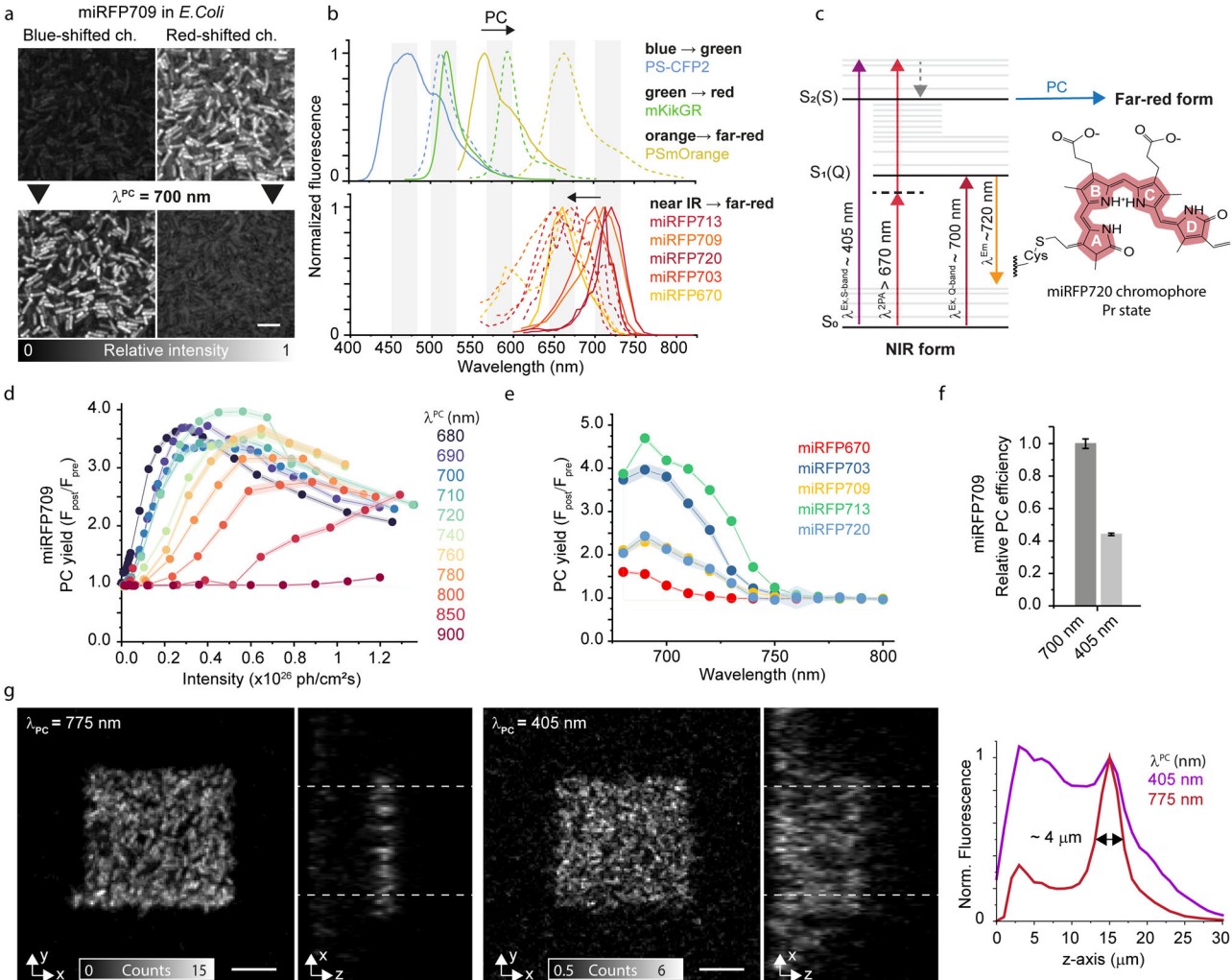

**Fig. 1 | Blue-shift photoconversion in miRFPs. a** Representative images of an experimental recording (blue-shifted channel: 580–620 nm, excitation at 570 nm; red-shifted channel: 700–750 nm, excitation at 670 nm). **b** Emission spectra for the initial (solid line) and photoconverted (dashed line) forms of different groups of PCFPs. One example for each group of GFP-like PCFPs (top). Emission spectra for the miRFP proteins expressed in bacteria before and after photoconversion (bottom). The gray blocks identify the spectral bands (~50 nm) occupied by the PCFPs. **c** Energy levels for miRFP720 with the paths of photoconversion. The BV chromophore is sketched in the Pr state. A single thioether bond attaches ring A to the Cys residue of the PAS domain in miRFP703, miRFP709, miRFP713, and miRFP720. miRFP670 has two thioether bonds from the protein to ring A. **d** Photoconversion yield for miRFP709 expressed in *E. coli* at increasing illumination intensity for a range of wavelengths at 680–900 nm. **e** Action spectra for miRFP proteins in the

blue-shifted channel. The photon flux of the photoconversion illumination was kept constant at ~ 0.1 × 10²⁶ ph/cm²s. **f** Comparison of the relative maximum photoconversion yield obtained for miRFP709 at 405 nm illumination compared to 700 nm illumination, the maximum of the action spectra. Each data point in **d**–**f** is the mean and SD for bacteria enclosed in a field of view of ~30 × 30 μm² (100–200 bacteria). In **d** and **e**, each point represents the mean, and the shaded area represents one standard deviation. In **f**, the bars represent the mean, the error bars represent one standard deviation. **g** Axial confinement dependency on the photoconversion-inducing illumination wavelength: 775 nm (120 fs, 80 MHz, left) and 405 nm (continuous wave, right). For both wavelengths, the photoconversion energy is below saturation (10 J/cm² and 3 J/cm² for 775 nm and 405 nm, respectively). The images are representative examples of 3 independent experiments. Scale bars, 5 μm. Source data are provided as a Source Data file.

Additional requirements such as higher brightness, monomeric state, yield of conversion and stability in both forms need to be met to use the Pr/Pfr spectral shift as a photoconvertible tool for biological imaging. A first attempt to engineer dark-to-bright photoactivatable NIR FPs led to PAiRFP and PAiRFP2[24]. Although their photoactivation results in an up to 20-fold fluorescence increase, their brightness is well below that of standard NIR FPs and their high sensitivity to light makes them unstable preventing their use in daylight and time-lapse experiments[24].

High effective brightness and a monomeric state have been achieved with the miRFP mutants of the BphPs of *R. palustris*[26,27]. miRFPs consist only of the PAS-GAF domains (Supplementary Fig. 11) and have fluorescence emission peaks ranging from 670 nm to 720 nm[26–28]. The absence of the PHY domain and the Asp substitution in the conserved -PXSDIP- amino acid motif of BphPs[29] stabilizes ring D of the BV chromophore in the Pr state. The spectral diversity of miRFPs is attributed to the degree of π-conjugation of ring A with the rest of the BV chromophore: it is partially reduced in the most blue-shifted variant miRFP670 while having π-π stacking with protein residues in the most red-shifted miRFP variants[22].

Here, we report on a photoconversion phenomenon common to various miRFPs (miRFP670, miRFP703, miRFP709, miRFP713, and miRFP720) that leads to a stable photoconverted species. The photoconversion can be driven by either NIR (640–780 nm) or UV/Vis (405 nm) illumination.

Because miRFPs are excited, emit and are photoconverted with NIR light, they open possibilities for photolabeling in a previously unexplored spectral region. Red-shifted illumination is advantageous for tissue and in vivo imaging, where samples are typically affected by scattering. Furthermore, the photoconversion driven by NIR illumination is found to be non-linear, thus providing optical axial confinement and minimal off-target phototoxicity[30]. It has been shown that the photoconversion of PCFPs in the visible range can also be axially confined through the primed conversion mechanism, at the expense of slower conversion efficiency relative to the photoconversion of the same PCFPs by UV light and it is therefore suboptimal for chasing fast dynamics[18]. miRFPs represent a viable option for cell and sub-cellular tracking applications featuring a faster photoconversion rate without compromising the time resolution, as with primed conversion. Additionally, we further show that the miRFP spectral shift is coupled to a fluorescence lifetime change, which can be used to further separate the photoconverted molecules from the original population enhancing the contrast. This is particularly desirable for in vivo studies over long observation time windows, where protein turnover might decrease the signal of photoconverted cells.

We demonstrate the compatibility of the miRFP photoconversion with in vivo imaging by tracking proteins and organelles in cell culture and zebrafish larvae with confocal and super-resolution stimulated emission depletion (STED) microscopy. The miRFP photoconversion experiments can be performed in commercially available microscopes, which typically host laser lines at 405 and 775 nm. Finally, we show how the spectra and kinetics of miRFPs' photoconversion are complementary to commonly used PCFPs in the visible range, allowing to multiplex photolabeling of proteins and single cells.

## Results

### Characterization of the miRFP NIR-to-far-red photoconversion

To investigate the light-driven spectral shift of several miRFP variants, we expressed them in *E. coli* bacteria. When we illuminated a selected region of interest of the bacteria pellet with pulsed light at 775 nm, an hypsochromic shift of ~53 nm in the fluorescence excitation and ~59 nm in the fluorescence emission spectra is detected (Fig. 1a, b, Supplementary Fig. 12). The same direction and length of the shift was found for the other miRFP variants (Fig. 1b, Table 1), suggesting a common underlying mechanism that is hypothesized to be

disturbance of the π-conjugated system in the BV chromophore (Fig. 1c). The photoconverted form exhibits blue-shifted spectra compared to the unconverted population, yet the spectra are still well separated from the reddest species of the GFP-like PCFPs (Fig. 1b and Supplementary Fig. 13).

The absorbance spectra of miRFPs are composed of two bands: the Q band (~640–700 nm) and the Soret band (~380–400 nm), where the latter corresponds to the absorption of individual pyrrole rings of the BV chromophore. Starting from the Q band spectral region, we investigated the efficiency of photoconversion as a function of different wavelengths for the five miRFPs variants using a femtosecond laser source (80 MHz, 120 fs pulse width, Fig. 1d). The photoconversion yield was quantified as the ratio of fluorescence intensity after and before photoconversion ($F_{post}/F_{pre}$) detected in the emission band of the photoconverted form ~ 600–700 nm (Supplementary Note 1). The highest yield was in the 680–720 nm, which is in good agreement with the miRFP absorbance (Fig. 1e).

At each wavelength of the 680–800 nm range, the photoconversion yield increased quadratically with the illumination energy (Fig. 1d and Supplementary Figs. 14–16), which suggests a two-photon absorption (2PA). As common for tetrapyrrole, the 2PA efficiency is resonantly enhanced in the 710–810 nm spectral region due to the nearby one-photon transition of the Q-band[31,32]. The photoconversion yield curve showed a similar ascendant trend until it reaches a maximum followed up by a fall. Previous ultrafast transient absorption studies[33] and 2PA cross-section estimation[34] on the dimeric iRFP, ancestor of miRFP, suggested that competing pathways such as stimulated emission and bleaching are the reason for the descendent trend, while ground state depletion, excited state absorption and 2PA promote higher yield of photoconversion (Fig. 1c, Supplementary Fig. 16). The efficiency and balance between these processes depend on the excitation wavelength. The dependency of the photoconversion yield to the wavelength reflects the resonance enhancement, with a peak position strongly influenced by the Q-band (Fig. 1e). The spectra were recorded at minimal energy (~$10^{25}$ photons/cm²s) to avoid artifacts introduced by photo-bleaching or stimulated emission. We measured a similar trend for all miRFPs, with peaks at 690–720 nm. An exception was miRFP670, where the peak shifted toward shorter wavelengths as a result of a more blue-shifted Q-band (Fig. 1e). We also recorded the related spectra of the unconverted population, which showed comparable dependence versus photoconversion's energies and wavelengths (Supplementary Fig. 14).

The Soret absorption band can be used to drive photoconversion in the UV/Vis spectral region (Fig. 1f and Supplementary Fig. 17). As opposed to the NIR-driven photoconversion, the 405 nm light promotes a linear dependency of the photoconversion yield versus illumination energy (Supplementary Fig. 16). Interestingly, the plateau level of the 405 nm driven conversion was lower than that of the Q band for all miRFPs, likely due to a higher bleaching rate (Supplementary Fig. 17).

The dual nature of the photoconversion in the two spectral regions influences the spatial confinement of the fluorescence signal. We illuminated two regions of a bacteria sample expressing miRFP713 and located at an axial depth of 15 µm from the coverglass with 775 nm (10 J/cm²) and 405 nm (3 J/cm²), respectively. As expected from a 2PA, the photoconversion at 775 nm generated a photoproduct axially confined (~4 µm) due to the non-linear dependence on the illumination energy, whereas the one-photon absorption at 405 nm photolabeled several planes (~20 µm) (Fig. 1g and Supplementary Fig. 18).

After photoconversion, the photoproduct was thermally stable for ~12 h at room temperature (Supplementary Fig. 19). We detected a 10–20% increase of the signal in the photoconverted species over the first 30 min, likely due to some relaxation from metastable dark states (Supplementary Fig. 19). Additionally, the photoconversion yield increased when bacteria are observed in an open chamber, indicating

**Table 1 | The spectral shift for miRFP proteins after photoconversion**

| NIR FP | miRFP670 | | miRFP703 | | miRFP709 | | miRFP713 | | miRFP720 | |
|---|---|---|---|---|---|---|---|---|---|---|
| | ground[a] | converted[b] | ground[a] | converted[b] | ground[a] | converted[b] | ground[a] | converted[b] | ground[a] | converted[b] |
| Ex. max [nm] | 642 | ~570 | 674 | ~630 | 683 | 630 | 690 | 630 | 702 | ~635 |
| ΔEx [nm] | 72 | | 44 | | 53 | | 60 | | 67 | |
| Em. max [nm] | 670 | ~590 | 703 | ~660 | 709 | 650 | 713 | ~655 | 720 | ~665 |
| ΔEm [nm] | 80 | | 53 | | 59 | | 58 | | 55 | |
| PC yield for $\lambda^{PC, Q\ band}$ | 3.5 ± 0.1 | | 7.6 ± 0.1 | | 3.9 ± 0.1 | | 7.8 ± 0.1 | | 4.3 ± 0.1 | |
| PC yield for $\lambda^{PC, S\ band}$ | 2.3 ± 0.1 | | 2.9 ± 0.1 | | 1.7 ± 0.1 | | 2.9 ± 0.1 | | 1.9 ± 0.1 | |

[a]Excitation and emission maxima in the ground state are indicated as measured on purified miRFP proteins in Matlashov M. E. et al., Nat. Comm. 11, 1–12 (2020).
[b]Photoconverted spectra were acquired in illuminated E.coli bacteria expressing miRFPs at 10 nm bandwidth, which provides some uncertainty for the determination of excitation and emission maxima.

an oxygen-dependent photoconversion process (Supplementary Fig. 20).

**miRFP NIR-to-far-red photoconversion: fluorescence lifetime**

In addition to the spectral change, the NIR and far-red forms of miRFP also differ in fluorescence lifetime. Common to all miRFPs studied, we observed an increase in the average fluorescence lifetime upon photoconversion: from 0.65 ± 0.05 ns to 1.10 ± 0.20 ns for miRFP720; from 0.80 ± 0.01 ns to 1.20 ± 0.20 ns for miRFP713; from 0.64 ± 0.01 ns to 1.15 ± 0.02 ns for miRFP703; and from 1.22 ± 0.01 ns to 1.80 ± 0.10 ns for miRFP670 (Fig. 2a and Supplementary Note 2).

We then focused on the miRFP720, as the reddest miRFP variant, and investigated its fluorescence lifetime using FLIM imaging[35,36] (Fig. 2b and Supplementary Note 2). To test the lifetime dependence on the spectral shift, we recorded one FLIM image at an emission band of 20 nm, from 640 to 740 nm upon excitation with 594 nm light. In the unconverted state, the mean fluorescence lifetime of miRFP720 moves along a line in the phasor plot starting from 0.6 ns in the most shifted NIR window (720–740 nm) to 1.5–2.5 ns for the far-red one (640–660 nm). Upon photoconversion, the fluorescence lifetime is less dependent on the spectral detection window compared to the unconverted population, and it is distributed toward the long lifetime extreme. This spectral dependency of the unconverted form suggests an initial heterogeneity of miRFP720 in the equilibrium state that converges in the slower form upon conversion[35] (Fig. 2c and Supplementary Note 2). Even if the second component of the ground state and the photoconverted form share similar lifetimes and spectral profiles, it is not possible to address them as the same form only with this spectral lifetime data. To confirm the results, we performed the experiments both in *E. coli* expressing miRFP720 and in mammalian cells expressing miRFP720-H2B and found that the fluorescence lifetime follows the same spectral dependency (Supplementary Note 2).

Such modulation in fluorescence lifetime is beneficial for photolabeling imaging experiments because it can be used to enhance the photoconversion contrast (Fig. 2d–g). Fluorescence lifetime is less affected by concentration variations than the ratio of intensities as metric, and therefore advantageous in, for example, tissue imaging or cases where there is a disparity in brightness for the two forms. The lifetime separation is done either on the phasor plot by graphically identifying the two fluorescence lifetime components in the fluorescence lifetime imaging (FLIM) dataset (Fig. 2d, e) or by linear decomposition of the two components of the converted miRFP fluorescence lifetime decay with average lifetimes of 0.65 ns and 1.1 ns before and after photoconversion (Fig. 2f, g). Both strategies allowed to recover the extent of the photoconverted area on a layer of *E. coli* expressing miRFP720 (Fig. 2e, f). By lifetime decomposition, the raw data and the image derived by the two components can be directly separated and compared. This resulted in a doubling of the measured effective photoconversion yield as compared to solely spectral separation (Fig. 2g and Supplementary Note 2).

**Photoconversion multiplexing**

The photoconvertible miRFPs diversify from the reported GFP-like PCFPs in spectral properties, photoconversion light and kinetics, the direction of the spectral shift upon conversion and fluorescence lifetime. Overall, those properties are not only appealing for imaging applications in need of red-shifted light but enable multiplexed experiments complementing well previously available PCFPs. We characterized side-by-side the photoconversion yield of NIR-to-far-red (such as miRFP713 and miRFP720) and green-to-red PCFPs (such as mEos and Dendra2) to design pulse schemes enabling their combined use in bioimaging (Fig. 3). The emission spectra of Dendra2 and mEoS in their converted (570–600 nm) and unconverted (500–550 nm) species can be separated from the miRFP720 and miRFP713 converted (640–680 nm) and unconverted (710–750 nm) (Fig. 3a).

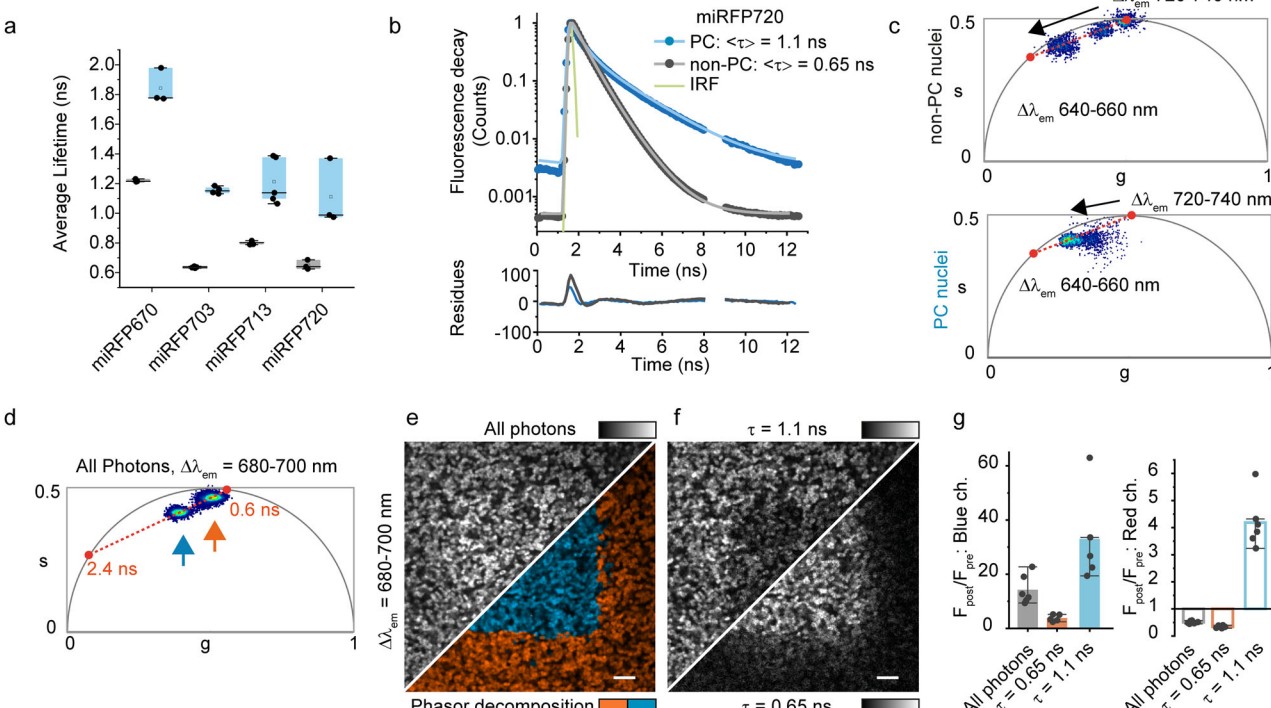

**Fig. 2 | Lifetime characterization of the miRFP photoconversion. a** The lifetime for the ground (gray) and photoconverted (blue) forms of different miRFPs is reported ($N = 3–5$ repetitions, boxes mark the IQR, line defines the median point, gray dot marks the mean, whiskers mark the maximum and minimum values). The photoconversion is triggered by 405 nm illumination at 4.8 J/cm². **b** Representative fluorescence lifetime decay for the far-red form (620–650 nm detection band, red dots) and NIR form (700–730 nm detection band, blue dots) of miRFP720 expressed in *E. coli*, together with the instrument response function (IRF) of the system (green line). **c** Spectral lifetime exploration for miRFP720 labeling of H2B in HeLa cells. For a non-photoconverted and photoconverted (405 nm light at 5 J/cm²) nucleus, a fluorescence lifetime image has been recorded spectrally from 640 to 740 nm, integrating intervals of 20 nm (excitation at 594 nm). The phasor plot reports the information at all the spectral window (after 8 pixels binning of the images). The two extremes of the orange dotted line correspond to lifetimes of

0.6 ns and 1.5 ns. The arrow follows the progression of the phasor in relation to the spectral progression. **d** Phasor plot for a layer of bacteria photoconverted only in a central square through 405 nm light in an emission region (**e**) where the photoconverted and non-photoconverted forms of miRFP720 overlap and therefore are indistinguishable by fluorescence intensity (680–700 nm). The two components identifiable in the phasor plot are highlighted in blue (slow component, photoconverted form) and orange (fast component, non-photoconverted form). Scale bar, 5 μm. **f** The same dataset has been decomposed according to the two identified lifetime components and the relative change of fluorescence is reported for the fast (0.65 ns) and slow component (1.1 ns). **g** Quantification of the change in fluorescence for the blue-shifted (filled bars, left) and red-shifted channel (hollow bars, right) (mean ± SD for $n = 6$ independent repetitions). Scale bars, Source data are provided as a Source Data file.

The separation between the two families of PCFPs extend also to the fluorescence lifetime, with green-to-red characterized by fluorescence lifetime of ~2–3 ns and miRFP characterized by sub-ns lifetime (Supplementary Fig. 21). Among the miRFP variants, miRFP720 and miRFP713 are of particular interest since they are not only the reddest-shifted variant but they also show the highest effective brightness in mammalian cells[28].

UV/Vis light at 405 nm is commonly used to drive the photoconversion of green-to-red PCFPs and can also be used for miRFP even if less efficiently than NIR light (Fig. 3b and Supplementary Fig. 22). We measured different kinetics of photoconversion at 405 nm for Dendra2 and miRFP713, which enabled to selectively photoconvert them by tuning the energy. At low illumination energy (<1 J/cm²) only Dendra2 is photo-shifted. Instead at higher energy (4–6 J/cm²), both proteins can be photoconverted (Fig. 3b). The NIR photoconversion can only convert the miRFP720, except when used in combination with blue light at 488 nm through primed conversion (Fig. 3c). We measured a 10-fold lower photoconversion efficiency for miRFP713 compared to Dendra2 upon NIR. Therefore, only miRFP720 is selectively photoconverted at illumination energy <1 kJ/cm² while >10 kJ/cm² both PCFPs are equally photoconverted.

Based on the investigated photoconversion kinetics elicited by NIR and UV/Vis light, we designed multiplexed schemes with spatio-temporal coupling or decoupling of the photoconversion (Fig. 3d, e).

On a mixed layer of *E. coli* expressing miRFP713 and Dendra2, the two-cell population was photo-labeled independently (Fig. 3d) using a low level of UV/Vis light (1.5 mJ/cm²) for Dendra2 and 775 nm (1.4 kJ/cm²) for the miRFP.

Other types of multiplexed experiments might require to photo-label of both bacterial populations in the same spatial location (Fig. 3e). With this purpose, we illuminated the bacteria with 775 nm (16 kJ/cm²) and simultaneously with 488 nm (4 mJ/cm²) observing photoconverted products from both Dendra2 and miRFP713.

To test phototoxicity, we monitored growth of *E. coli* colonies over 100 min after photoconversion (Fig. 3f and Supplementary Note 5). At the energy needed to saturate the photoconversion of miRFP720 ($\lambda^{PC} = 775$ nm) or Dendra2 ($\lambda^{PC} = 405$ nm), no alteration of the bacterial growth was observed. Similarly, upon 775 nm illumination, we did not observe a decrease in cell viability in HeLa mammalian cells (Supplementary Note 5).

The miRFP photoconversion can also be effectively performed in mammalian cells (Supplementary Fig. 23). The relatively fast kinetic of the miRFP photoconversion enabled to perform multiplexing experiments in living systems, tackling rapid dynamics (<min) of subcellular organelles (Fig. 4). After one frame of illumination with 405 nm light at 4.7 J/cm², a 15-fold average increase of fluorescence intensity in the blue-shifted channel was observed for miRFP720 labeled to different protein of interest (Supplementary Fig. 24). The contrast in the blue-

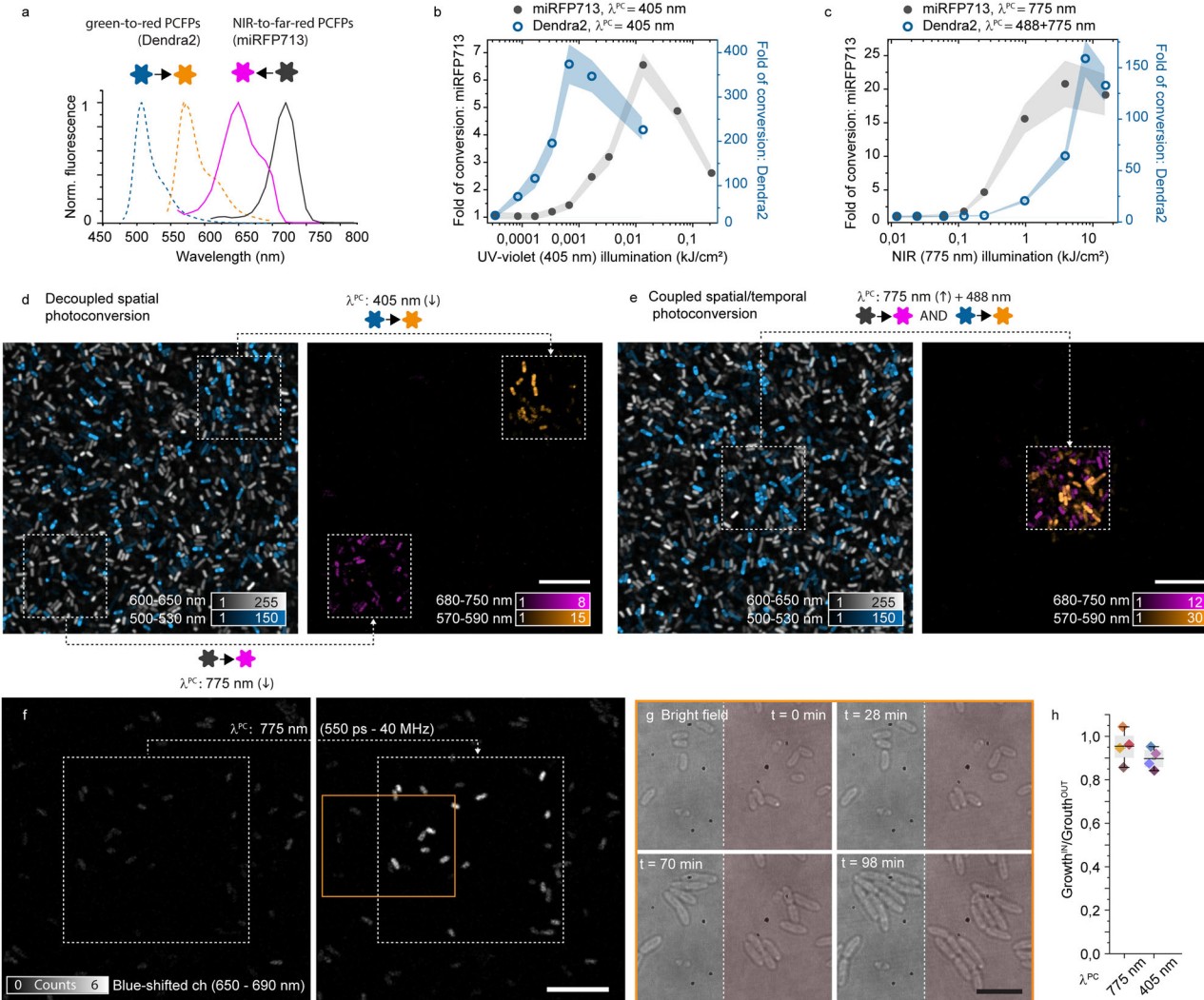

**Fig. 3 | Multiplexed photoconversion strategies for green-to-red and NIR-to-far-red PCFPs. a** Emission spectra for Dendra2 and miRFP713 before (solid) and after (dashed) photoconversion. **b, c** Efficiency of photoconversion upon various illumination for miRFP713 (dark gray) and Dendra2 (blue). **b** The fold of conversion for the two proteins is compared over the same energy range of 405 nm illumination (CW). **c** The energy dependence of the conversion yield for miRFP with 775 nm (700 ps, 40 MHz) illumination is compared to the primed conversion for Dendra2 with fixed 488 nm (CW, 0.13 kW/cm²) and increasing 775 nm (700 ps, 40 MHz) illumination. The comparisons in (**b, c**) have been performed on a layer of *E. coli* expressing miRFP713 or Dendra2. Each data point is the mean and SD for bacteria enclosed in a field of view of ~ 15 × 15 μm² (=50–100 bacteria). Representative example of 3 replica for spatially uncoupled (**d**) and coupled (**e**) photoconversion. Scale bars, 5 μm. **d** Using photoconversion wavelengths and illumination conditions unique for the PCFPs, like pulsed 775 nm for miRFP and low energy of 405 nm light for the green-to-red PCFPs, two different spatial locations (within the dotted

square) can be selectively photoconverted in the field of view. **e** The spatially localized illumination with a combination of high 775 nm and 488 nm light (within the squared dotted area) photoconverts both PCFPs in the same area. Scale bar, 10 μm. **f** *E. coli* expressing miRFP720 photoactivated with 775 nm light at saturation level in a region of around 25 μm (dotted square) and observed in the blue-shifted channel (650–690 nm). **g** Brightfield images of four-time points in the 100 min observation window for the inset in orange. The red square marks the photoconverted area. Scale bars, 10 μm. **h** Ratio of growth between the photoconverted and non-photoconverted area illuminated with 775 nm (with the energy required to saturate the miRFP720 photoconversion) or 405 nm (with the energy required to saturate Dendra2 photoconversion). The box plot reports 4 independent repetitions (box encloses the 25th to 75th percentile, whiskers include all values and line shows the mean, 0.95 and 0.90, respectively) of which the images in **f** and **g** are examples. Source data are provided as a Source Data file.

shifted channel and the photoconversion yield were minimally affected by crosstalk from the red-shifted form. The fluorescence bleed-through generated by the converted form in the red-shifted channel was minimized by selecting the excitation wavelengths >670 nm (Supplementary Note 1).

We applied the coupled photoconversion imaging scheme using UV/Vis illumination to track peroxisomes and lysosomes in Hela cells expressing mEos2 and miRFP720, respectively (Fig. 4a). In case of interference between the photoconverted channels of the two PCFPs, spectral unmixing (Fig. 4a) or lifetime separation (Fig. 4b) could be used to disentangle the signals (Fig. 4c). Nevertheless, a sequential

recording strategy generally minimized the crosstalk of the red-shifted mEos form in the miRFP far-red channel (Fig. 4d and Supplementary Note 3). By shining relatively low doses of light at 405 nm (3–5 J/cm²), more than 80% of the mEos and miRFP720 proteins were converted (Fig. 4a–c), resulting in photolabeling of organelles at cell periphery. These subsets of lysosomes and peroxisomes were tracked over time while redistributing within the whole cell (Fig. 4f–h and Supplementary Figs. 25–26). This multiplexing strategy enabled the study of organelle fusion and dynamics with an imprinted temporal (time of conversion) and spatial information (cellular subregion of origin) (Fig. 4i).

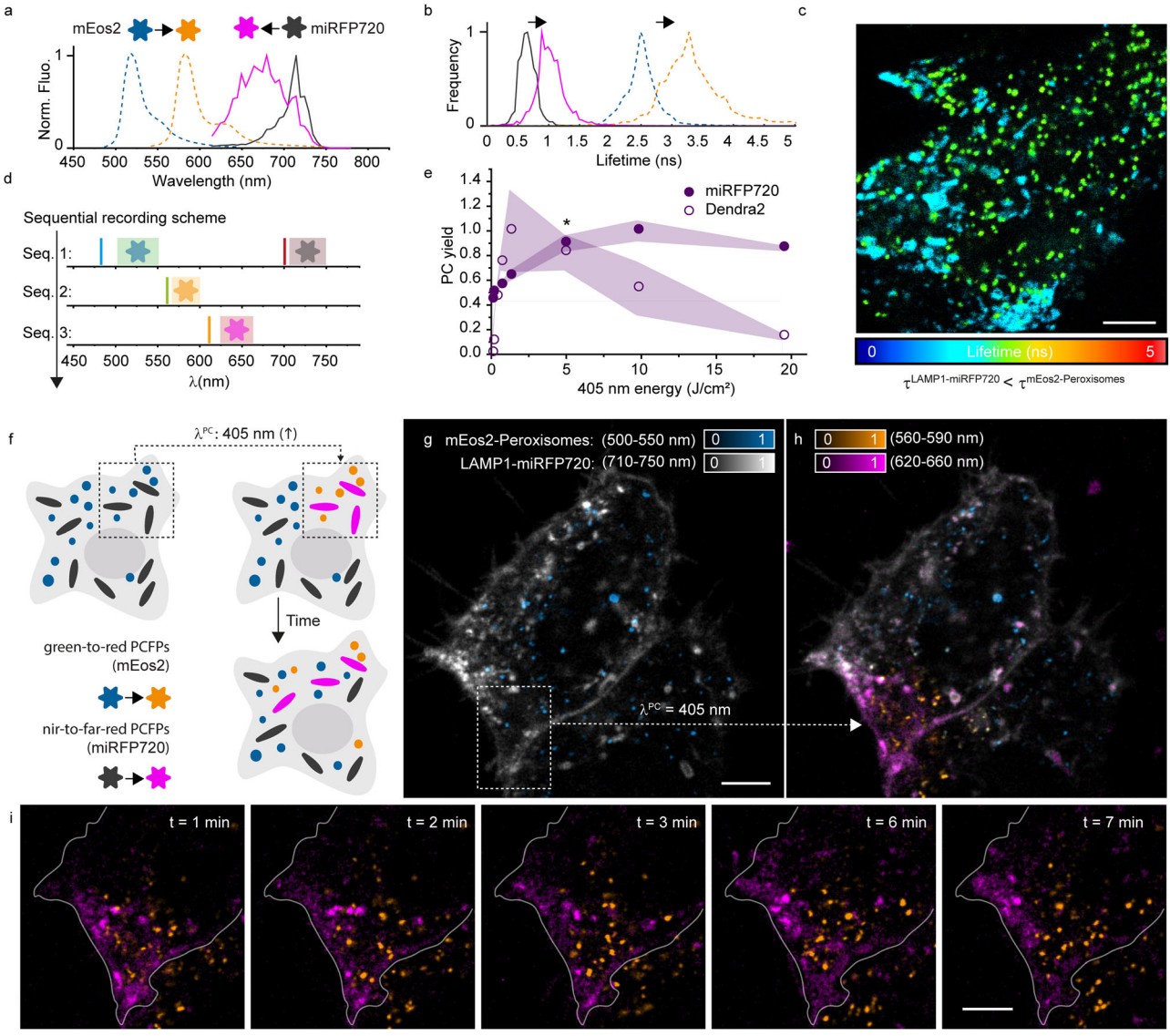

**Fig. 4 | Multiplexed photoconversion of green-to-red and NIR-to-far-red PCFPs for organelle tracking.** Assessment of the (**a**) spectral, (**b**) fluorescence lifetime compatibility of miRFP720 (gray-to-magenta color) and mEos2 (blue-to-orange color) when expressed in Hela cells. The fast FLIM lifetimes are representative of cells expressing mEos3.2-clathrin and LAMP1-miRFP720. **c** FLIM image reporting the crosstalk between LAMP1-miRFP720 and mEos2-Peroxisomes in the 620–670 nm detection band. The fluorescence lifetime information encoded in the colormap allows to distinguish the two proteins and, therefore, correctly segment the two labeled organelles. **d** Sequential scheme of recording, for the multicolor photoconversion imaging. The excitation and detection band are color coded to reflect the signal of the protein that is predominant under that excitation and detection condition. **e** Reactivity to the 405 nm light as photoconversion wavelength measured in Hela cells expressing LAMP1-miRFP720 (filled circles) and mEos3.2-clathrin (empty circles). The reactivity to 405 nm illumination is measured as the ratio between the fluorescence of the photoconverted state and the ground state. The graphs are normalized for comparison between the two PCFPs. The

asterisk identifies the 405 nm power used to photoconvert both proteins simultaneously. Each data point is the mean and SD of the photoconversion yield for $N = 25$–50 organelles. **f** Illustration of coupled photoconversion multiplexing scheme with a common PC wavelength. Through illumination with 405 nm light, green-to-red and NIR-to-far-red PCFPs can be combined to achieve spatially colocalized photoconversion at the subcellular level. **g** Coupled photoconversion experiment in HeLa cells, with the labeling of peroxisomes, mEos2-Peroxisomes, and lysosomes, LAMP1-miRFP720. Before photoconversion, all the fluorescence was emitted by the green (500–550 nm, cyan) and NIR (710–750 nm, gray) forms. **h** Upon illumination with 405 nm light (3.16 J/cm²), a confined region (dotted rectangle in **g**) of the cells was photolabeled, where the two proteins photoconverted to the red (570–590 nm, orange) and far-red (620–660 nm, magenta) forms, respectively. **i** A closer look at the photoconverted region and motility of the organelles disentangled over time for the photoconverted area (time interval 1 min). Scale bars, 5 μm. Representative images of 5 experiments. Source data are provided as a Source Data file.

## miRFP NIR-to-far-red photoconversion: combination with STED microscopy

The high effective brightness, photostability, and monomeric state of miRFPs make them suitable probes for STED microscopy[28]. In the NIR range, STED can be performed using a depletion wavelength of 775 nm (40 MHz, 550–700 ps pulse width) with an illumination dose close to that needed for photoconversion (Fig. 5a, b and Supplementary Note 4). Interestingly, when we performed STED imaging, we observed

that the super-resolved spatial information is predominantly coming from the blue-shifted form (Fig. 5c). This distinction was invisible in the former study[28] where the fluorescence of miRFPs was detected in a broad 655–755 nm detection bandwidth, integrating the contributions of the converted and unconverted population[29].

The 775 nm fiber laser (40 MHz, 550–700 ps pulse width) commonly used in STED microscopy could similarly drive the blue-shifted photoconversion as the femtosecond pulsed laser

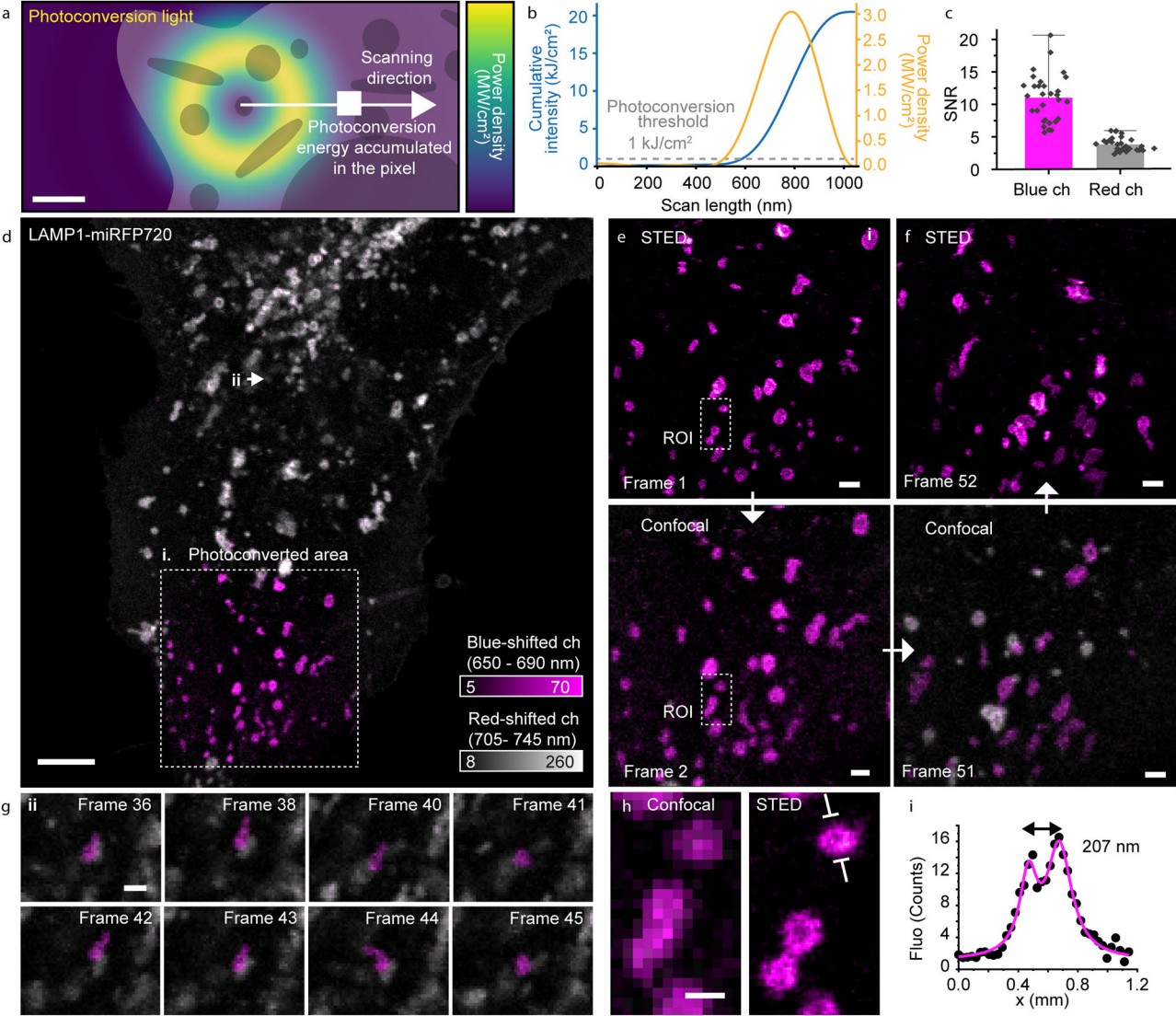

**Fig. 5 | Photoconversion of miRFP720 in STED microscopy. a** Representation of the 775 nm depletion beam during the scanning process in STED image acquisition. A region of the sample will be exposed to all the light of one crest of the depletion beam light pattern before ending up in the center of the donut where its emission photons will be registered. Scale bar, 250 nm. **b** Building up of the energy level (blue line) based on the previously described scanning of the depletion beam (the profile of which is reported as the yellow line). The dashed gray line marks the energy threshold for the photoconversion to happen. The donut is calculated for the intensity of the 775 nm light at the back aperture of 15 mW. **c** Signal-to-noise ratio in the blue channel (650–690 nm) and the red channel (705–745 nm), extracted from $N = 29$ filaments of vimentin-miRFP720 (Supplementary Fig. 28). Bars represent the mean value; error bars represent one standard deviation. **d** Lysosomes labeled with miRFP720, LAMP1-miRFP720, in live U2OS cells and imaged in the blue-shifted channel (650–690 nm, magenta) and red-shifted channel (705–745 nm, gray).

A $20 \times 20\ \mu m^2$ area was photoconverted by recording a STED image with a cumulative depletion illumination intensity of ~1 kJ/cm² at each point. For the photoconverted area, the first (**e**) and last (**f**) frames of a time-lapse of 52 consecutive frames are shown in confocal and STED. Time interval in the confocal time-lapse is 10.6 s. **g** Dynamics of the interaction between a photoconverted (magenta) and a non-photoconverted lysosome (gray) outside of the photoconverted area. **h** Zoom-in to a cluster of vesicles in a photoconverted area, where the increased spatial resolution allowed distinguishing them and their hollow structure. Scale bar, 5 μm (**d**), 1 μm (**e**–**g**) and 500 nm (**h**). **i** The intensity profile was evaluated across the region highlighted with white arrows in (**h**) and averaged over 100 nm in width. The double peak has been fitted with two Lorentzian functions to localize the membranes. Representative images of 3 experiments. Source data are provided as a Source Data file.

(Supplementary Fig. 15). However, an increase of ~50 times the average power of the NIR illumination is needed, going from 4 J/cm² for femtosecond pulsed (120 fs) to 200 J/cm² for picosecond pulsed (550 ps)[37] NIR light to reach saturation of the photoconversion.

By combining STED microscopy and photoconversion, a region of interest can be imaged at higher resolution in the first frame (Fig. 5d, e) where at the same time a blue-shift is induced, whereby the photoconverted structures can be followed subsequently over the entire field of view using confocal imaging. Finally, the entire field of view can be recorded at a higher resolution again using STED imaging. We used this imaging strategy to follow the dynamics of lysosomes tagged with

miRFP720 (Fig. 5d–h). Over a time-lapse of 9 min (51 frames, 10.6 s time interval), the photoconverted population of lysosomes at the cell periphery gradually populated the whole volume. Interaction between photoconverted and non-photoconverted lysosomes was also observed (Fig. 5g and Supplementary Fig. 27). STED imaging could resolve single organelles and the outer membrane localization of LAMP1, showing lysosomes as hollow structures (Fig. 5h). Another example where the blue-shift of miRFPs could be used to decouple the motion of highly packed and mobile cellular structures was the vimentin network, which could be finely resolved with STED microscopy[28] (Supplementary Fig. 28) Dynamic network rearrangement could be tracked over time

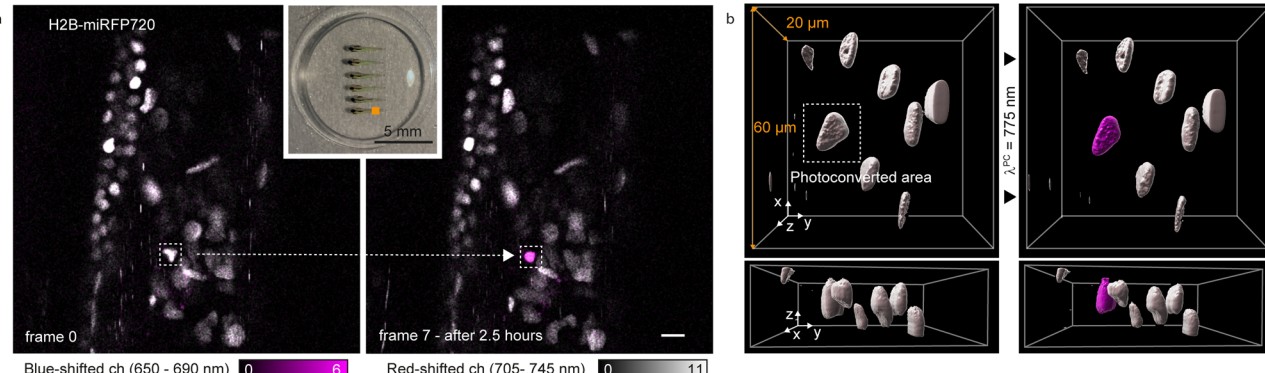

**Fig. 6 | Photoconversion of miRFP720 in vivo. a** Photoconversion in zebrafish larvae with a mosaic expression of miRFP720-tagged histones, H2B-miRFP720. The area of imaging corresponds to the aorta-gonad-mesonephros region. Time-lapse recording over 2.5 h in blue-shifted (650–690 nm, magenta) and red-shifted (705–745 nm, gray) detection channels, after selective photoconversion of one specific nucleus. Scale bar, 10 μm. **b** Volumetric recording before (left) and after (right) photoconversion with 775 nm light. Source data are provided as a Source Data file.

with photolabeled and non-photolabeled filaments moving in and out of the photoconverted region (Supplementary Fig. 28).

### miRFP NIR-to-far-red photoconversion: in vivo

We used zebrafish as a proof-of-principle to test the applicability of our method in a multicellular organism. We injected in 1-cell stage zebrafish zygotes a Tol2-based plasmid carrying the nuclear histone H2B-miRFP720 fusion protein under the expression of a ubiquitous promoter (*ubb* promoter), and we successfully observed labeled nuclei in Zebrafish larvae (Fig. 6). We tested the photoconversion in living day 4 post-fertilization zebrafish larvae. We focused on the aorta-gonad-mesonephros area and selectively photoconverted a single nucleus of the tissue (Fig. 6a). The axial confinement could be confirmed on a volumetric recording (Fig. 6b), and the signal could be followed over an extended period of 2.5 h under live-cell imaging conditions (Fig. 6a and Supplementary Fig. 29). Single nuclei were photoconverted without damaging the tissue and adjacent blood vessels (Supplementary Figs. 30–31).

## Discussion

We have identified a photoconversion phenomenon in the miRFP family of NIR FPs and characterized the spectral properties and kinetics of the photoconverted miRFP species.

The miRFP family is the result of extensive engineering efforts toward maximized effective brightness, monomeric state[38], and spectral tuning[26]. As the brightest genetically encoded NIR FPs they have been extensively used in live-cell imaging, from large-scale tissue imaging down to sub-cellular nanoscale imaging[28]. Here, we have observed that upon selective illumination in the characteristic bands of absorption of BphPs (Soret band ~405 nm and Q band ~700 nm), miRFPs ($\lambda_{em}$ ~ 670–720 nm) are stably photoconverted to a spectrally-shifted form ($\lambda_{em}$ ~ 590–665 nm). The converted proteins exhibited an increase of the fluorescence lifetime, from sub-nanosecond to 1.0–1.2 ns. All the miRFPs followed the same trend except for miRFP670, which unconverted has a fluorescent lifetime of 1.2 ns and after photoconversion increases to 1.8–2.0 ns. The photoconverted form originates from an excited state that can be reached either by 1PA in the UV/Vis (corresponding to the Soret band) or by 2PA achieved through 680–780 nm pulsed illumination. Among the two, the non-linear path at >700 nm generates a higher conversion yield than the UV/Vis.

The miRFPs are the most red-shifted PCFPs reported, with miRFP720 shifting from an emission peak at 720 nm in the unconverted state to 665 nm in the photoconverted state. Besides the spectral emission window, miRFPs can be also efficiently photoconverted with NIR light. The non-linear nature of the

photoconversion at 700 nm enables to confine the fluorescence signal axially, which facilitate single cell photolabeling as well as imaging experiment requiring optical sectioning. Once converted, the miRFPs are stable for several hours enabling live-cell and in vivo imaging. For challenging imaging applications in vivo where fluorescence intensity changes, often influenced by low signal-to-noise ratios, brightness disparity, or possible fluorescent protein turnover occur, the reported change in a fluorescent lifetime can be used to spot and enhance the converted population.

Phototoxicity can be reduced by using NIR light illumination, especially to drive the photoconversion where the reported non-linear process limits off-target photobleaching. Indeed, the illumination dose needed for the photoconversion, ~2.6 J/cm² at 700 nm, is orders of magnitude below the photodamage threshold of $5 \times 10^4$ J/cm² for DNA[30,39,40] and in the range of illumination doses normally used for two-photon microscopy and STED microscopy, where similar laser sources and microscope acquisition setting are used (Supplementary Note 5). Furthermore, less UV/Violet light reduces the risk of transcriptomic alterations, pushing the synergy between PCFPs and single-cell RNA sequencing techniques where photoconversion is crucial for obtaining spatial and functional information[41].

The relatively high photoconversion rate enabled fast recording, i.e., exposure/dwell times in the μs range. This was pivotal to tracking organelle over time in live cells or to combining photolabeling and STED nanoscopy. NIR light at 775 nm is a standard laser line for commercially available STED microscopes and can be used for miRFP photoconversion (Supplementary Note 4).

miRFPs occupy a spectral region without overlaps with GFP-like PCFPs, neither in the unconverted nor in the photoconverted form, which enabled their combined use in bioimaging experiments. While the two groups of PCFPs share the possibility to be converted with 405 nm, the different rates can be used to convert distinct PCFPs within the same region of the sample. This enables to design multiplexing imaging schemes with either spatiotemporally coupled or decoupled photoconversion. On the one hand, the possibility to photoconvert miRFPs at 405 nm with comparable illumination doses as for the majority of GFP-like PAFPs and PCFPs (~1–5 J/cm² at 405 nm), but with distinct excitation and emission spectra for all four forms, allows for spatiotemporally colocalized photoconversion of multiple protein populations. On the other hand, the possibility of combining two PCFPs converted by illumination at different wavelengths and illumination intensities allows for entirely spatiotemporally independent photoconversion, highlighting interactions between distinctly labeled proteins or temporally marking the same labeled protein in different time points (Supplementary Fig. 32).

The photoconversion yield of miRFP is currently lower than the one reported for GFP-like PCFPs. Nevertheless, future tailored engineering efforts have the potential to further optimize the miRFP toward higher photoconversion yield, firstly by minimizing crosstalk and heterogeneity in the equilibrium state. From our characterization, we can draw hypotheses on the possible mechanism behind the photoconversion that could guide the mutagenesis of optimized NIR PCFPs. On the molecular scale, a fundamental step for the optimization of miRFPs as fluorescent tags has been the suppression of the typical photoswitching behavior of BphPs and the stabilization of the protein in the Pr form by the formation of the strong hydrogen bond between ring D and the histidine residue introduced into the immediate chromophore environment. For this reason, the miRFP absorption should not result in the rotation of ring D observed in BphPs[29]. Nevertheless, it may still cause BV chromophore movement at the initial picosecond timescale similar to that observed in BphPs truncated to the PAS-GAF domains[42]. In the previous study, it was found that not only ring D but also ring A of the BV is displaced, accompanied by the photodissociation of the conserved A-ring-bound water molecule[42]. This initial chromophore movement further caused the transient displacement of residues in the BV-binding pocket, including that of the -PXSDIP- and -SPIH- motifs. We hypothesize that one of the displaced rings may form a transient bond with conserved residues, such as Ser or His, decreasing the BV chromophore π-conjugation. Indeed, the thermal stability of the photoconverted state of the miRFPs suggests a covalent modification of BV. This hypothesis is supported by the SDS-PAGE and $Zn^{2+}$-staining (Supplementary Fig. 33), where the photoconverted form of miRFP720 exhibited higher electrophoretic mobility. As has been previously shown[22], this can be attributed to the formation of an additional bond between the protein and the BV chromophore. The photoconverted states of the four reddest miRFPs have similar excitation (630–635 nm) and emission (650–660 nm) maxima, suggesting the similarity of their photoconverted BV chromophores. The exception is the bluest miRFP670 protein, in which Cys in the -SPCH- motif is covalently bound to ring A, eliminating one double bond from the chromophore π-conjugation system[22].

In conclusion, we have demonstrated that miRFPs can function as photoconvertible genetically encoded probes in the NIR spectral range suitable for precise non-invasive photolabeling and spatiotemporal tracking of tagged proteins, organelles, cells and tissues in a variety of bioimaging applications.

## Methods

### Photoconversion with femtosecond laser source on bacteria

The photoconversion experiments have been performed on a Leica SP8 equipped with a Ti:Sapphire laser (SpectraPhysics INSIGHT DUAL X3) tunable between 680 and 1300 nm. The pulse width of this laser source was 120 fs with a repetition rate of 80 MHz. The objective used was a Leica $63 \times 1.4$ NA oil. The conversion was evaluated by monitoring the fluorescence variation before and after exposing the same field of view to the light of different wavelengths. Over the sequence of confocal images, the dwell time was constant, 3.16 μs, and a pixel size of 72 nm was used to scan an area of $36 \times 36$ μm². The recording of the two channels was sequential and completely decoupled, both in excitation and detection windows. In particular, the red-shifted channel was excited at 670 nm and detected at 710–750 nm for miRFP720, miRFP713, miRFP709 and miRFP703, the excitation and detection were moved to 640 nm and 650–690 nm for miRFP670. The blue-shifted channel was excited at 610 nm and detected at 620–650 nm for miRFP720 and miRFP713, exited at 590 nm and detected at 600–640 nm for miRFP709 and miRFP703, and excited at 550 nm and detected at 570–600 nm for miRFP670. The photoconversion wavelength was tuned between 680 and 900 nm, maintaining the same power after the objective. Measuring the power at the objective can underestimate the power of the focal spot, but it gives an accurate relative comparison between the different conditions explored in this study. The intensity increase in the blue-shifted channel and decrease in the red-shifted channel have been studied as functions of changing the wavelength and the illumination power at a fixed wavelength. In the latter case, the photoconversion yield was presented as a function of the photon flux to allow the comparison between different wavelengths. A diode laser was used to assess the photoconversion at 405 nm (50 mW, continuous wave). The photoconversion as a function of the intensity was measured by varying the laser power over a wide range. In contrast, the degree of photoconversion for a given wavelength was assessed at $2 \times 10^{25}$ photons/cm²s, corresponding to the plateau level for the power series associated with that wavelength.

### Emission and excitation spectra

The excitation and emission spectra for miRFPs have been recorded on the same Leica SP8. The spectra have been recorded in bacteria expressing the miRFP of interest before and after photoconversion (at saturating power). The emission spectrum was detected in the range of 600–780 nm with 10 nm bandwidth steps while excited at 550–590 nm (in particular, 550 nm for miRFP670, 570 nm for miRFP709 and miRFP703, and 590 nm for miRFP713 and miRFP720). The detection of the system was limited to 775 nm. This can result in an artificial shift of the peak for the more red-shifted variants. To prevent systematic bias, the estimation of the blue-shift was calculated considering the value reported in literature[43] as starting wavelength. The excitation spectrum was recorded with excitation in the range 470–670 nm with 10 nm steps and detection in the 700–750 nm window (except for miRFP670 where the detection interval 650–700 nm was used). The emission spectrum was detected in the range 550–780 nm for the blue-shifted form with 10 nm bandwidth steps while excited at 500 nm. The excitation spectrum was recorded in the range 470–645 nm with 10 nm steps and detected in the 649–660 nm window. The miRFP720 spectra have been recorded on a different Leica SP8 system, the same used for the photoconversion in live-cell experiments, where the limit in detection was set at 750 nm, resulting in the cut of the emission peak. The specific imaging parameters are collected in Supplementary Table 1.

### STED microscopy setup

STED imaging has been performed with a custom-built STED set-up[44]. The miRFP variants were excited with a 640 nm pulsed diode laser (LDH-D-C-640, PicoQuant) and subsequently depleted with a 775 nm pulsed (width of 550 ps) fiber laser (KATANA 08 HP, OneFive), both operating at 40 MHz. The donut-shaped depletion beam at the focal plane was created using a spatial light modulator (LCOSSLM X10468-02, Hamamatsu Photonics). The excitation and depletion laser beams were coupled together and scanned over the sample using fast galvanometer mirrors (galvanometer mirrors 6215H and servo driver 71215HHJ 671, Cambridge Technology). The laser beams were focused onto the sample using an HC PL APO 100× 1.40 NA oil STED white objective lens (15506378, Leica Microsystems), through which also the fluorescence signal was collected. After de-scanning and decoupling, the fluorescence signal passed through a common confocal pinhole (1.28 Airy disk units) to then be split by a dichroic mirror (Di02-R635-25 × 36, Semrock) in two channels, 650–690 nm (GT670/40 M, Chroma) and 705–745 nm (FF01-725/40–25, Semrock), both equipped with a notch filter (NF03-785E-25, Semrock). The signal was finally recorded by two free-space APDs (SPCM-AQRH-13-TR, Excelitas Technologies). The microscope was controlled through two separate software; image acquisition and some hardware control were done through the ImSpector software (Max-Planck Innovation, Göttingen, Germany) while the rest of the hardware control (SLM, focus lock and 775 nm laser) was done through Python-based custom-written microscope control software ImSwitch (https://github.com/kasasxav/ImSwitch)[45].

The specific imaging parameters are collected in Supplementary Table 1.

## STED imaging

Single-protein imaging of miRFP720 was done with a 640 nm excitation laser power of 5.6–26.1 µW and a 775 nm depletion laser power of 12–25 mW, both measured at the first conjugate back focal plane of the objective. The pixel size for the STED images was set to 29 nm and for the confocal images 125 nm. The pixel dwell time for the STED images was around 50 µs and for the confocal images 10 µs. All images shown are raw data. The STED images are displayed with a Gaussian smoothing of 0.65 pixels to help visualization.

## Zebrafish larvae imaging

For the imaging of zebrafish larvae, a glycerol objective (HC PL APO 93×/1.3 N.A. Glycerol STED White motCORR objective lens (Leica Microsystems) has been used in the custom-built STED microscope. The photoactivation of the nuclei has been achieved by illuminating 3 planes at a 1 µm axial distance enclosing the nucleus of interest (around $8 \times 8$ µm²). For the photoconversion around 20 MW/cm² of 775 nm light has been scanned over the area with a pixel size of 60 nm and pixel dwell time of 100 µs. The time-lapse recording has been acquired at room temperature with intervals of 30 min between acquisitions.

## Blue-shift at 775 nm

The characterization of the photoconversion at 775 nm has been predominantly performed with the custom-built STED microscope presented before. Here, the SLM was controlled to generate a Gaussian profile instead of a donut shape. The laser source used is a 775 nm pulsed fiber laser, with 550 ps pulse width and a 40 MHz repetition rate. To quantify the blue-shift induced by light at 775 nm, we monitored the variation of fluorescence for confocal images in two detection windows, a blue-shifted channel (650–690 nm) and a red-shifted channel (705–745 nm), before and after illumination with 775 nm light and calculated the ratio between them ($F_{post}/F_{pre}$). The dwell time of the confocal images (100 µs) and the power of the excitation (2.3 kW/cm²) were kept constant. The fluorescence of all individual bacteria in a field of view of $15 \times 15$ µm² (pixel size 60 nm, frame time 6.5 s) was averaged to calculate the global fluorescence (mean ± std). For the photoconversion's power dependence, the 775 nm light has been increased to 50 MW/cm². The initial part of the power-dependent series has been fitted to quantify the dependence of the blue-shift as a function of the photoconversion wavelength intensity.

To see the stability of the photoconverted blue-shifted form, we monitored the fluorescence in both channels for sequential frames after one initial 775 nm activation step (21.2 MW/cm², corresponding to the plateau level for the yield of photoconversion). The stability has been tested both for subsequent recordings corresponding to a minute time scale or with a frame interval of 10, 20, and 30 min. To exclude the influence of bleaching in assessing the stability of photoconversion, fluorescence in the $10 \times 10$ µm² areas was corrected for the bleaching induced by the 640 nm, using the surrounding non-photoconverted area as a reference. The signal was finally normalized to the level of fluorescence before photoconversion.

Further characterization to compare the photoconversion of miRFP720 or miRFP713 relative to Dendra2 with NIR light has been performed.

## Photoconversion in living cells and FLIM analysis

To compare and combine the photoconversion of spectrally distinct PCFPs, a modular Leica STED and TCS SP8−FALCON (FAst Lifetime CONtrast) system (Leica Microsystems GmbH, Wetzlar, Germany) equipped with a 100 × 1.4 NA STED WHITE oil objective was used.

The miRFP720 and miRFP713 NIR form was excited at 670 or 633 nm and the fluorescence was recorded in the interval 710–750 nm, while the far-red form was excited at 610 or 594 nm and its fluorescence was recorded in the interval 620–670 nm. Experiments varying this bandwidth (either 620–670 nm or 640–680 nm) were also reported to test and characterize the crosstalk between the photoconverted forms. For Dendra2, mEos3.2, and mEos2 the green form was excited at 488 nm while the orange at 561 nm, and the emission intervals were 500–550 nm and 570–600 nm, respectively. The images were recorded at a pixel size of 57–83 nm and a dwell time of 3.12 µs. The photoconversion was driven by 405 nm light, with both intensity and line average used to tune the intensity (ranging at around 3–5 J/cm²). Similarly, the 775 nm STED beam of the system (700 ps, 80 Mz) has been used to trigger the photoconversion in the NIR range. The pulsed 775 nm light in combination with 488 nm light was used for primed conversion of the green-to-red variants. Although less efficient than illumination with continuous wave light this configuration allows the exploration and integration of the mechanism in the commercial microscope used. The photoconversion has been implemented sequentially either in a frame-by-frame or line-by-line fashion. At first, the pre-converted forms for the two proteins were acquired simultaneously, given their clear spectral separation. Secondly, the orange form was recorded and finally, the red form. The frame-by-frame approach was used to retrieve the characteristics of photoconversion, such as yield, power dependency, and lifetime. In this case, a first sequence pre-photoconversion was followed by a frame where only 405 nm light was shined to the sample, and the line average was tuned to reach different power densities. In the experiments at fixed 405 nm, the light intensity was set to 4.8 J/cm². The line-by-line was instead used for the spatially localized experiments, where the different channels need to reflect the same time point, minimizing the dynamic rearrangement of the labeled organelles. For this experiment, a ROI was drawn on the image and during one frame the 405 nm light was switched on only inside the ROI. This scheme accounts for the irradiation of 3 J/cm² of 405 nm light. Depending on the intensity of the red form of the green-to-red PCFPs the images were spectrally unmixed using the Spectral Unmixing plug-in of ImageJ. The lifetime was recorded on the same system and the integrated FLIM-Phasors-analysis software was used to visualize the fluorescence lifetime change upon photoconversion for the different fluorescence proteins and their analysis. The lifetime measurements integrated the signal in the same emission intervals stated before. For the measurement done in a bacterial layer, given the homogeneity of the system, a pixel average of 8 has been used. For miRFP720, the lifetime has been further investigated, correlating the evolution of the lifetime with the emission spectral window.

## SDS-PAGE and zinc staining

To test the stability of the photoconversion and the presence of a covalent binding as a BV chromophore stabilizing mechanism, a $Zn^{2+}$ fluorescence assay was performed. Approximately 1.5 µg of protein is diluted 1:1 in buffer (62.5 mM Tris-HCl, pH 6.8, 2% SDS, 25% glycerol, 0.01% bromophenol blue) and denatured at 95 °C for 5 min. After removing debris by low-speed brief centrifugation, the protein was resolved by 4–20% (wt/vol) polyacrylamide gel electrophoresis (PAGE) in the presence of sodium dodecyl sulfate (SDS) (Mini-PROTEAN TGX Precast Protein Gels, BioRad). After SDS-PAGE, the gel was soaked in 100 µM $ZnCl_2$ solution at room temperature for 30 min. The $Zn^{2+}$-enhanced fluorescence of miRFPs was visualized using a fluorescence reader (BioRad XR + ). Ultimately, the gel was stained with Coomassie blue dye.

## Bulk photoconversion

A 5 µl volume of miRFP720 (6.27 µM) purified protein was photoconverted. We placed the drop in a humidified and closed chamber

with a glass coverslip bottom and illuminated it with 775 nm light (Katana, 550 ps, 40 MHz) in the Leica SP8 system. The power of the 775 nm laser was calibrated to reach saturation of the photoconversion on a bacterial layer expressing miRFP illuminated with the same microscope. The microscope was set to scan an area of $63 \times 63 \, \mu m^2$ (divided per $512 \times 512$ pixels) with a dwell time of $7.69 \, \mu s$ (frame rate of 0.093 Hz). Averaging of 4 lines and 10 frames as well as 35 repetitions was performed to finally cover 10% of the volume of the drop. The control drop was placed on the same coverslip and went through the same procedure without being exposed to the photoconversion light.

## Protein expression and purification

For expression in bacteria, the miRFPs were cloned into pBAD/His-B vector (Life Technologies/Invitrogen). TOP10 host *E. coli* cells (Invitrogen) were used for protein expression. A pWA23h plasmid[46] encoding heme oxygenase from *Bradyrhizobium ORS278* (hmuO) under the rhamnose promoter was co-transformed with a pBAD/His-B plasmid encoding a miRFP. Bacterial cells were incubated overnight at 37 °C in an LB medium supplemented with ampicillin and kanamycin. To start protein expression, 0.002% arabinose and 0.02% rhamnose were added. After growing for 12 h at 37 °C, the bacteria were incubated at 18 °C for 24 h. Proteins were purified with Ni-NTA agarose (Qiagen). For elution, PBS containing 100 mM EDTA was used instead of imidazole. The samples were then desalted using PD-10 desalting columns (GE Healthcare).

## Preparation of bacterial samples

The photophysical properties of photoconversion have been characterized using a paste of bacterial cells. About 2 μl of bacterial paste expressing the miRFP protein of interest was placed on a #1.5 coverslip and compressed on a slide, to obtain a homogeneous layer of bacteria, and finally sealed with biphasic glue. To probe the influence of oxygen on photoconversion, this mounting approach (referred to as "closed-chamber" in Supplementary Fig. 20) was compared to a similar amount of bacteria paste placed on an open chamber (MatTek, P35G-1.5-20-C) with a #1.5 coverslip bottom. Plasmids used for bacterial expression of the green-to-orange PCFPs were purchased from Addgene and are the following: pGEX6P-1-Dendra2 (Plasmid # 82436).

## Cell culture

U2OS (ATCC HTB-96) cells and HeLa cells (ATCC CCL-2) were cultured in Dulbecco's modified Eagle medium (DMEM) (Thermo Fisher Scientific, 41966029) supplemented with 10% (v/v) fetal bovine serum (Thermo Fisher Scientific, 10270106) and 1% penicillin-streptomycin (Sigma-Aldrich, P4333), and kept at 37 °C and 5% $CO_2$ in a humidified incubator. For transfection with the miRFP proteins, cells were seeded on coverslips in a 6-well plate or on 8-well μSlide (Ibidi). After 24 h, cells were transfected using FuGENE transfection reagent (Promega, E2312) according to the manufacturer's protocol. At 24–72 h after transfection, cells were washed in phosphate-buffered saline, placed with Leibovitz's L-15 Medium (ThermoFisher Scientific, 21083027) in a chamber and imaged at room temperature. No BV was added to the culture or imaging medium. The plasmids used for the green-to-orange PCFPs were purchased from Addgene and are the following: Dendra2-CD9-10 (Plasmid #57705), mEos2-Peroxisomes-2 (Plasmid #54750), mEos3.2-Clathrin (Plasmid #57452). Plasmids for miRFP fusion were previously presented[28].

## Zebrafish studies

The coding sequence of the histone-H2B-miRFP720 fusion fluorescent protein was cloned in a pENTR/D-TOPO vector (Invitrogen, 450218), which contains Gateway-compatible attL1 and attL2 sites flanking the site of gene insertion, thus generating the pME-H2B-miRFP720 entry plasmid. A zebrafish-specific construct allowing the expression of H2B-miRFP720 was generated through the Gateway multisite recombination system using the above-mentioned pME-H2B-miRFP720 plasmid, the entry vectors p5E-*bactin2* and p3E-polyA from the Tol2kit[47] and the destination vector pDestTol2pACryGFP[48], which contains a cassette that drives expression of GFP fluorescent protein in retinal cells. The Gateway LR Clonase II enzyme mix (Invitrogen, 11791020) was used for the Gateway LR recombination. Clones were selected and verified by restriction enzyme analysis and sequencing, thus obtaining the Tol2pACryGFP_bactin2:H2B-miRFP720 construct. Zebrafish (*Danio rerio*, AB strain) were kept in the Karolinska Institutet Zebrafish core facility according to standard protocols and under the ethical permits Nr 5756/17 and 14049/19, approved by the Stockholm Ethical Committee (Stockholms djurförsöksetiska nämnd) and issued by the Swedish Board of Agriculture. B eggs were collected at 1-cell stage and injected with 1nL of an injection solution containing 50 ng/μL of the Tol2pACryGFP_bactin2:H2B-miRFP720 construct and 35 ng/μL of Tol2 transposase mRNA. At two days post fertilization (dpf), injected embryos expressing GFP in the retina, as a result of construct integration in the genome, were selected for subsequent photoconversion experiments. At 4 dpf, selected larvae were mounted in 1% low gelling point agarose in 12 mm Nunc Glass Base dishes (Thermo Scientific, 150680) and used for photoconversion of H2B-miRPF720$^+$ cells in the aorta-gonad-mesonephros (AGM) region.

## Cell-viability assessment

The overnight culture of *E. coli* bacteria expressing miRFP720 has been diluted 1:20, of which 3 μl has been immobilized between a coverglass and an agar pad with the corresponding antibiotic resistance. The measurements have been conducted at 37 °C with a brightfield recording every 2 min. The experiments with NIR light have been conducted on the custom-made STED microscope described previously, equipped with a CMOS camera DMK 33UP1300 (The Imaging Source Europe GmbH, Bremen, Germany) for brightfield imaging. The area of interest has been first exposed with the 775 nm illumination (100x objective lens, scanning modality, pixel size 150 nm, dwell time 50 μs, power at the BA ~ 7.4 mW), then visualized in fluorescence to assess the photoconversion over a bigger area and finally followed in bright field imaging with an exposure time of 250 ms. The experiments with 405 nm light have been conducted on a Zeiss confocal microscope (63x objective). The 405 nm power has been calibrated with *E. coli* expressing Dendra2 to define the imaging setting for photoconversion saturation (pixel size 150 nm, dwell time ~ 10 μs, power at the sample plane ~ 55 μW). The assessment has then been conducted on *E. coli* expressing miRFP720 to be consistent with the previous measurement. For the analysis, the bacteria have been segmented by training the deep learning network StarDist[49] through the ZeroCostDL4Mic[50] notebook and the area of the bacteria colonies were linked and followed for the full timelapse. This strategy, where we follow a bigger area of what is subjected to photoconversion, gives the possibility to directly compare the two conditions in the same measurement and thus minimizes possible biases due to sample-to-sample or day-to-day variability as well as possible differences in the segmentation due to using different microscopes. For the cell viability assessment, HeLa cells expressing H2B-miRFP720 have been followed over an extended period (7–11 h) after photoconversion with NIR light (10x objective lens, scanning modality, pixel size 613 nm, dwell time 12 ms, power at the BA of 16 mW). The temperature 37 °C, 5 % $CO_2$ and 90% humidity have been set on an Okolab incubator chamber (Naples, Italy). A bright field image was recorded every 5 min with an exposure time of 100 ms. The fate of all the cells enclosed in the first frame have been classified into: dead, dividing and live. All the cells have been considered for analysis, both transfected and not transfected.

## Statistics and reproducibility

Data were analyzed using Origin Pro 2018 (64 bit) and custom pipelines in Fiji macros. The NIR-to-far-red photoconversion contrast is

mainly power dependent, therefore, the energy used in the live-cell experiments is guided by the calibration curves at different wavelengths and, therefore, different microscopes, such as the one reported in Figs. 1d, 3b, c, Supplementary Figs. 14 and 15 for the proteins expressed in bacteria or Fig. 4e, Supplementary Figs. 22 and 23 for the protein expressed in mammalian cells. Sample sizes, means, and standard deviations are indicated in the relevant figure legends. All imaging results are presented as representative examples of an experiment repeated at least twice or as indicated in the respective figure legend. Detailed information about the experiments behind each figure is reported in Supplementary Table 1.

### Reporting summary

Further information on research design is available in the Nature Portfolio Reporting Summary linked to this article.

## Data availability

The data generated in this study have been deposited in Zenodo under accession code 5884553. Source data are provided with this paper.

## Code availability

The scripts and pipeline of analysis used in the manuscript are available in the Zenodo repository https://doi.org/10.5281/zenodo.5884553.

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

## Acknowledgements

We thank the Imaging Facility at Stockholm University (Christiane Peuckert) for the access to the two-photon Leica SP8 microscope and the Advanced Light Microscopy Facility at SciLifeLab (Hans Blom) for the access to the STED Leica SP8 microscope (Campus Solna RED grant project). We thank Andrea Volpato for the discussion about the photoconversion mechanism. This work was supported by the ERC starting grant MoNaLISA 638314, Swedish Research Council VR starting grants 2016-03572 and 2021-04528, the Swedish Foundation for Strategic Research project FFL15-0031, the GM122567 grant from the US National Institutes of Health, and the grants from the Finland Cancer Foundation, and the Jane and Aatos Erkko Foundation.

## Author contributions

I.T. and V.V.V. designed and supervised the entire project. F.P. performed the photophysical characterization and imaging experiments. J.A. built and operated the STED microscope. F.P. and M.D. prepared the eukaryotic samples. F.P. and D.O. performed the zinc staining. O.S.O. and D.M.S. designed the expression plasmids, prepared bacteria and purified proteins. D.M.S. and V.V.V. studied the photoconversion mechanism. R.A.M. and E.J.V. performed the zebrafish experiments. F.P., I.T. and V.V.V. wrote the manuscript with input from all authors.

## Funding

## Competing interests

The authors declare no competing interests.
