## [Peer Review File · Nature Communications]

REVIEWER COMMENTS

Reviewer #1 (Remarks to the Author):

Overall:

The manuscript has been much improved and streamlined since the last time I reviewed it. The authors have invested much thought into considering cross-talk/bleed-through effects for the conversion analysis which is commendable.

Importantly though, the maximum conversion yield reported is rather low. In comparison to green-to-red photoconversion that can achieve up to 80-fold changes using dual-laser illumination via primed conversion (see Dempsey et al. 2015, Nat Meth) or to a lesser degree double digit fold changes using 405nm illumination due to various photochemical effects (see Thédié et al, J. Phys. Chem. Lett. and De Zitter et al., 2019, Nat Meth,). Hence, green-to-red photoconversion using primed conversion is still 10-fold higher than the values reported for the most optimal procedure of miRFP photoconversion selected by the authors. To be sincere and transparent to the interested reader, the reviewer suggests emphasising the conversion yield limitation in the discussion. The authors are also encouraged to mention that it is likely that further protein engineering efforts can e.g. convey higher photostability to the miRFP photoconversion versions as previously accomplished for green-to-red photoconvertible fluorescent proteins (see Mohr et al. 2017, Angew Chem, Turkowyd et al. 2017, Angew Chem) which will likely increase the conversion rate.

I congratulate the authors for the FLIM data inclusion and their prominent placement into the main text. This important addition renders the comparison more robust given that photoconversion yields are currently rather low. The interested user will be very eager to use this alternative approach, as it can be conducted in a commercially available system as the authors demonstrated. I am though interested to know why the lifetime is several folds shorter than for other FPs?

Here are my remaining comments to the revised manuscript:

Introduction:

‘These are characterized by high brightness and photostability in both states.’

Technically not correct - the pre-versions are often inferior to most non-modifiable FPs. Recent efforts have improved the photostability of the green versions by several folds (see Mohr et al., 2017, Ang Chem Int Ed). This deserves a reference.

'To bypass the use of highly energetic UV light in the green-to-red PCFP variants, two-photon photoconversion^{13,14} or simultaneous blue and near-infrared (NIR) light conversion¹⁵ have been used with different levels of efficiency.'

Primed conversion can be performed simultaneously as well as with a temporal delay using red (see Mohr et al., 2017, Ang Chem In Ed) or near-infrared (NIR) (Dempsey et al. 2015, Nat Meth). Hence, please correct the wording and refer also to Mohr et al., 2017, Ang Chem In Ed as follow:

'To bypass the use of highly energetic UV light in the green-to-red PCFP variants, two-photon photoconversion or primed conversion (i.e. dual laser illumination of blue and red/near-infrared (NIR) light) have been used with different levels of efficiency.'

Further, in its current form, the authors' chosen modification does not inform the interested reader that primed conversion can achieve a very high conversion efficiency (up to 80-fold, see Dempsey et al. 2015, Nat Meth and Mohr et al., 2017, Ang Chem In Ed) whereas two-photon photoconversion displays a very minor conversion efficiency (less than 2-fold, see Dempsey et al. 2015, Nat Meth), which renders this methodology not a practical option. I do not insist on defining this difference, yet I encourage the authors to at least mention in their discussion that in their particular case conversion efficiencies could be much higher comparable to values achieved by primed conversion, echoing my previous comments.

Figures:

Figure 1b - The colour yellow is barely visible. Please opt for another contrast fitting for PSmOrange.

Figure 1e - The maximum green-to-red photoconversion using primed conversion is still 10-fold higher than the values reported for the most optimal procedure of miRFP photoconversion selected by the authors.

To be sincere and transparent to the interested reader, the reviewer suggests emphasising the conversion yield limitation in the discussion as mentioned above.

Figure 2b – Please include labels for immediate identification.

Figure 2c (right) - Why does the red channel increase – should the value not decrease, as the pool of red FP is reduced?

Figure 4 - To provide a convincing case for in vivo axial confinement authors are advised to use cytoplasmic label and photoconvert one single cell in vivo rather than using a nuclear label that is mosaically expressed. Further, to rule out any photodamage the authors are advised to provide datasets where the photoconverted cells successfully undergo division. No need to do lineage tracing.

S2 - The colour yellow is barely visible. Please opt for another contrast fitting for PSmOrange.

S8 – figure legends: Please replace ‘matains’ with ‘maintains’.

Reviewer #2 (Remarks to the Author):

The manuscript reports on the photo-convertibility of fluorescent proteins (FP) to shorter wavelengths upon irradiation with near-infrared light. The studied FPs are called miRFP (as in monomeric, infrared red-fluorescent protein) and were originally developed a few years ago by one of the PIs (Shcherbakov et al, Nat Comm. 2016) and subsequently shown to be useful for live-cell super-resolution microscopy (Matlashov et al, Nat Comm 2020) by the two PIs of the present study.

Extending the color palette of FPs into the infrared spectral range is an important endeavor for live-cell imaging in tissue, not only to increase the number fluorescence signals that can be spectrally separated but also to exploit the fact that biological tissue can be imaged better at longer wavelengths where autofluorescence, absorption and scattering tend to be reduced.

What's new in this study is the observation that these miRFP proteins undergo a blue-shift in their spectral properties upon exposure to infrared light. This is a curious and counter-intuitive effect, which is potentially interesting, as most photo-conversions involve a red-shift to longer wavelengths.

In general, light-controlable spectral shifts can be a very useful feature for pulse-chase type experiments, but they may also be a bug that complicates quantitative measurements, if the fluorescence intensities or lifetimes cannot be fully accounted for.

It's a bit unclear if this may be the case here or not, because for instance the wavelength used for STED imaging by itself triggers the photo-conversion, convolving the imaging with the photo-conversion.

I have these two general criticisms:

- The benefit of the blue-shifted photo-convertibility remains a bit theoretical. If the authors could demonstrate more clearly its advantages over existing FP imaging strategies, it would increase the impact of the study.
- The paper is not so well written or organized, jumping straight into technical details in the Introduction and Discussion sections, whereas a broader view of the context, motivation and significance of the study would help the reader judge its merits.

Reviewer #3 (Remarks to the Author):

Report to 393188_Pennacchietti et al.

This study presents properties of five derivatives of bacterial phytochromes tailored to minimal size, but still maintaining their capability of photoconversion and fluorescence emission. Fused proteins were generated with other proteins characteristic for various organelles and cellular compartments. The major intention on the molecular level was a variation in absorption and emission properties highlighting here the switch from the parental state to a blue-shifted photoproduct, in contrast to most GFP-derivatives that are mentioned for comparison.

Taking together the work in living cells, the application of various microscopic methods, and the data handling, all this is performed and presented with the well-known great expertise of this ensemble of authors. No critics here.

The treatment of the basics in phy photochemistry (abs and fluorescence), however, shows flaws and logical inconsistencies and needs thorough revision (see details further down). As a general comment, many sections of the paper can only be understood after detailed study of the SI material. To the understanding of this reviewer, SI, however, should only give details to the specialists and should be part of the main body of information.

In general, the manuscript demonstrates that in many cases, also here, scholars live in a bubble of their own community. Many arguments, results, and comments cannot be understood without reading the cited literature. Acronyms and abbreviations are not explained. Nowhere in the manuscript is the origin of these miRFPs (species name?) mentioned. Also, changes by mutagenesis (amino acid positions and exchanges from/to) that yielded the generated five RFPs are not presented. A cartoon of the 3D crystal structure would be more than helpful. It is assumed that the naming of the compound, example ,miRFP670' identifies the absorption maximum. Detailed information can only be extracted from the SI-table.

Comments to selected paragraphs

I. 60/61

The presentation of bacterial phytochromes focussing on the here employed proteins should contain an additional information, as such that some of these bacterial proteins are so-called ‚bathy‘-phys (this difference is to be mentioned and cited) highlighting the advantage that the parental state is the one absorbing at longer wavelengths. So, this phenomenon is well known in the community and not a novelty.

I. 63

Here, an improvement of the fluorescence is mentioned as 20-fold. This value sounds impressive, however, a principal drawback of many – including those used here – FPs of various origin is their extremely low genuine fluorescence efficiency. This holds true also in this manuscript, as an absolute parameter of the fluorescence (always shown as normalized to 1.0) is never given. Use of a well characterized reference compound would have been most welcomed. In this context, several of the presented emission spectra are suffering from a low S/N ratio pointing to either a low expression of these proteins or a very low fluorescence efficiency. The authors claim a ‚high sensitivity to day light‘, yet, the quantum yield for fluorescence – as said above – is marginal. This reads like a contradiction.

I. 74

The rationale of this paragraph at the end of the introduction does not easily disclose to the readers as it leaves them somehow confused. In the beginning the authors emphasize the advantages of the miRFPs as being extremely red shifted advantaging over autofluorescence and other obstacles and allowing more easy multiplexing by extending the wavelength range for applications. So, why a blue shift?

I. 85

The authors might add in brackets the maxima for absorption and emission (although the parameters are given in the SI material). This would make further discussion more easy, see I. 87 (ca. 53 nm blue shift: from where to where is this blue shift?)

I. 93/94

Determining the yield of conversion: this reviewer is not convinced that the ratio of fluorescence is a good measure for this photochemical process, except the authors have determined independently the fluorescence quantum efficiencies of both states (that surely are different!!). If this has been taken into account, it should be mentioned.

I. 96-99

This seems to be trivial, as the conversion efficiency (if the absorption band is homogeneous) follows the absorption band intensity = oscillator strength = probability to catch a photon.

I. 117

This paragraph outlining the spectral properties of the Soret is surely correct, however, own experiences in phy spectroscopy have told that this band is of low use for activation phy proteins for (i) a lower efficiency (correctly cited), and (ii) the shift of the Soret band upon conversion is so small that selective excitation seems to be impossible. Further, the absorption is placed in the region of autofluorescence caused by flavins, if the experiment goes to living cells. The authors might consider to minimize this paragraph in length.

This aspect is picked up again in l. 134. So, it might be worth to present the spectra (abs, fluo) for the Soret band.

The use of the Soret band for excitation only discloses later when the same wavelength is used for two different photolabels with same excitation wavelength range but differently tagged proteins.

l. 207

The advantage of not damaging DNA by 700 nm light is apparent and trivial. DNA absorbs at 260 nm (maximum), and even excitation of the Soret band around 400 nm would not be harmful (if wavelength selection would be performed properly).

l. 210

This paragraph required several-fold reading to understand the message. Also, as at first glance some of the arguments appear to contradict generally accepted phy results. There are several aspects to address: (i) why does the excitation not induce isomerization of the D-ring? If mentioned in the text above, the reason for this statement is not clearly outlined. (ii) Where is the result that neighboring residues move in response to the ring movements? There is literature to this conformational change from other phys, but here, is there a crystal structure or mutagenesis experimental evidence (for second bond formation)? In addition, how is the 'transient bond formation' concluded?

(iii) Why shows the denatured protein a higher mobility in SDS-PAGE? Proteins under these conditions are denatured and adopt a spheric, oblong-oval conformation that not necessarily is disturbed by an additional bond (l. 219).

(iv) 'Chromophore protonation': Detailed studies have shown that phys carry the chromophore in the protonated state in both states parental and photoproduct. This feature has been intensely discussed in the phy literature, and also deprotonation/reprotonation is known, e.g., meta-R II intermediate in plant phys that still carry the chromophore protonated in both states. Also deprotonation in few CBCRs have been reported, still with intense discussion in the community. Thus, the comment that the chromophore is protonated seems to be in agreement with common knowledge and should not be taken as a special exception. Also, the authors should be aware that bilin protonation in general causes a BATHOCHROMIC shift. If the authors wish to keep the protonation argument they should make this more clear by a detailed discussion.

l. 373/4

The authors record here fluorescence. Thus 'fluorescence' should be added: ... and fluorescence was recorded in ...

l. 375, change tense (characterize): ... to test and to characterize ...

l. 384, add comma: .. dependency, and lifetime

l. 386-388: this sounds confusing, as 'line average was tuned to reach different ...', EXCEPT (highlighted by reviewer) for ... 405 nm light, the others were set to a 405 nm light intensity of' So, what (which wavelength) was kept and which was tuned?

l. 391: ',... accounts for ... of 3..' 3 what?

I. 403 and following

This reads somehow confusing. The protein is denatured and this reviewer assumes it is no longer soluble, but will be found in the precipitate. If then the ,debris' is removed, also denatured protein might be lost. The authors should clarify this point.

Supplementary material

Table S1

It is assumed that ,excitation' maxima are absorption maxima, and should better be named as such. Notably, the $\Delta\lambda$ values in absorption and emission (Stokes shifts) become smaller the longer the absorption/emission wavelength are. This diminishes to some extent the proposed advantage of the novel compounds. Further, as this manuscript presents far red-shifted compounds the information on changes in the Soret band is not complete in the table, but is not helpful in a discussion of potential applications, the more so, as the nm-shift of the Soret bands in absorption and emission is marginally small and do not practically allow selective excitation and detection.

Comment to note and fig N1. The authors show examples highlighting the principle for cases of large $\Delta\lambda$ A/E shifts. Correct, however, they use as example (panel a) an absorption of 500 +/- nm that is not given for any of their compounds. Further, both cases red- or blue shifts of the photoproduct does not escape the dilemma: panel a (for a large shift of 100 nm!) still has contributions of exciting also the photoproduct, and even if this can be minimized, there might appear a FRET process falsifying the result. This is even worse in panel b, where excitation of the parental state generates a significant amount of photoproduct, again causing false positive results. The authors address this point in their note, but clearly, precise manipulation of the data is required and may bear difficulties.

Fig N2. Why do the authors excite at 405 nm? Apparently, this is the Soret band, but why not escaping autofluorescence of the cells and using a longer wavelength?

Reviewer #4 (Remarks to the Author):

The paper is interesting and innovative, but has also several weaknesses as will be pointed out below. The major concerns with the manuscript are that

- 1) The main aim of the paper is unclear.
- 2) Statistics and information about reproducibility are lacking
- 3) Many of the conclusions drawn are discussed at a superficial level

Key results

The manuscript contains novel concepts. Particularly the approach of employing photoconversion to enable intracellular diffusion studies in combination with STED; however, the key finding is not clearly defined. Is it the discovery of the blue shift photoconversion of the biliverdin chromophore using NIR excitation that is the primary results, or is it the technical realization of a sophisticated approach combining photoconversion in a more general sense? Based on the title the blue-shifted photoconversion using NIR is highlighted as the discovery, but as the manuscript also presents data in which 405 nm excitation is utilized for photoconversion, as well as another fluorophore which is green-to-red shifted (rather than NIR-to-far-red shifted), this is confusing. Furthermore, a mechanistic understanding of the “discovery” of blue shift in the specific fluorophore is not adequately substantiated, which is a weakness.

Validity

Although the data presented in each figure is analyzed and interpreted thoroughly, the overall robustness and validity of the approach is difficult to assess since statistics and reproducibility is not commented. This is a general weakness throughout the manuscript.

Significance

As pointed out above, the approach employing photoconversion to enable intracellular diffusion studies in combination with STED is of interest for the biophotonics community, as well as the possibility to combine the approach of photoconversion at multiple wavelengths enabling multiplexing; however this is not pointed out and supported as the main finding. Instead, several different sub-goals are highlighted as the main conclusions. Many of these are demonstrated as proof-of-concept, and discussed briefly as will be more specifically criticized below.

Thus it is advised that the manuscript is revised to limit the scope and thereby be more clear about the main findings and/or discovery. The revision should preferably include statistical analyses and elaborate on the theoretical anchoring to support the conclusions.

Critical review of the conclusions drawn

Report on a NIR-to-far-red photoconversion phenomenon.

The manuscript contains data acquired from E.coli bacteria expressing mRFP709 and analogue mutants supporting this claim; however, the underlying mechanism being the phenomenon is not discussed at a satisfactory detail.

Neither of the mutants demonstrate Q-band emission (and therefore also unlikely absorption) of NIR at 775 nm, which is the excitation wavelength chosen for the photoconversion. It might be speculated that at this wavelength using 120 fs pulsed laser light at quite high powers (4 MW/cm²) non-linear excitation routes (targeting the Soret-band?) might come into play; however, nothing is stated about this. Instead, it is vaguely stated that “Likely, the blue-shift of the photoconverted form is caused by disturbance of the π -conjugated system in the BV chromophore”. If, for example, two-photon excitation is expected to perturb the π -conjugated system, this should be further supported and related to the Soret-band absorption. Furthermore, it would have been appropriate to include two-photon excitation and cross sections data to support the mechanistic claims. In fact, the term two-photon excitation is just briefly mentioned in the discussion section and not explained in the context of data, which is a weakness.

In the discussion section (line 210-228), there is a paragraph more focused on the mechanistic understanding; however, it is difficult to connect this paragraph to the actual data presented. The paragraph brings up quite specific information about the protein structure, without connection to actual data. Bringing back to what is pointed out above, the key question still concerns how the excitation using NIR can induce this photoconversion effect. Thus it is advised that the mechanistic discussion is merged and integrated with the data commentary section.

Another confusion is that both NIR induced as well as 405 nm induced photoconversion are explored in the paper, but their presentations are not clearly separated and explained. For example, the photoconversion taking place in HeLa cells (Note in materials and methods cells are stated to be U2OS?) seem to be restricted to 405 nm induction. Also the reported lifetime shift from the photoconversion seem to rely on 405 nm irradiation. This mixing up and lack of systematic analysis is confusing to the reader and needs to be clarified.

The observed photoconversion is further claimed based on the observation of a shift in fluorescence lifetime. Data supporting the observation of increase of fluorescence lifetime is presented for photoconversion using 405 nm excitation (and not supported for the NIR induced photoconversion?). It is unclear for what reason the lifetime measurements are included. Is it to support the understanding of the underlying mechanism, or more as technical means to improve imaging contrast? Stated in the manuscript is a sentence “The photoconversion introduces an unbalance in the protein population by shifting it towards the blue shifted form with a longer lifetime”. What is the rationale for drawing this conclusion, and what is meant by “unbalance in protein population”? Instead it would be interesting to elaborate the discussion about the origin of the observed shift in lifetime from a more fundamental level. Does the longer lifetime relate to a shift of the natural lifetime of the fluorophore due to the hypsochromic shift, or can it be speculated that the non-radiative decay routes of the chromophore-protein complex is affected?

The “prolonged stability” is promoted in the discussion as an advancement; however, the present data is not discussed in the context of state-of-the-art in the field. What is the expected stability of the photoconversion technologies known today? This needs to be stated in order to support the statement.

High spatiotemporal precision

It is demonstrated that the concept of photoconversion of the miRFP is compatible with STED microscopy, known to have superior spatial precision compared to e.g. confocal microscopy. However, it is unclear what is referred to in the context of temporal precision? This should be clarified, or the claim removed.

Study dynamic processes in living cells

Photoconversion is reported to be demonstrated in mammalian cells. The manuscript includes 3-5 images from single HeLa(or U2OS?) cells, using fusion to different cellular proteins; however, statistics are missing. In order to claim successful photoconversion in mammalian cells, it needs to be stated what cell types the claim is restricted to and something about the success rate of the method. Was it a one time experiment, or could it be repeated?

Data from a single multiplexing photolabeling experiment is demonstrated as proof-of-concept in living cells (Figure 3, cell type not stated in legend). It is claimed that “using this method, subsets of lysosomes and peroxisomes targeted with miRFP720 and mEos2 respectively were photolabeled and tracked over time”. Statistics are lacking. Furthermore, the biological implications of the findings in the experiments are not discussed. Is the finding realistic/unexpected? Is the method limited to this cell type, this organelle? What alternative methods is this approach complementary to?

In Figure 4 g-h, data are presented from zebra fish embryo. It is stated that “single cells could be photolabeled without any sign of photodamage, opening new possibilities for tracking cell dynamics in vivo.” How was the viability and lack of photodamage assessed/validated in order to make this claim.

Background of Photoswitchable fluorescent proteins

The general concept of photoswitchable fluorescent proteins is introduced in context of two specific references (Ref 1 and 2), restricted to co-authors of the present manuscript. It can be questioned if this is a true representation of the state-of-the art in the field. Photoswitching and photoconversion is of broad general interest e.g. in the fields of photopharmacology and optogenetics, and should preferably be introduced at a more general level.

Additional comments

In the abstract, several claims are made at a more general level. Therefore, the abstract is not conveying in what specific way the work is advancing the knowledge in the field. The abstract should be rewritten to better reflect the scientific work, the findings made, and the advances of understanding.

Figure 2. Are data from HeLa cells or U2OS as stated in materials and methods?

Figure 4 a-f show data from a (single?) cell, while fig 4g-h demonstrate data from zebra fish larvae. It is confusing to combine these data in the same figure.

The manuscript lacks a concluding paragraph that summarizes the main findings and how the paper actually advances the knowledge in the field.

It would be advised to present an overview of what different types of organisms are investigated in the manuscript and for what purpose: E.coli, HeLa cells (or U2OS?), and zebra fish larvae. Particularly useful would have been to motivate the choice of model organisms in the introduction, in connection to biological relevance and stat of the art in the field.

Point-by-point responses to reviewers of the manuscript NCOMMS-22-41203:

Blue-shift photoconversion of near-infrared fluorescent proteins for labeling and tracking in living cells and organisms

Francesca Pennacchietti, Jonatan Alvelid, Rodrigo A. Morales, Martina Damenti, Dirk Ollech, Olena S. Oliinyk, Daria M. Shcherbakova, Eduardo J. Villablanca, Vladislav V. Verkhusha, and Ilaria Testa

We appreciate the comments and the suggestions of all four Reviewers. All the inputs helped to improve both the structure and clarity of the manuscript.

In the revised version we performed new experiments to provide additional insight into the photophysical mechanism behind miRFP photoconversion, a new multiplexed photoconversion application, a direct comparison to primed conversion, and further assessment of phototoxicity. With the current work we want to report, for the first time, the unique phenomenon of the NIR-induced fluorescence hypsochromic shift in bacteriophytochrome-derived NIR FPs, as well as to demonstrate the impact of this novel photoconversion mechanism for bioimaging, where it enables several novel applications in live cells and living animals. The text now clearly outlines this perspective. We have expanded the Introduction section to provide a deeper context both of existing photoconversion mechanisms and the origin of the miRFP family. The Results section has been restructured to first provide the photophysical characterization, and then present the newly enabled imaging strategies, namely photoconversion multiplexing, combination with STED microscopy, and *in vivo* imaging.

Below we provide a point-by-point response to the comments of each Reviewers. The comments of the Reviewers are written in blue and our responses in black.

Reviewer comments

Reviewer #1

Overall

The manuscript has been much improved and streamlined since the last time I reviewed it. The authors have invested much thought into considering cross-talk/bleed-through effects for the conversion analysis which is commendable.

We thank the reviewer for his/her positive evaluation of our work.

Importantly though, the maximum conversion yield reported is rather low. In comparison to green-to-red photoconversion that can achieve up to 80-fold changes using dual-laser illumination via primed conversion (see Dempsey et al. 2015, Nat Meth) or to a lesser degree double digit fold changes using 405nm illumination due to various photochemical effects (see Thédié et al, J. Phys. Chem. Lett. and De Zitter et al., 2019, Nat Meth,). Hence, green-to-red photoconversion using primed conversion is still 10-fold higher than the values reported for the most optimal procedure of miRFP photoconversion selected by the authors. To be sincere and transparent to the interested reader, the reviewer suggests emphasising the conversion yield limitation in the discussion. The authors are also encouraged to mention that it is likely that further protein engineering efforts can e.g. convey higher photostability to the miRFP photoconversion versions as previously accomplished for green-to-red photoconvertible fluorescent proteins (see Mohr et al. 2017, Angew Chem, Turkowyd et al. 2017, Angew Chem) which will likely increase the conversion rate.

Following the reviewer's suggestion, we have expanded the discussion and provided new experiments to clarify the applicability of the miRFP photoconversion mechanism compared to previous PCFPs.

The maximum conversion yield is indeed an important parameter, but the exact value can differ depending on the used metric, and from sample to sample and instrument to instrument. In our study, as in the studies mentioned by the reviewer wherever reported in the supplementary materials (i.e., Dempsey et al. 2015, Nat. Methods, doi: 10.1038/nmeth.3405; Mohr et al. 2017, Angew. Chem., doi: 10.1002/anie.201706121; Turkowyd et al. 2017, Angew. Chem., doi: 10.1002/anie.201702870), the fold of conversion is calculated as the ratio between the fluorescence intensity after and before photoconversion in the detection channel of the photoconverted form, i.e., $\text{Fluo}^{\text{after-PC}} / \text{Fluo}^{\text{before-PC}}$. Using this metric in a direct comparison, the presence of any level of spectral crosstalk between the two forms and the presence of any blue-shifted species at equilibrium define the $\text{Fluo}^{\text{before-PC}} > 0$, as is the case for the miRFPs. Instead, for most green-to-red PCFPs, the excitation of the photoconverted red-shifted species does not cross the absorption spectra of the green-shifted species and at equilibrium all the population is in the non-photoconverted red-shifted form. Therefore, the fluorescence emission of the photoconverted red-shifted species before photoconversion is very close to zero, $\text{Fluo}^{\text{before-PC}} \sim 0$. With a value close to zero as the denominator in the ratio, the metric becomes extremely dependent on the background intensities of the detectors in use and on the acquisition settings (excitation power, dwell time, detection bandwidth etc.) for the photoconverted channel, $\text{Fluo}^{\text{after-PC}}$. We have discussed these considerations in Supplementary Note 1.

We also discuss how future miRFP engineering efforts can be directed to the optimization of the photoconversion mechanism in the Discussion section:

The photoconversion yield of miRFPs is currently lower than the one reported for GFP-like PCFPs. Nevertheless, future tailored engineering efforts have the potential to further optimize the miRFPs toward higher photoconversion yield, firstly by minimizing crosstalk and heterogeneity in the equilibrium state.

Furthermore, we envision that the use of the phasor approach that helped us identify the presence of some heterogeneity in the equilibrium state, where an amount of blue-shifted species is already present, would be helpful in guiding the mutagenesis. These efforts would be focused on both minimizing the equilibrium-presence of the blue-shifted species, as well as increasing the spectral shift between the converted and non-converted form.

To conclude, even if the calculated photoconversion yields of the current miRFP variants are an order of magnitude lower than green-to-red PCFPs, we believe that this NIR-to-far-red photoconversion mechanism can be a valuable tool due to the NIR spectral region, axial confinement, and lower illumination doses for conversion compared to primed conversion.

I congratulate the authors for the FLIM data inclusion and their prominent placement into the main text. This important addition renders the comparison more robust given that photoconversion yields are currently rather low. The interested user will be very eager to use this alternative approach, as it can be conducted in a commercially available system as the authors demonstrated. I am though interested to know why the lifetime is several folds shorter than for other FPs?

We thank the reviewer for acknowledging the FLIM data and motivating us with her/his previous comments to further investigate the lifetime dimension.

Fluorescence lifetime for most FPs of the GFP family is in the range of 2.3–3.5 ns, with few exceptions in the extended interval of 0.7–5 ns (Mamontova et al., *Sci. Rep.*, 2018. doi: 10.1038/s41598-018-31687-w). On the contrary, for bacteriophytochromes fluorescence lifetimes in the sub-nanosecond domain are common, and miRFPs are in line with this general behavior. The short lifetime is generally linked to a greater instability of the chromophore in the pocket and the nature of the hydrogen bond network that results in multiple non-radiative decays to the ground state (Hontani et al., *Sci. Rep.*, 2016 doi: 10.1038/srep37362).

More specifically, the lifetimes in the miRFP family show a correlation with the emission spectra, with longer lifetimes, ~ 1.1 ns, for blue-shifted miRFPs and shorter lifetimes, ~ 0.7 ns, for red-shifted miRFPs (Rice et al., *Cancer Res*, 2015, doi: 10.1158/0008-5472.CAN-14-3001). This dependency is also shown in

the reported measurement, Fig. 2a of the main text. The longer lifetime is associated with an increased rigidity of ring A for blue-shifted miRFPs (i.e., miRFP670), where two cysteines covalently bond the chromophore instead of one. This reduces the mobility, affecting the π -electron conjugated system, and minimizes the dispersion (Hontani et al., *Sci. Rep.*, 2016, doi: 0.1038/srep37362). Another factor that has been demonstrated to influence the fluorescence lifetime of this type of near-infrared FP is the hydrogen bond network of the chromophore, specifically demonstrated in iRFP713 (Zhu et al., *Sci. Rep.*, 2015 doi: 10.1038/srep12840). Indeed, the hydrogen bond network is a regulator of the overall flexibility of the chromophore.

Relying on those previous studies, the observed increase in lifetime upon photoconversion can be linked to a stabilization of the chromophore, either by the formation of an additional covalent bond to the biliverdin, as we observe in the Zn^{2+} assay (Supplementary Fig. 22), and/or by a rearrangement of the hydrogen bond network.

Here are my remaining comments to the revised manuscript

Introduction

‘These are characterized by high brightness and photostability in both states.’

Technically not correct - the pre-versions are often inferior to most non-modifiable FPs. Recent efforts have improved the photostability of the green versions by several folds (see Mohr et al., 2017, *Ang Chem Int Ed*). This deserves a reference.

We updated the sentence in the paper to enclose the evolutionary effort that has been done in the green-to-red PCFPs. The updated version is:

These have been engineered to achieve high brightness, contrast of photoconversion and photostability in both states^{10,11}.

Where the citations are [10] Turkowyd et al *Angew Chem* 2017 doi: 10.1002/anie.201702870, and [11] Mohr et al *Angew Chem* 2017 doi: 10.1002/ange.201706121, both key in the recent development of the green-to-red PCFPs.

‘To bypass the use of highly energetic UV light in the green-to-red PCFP variants, two-photon photoconversion^{13,14} or simultaneous blue and near-infrared (NIR) light conversion¹⁵ have been used with different levels of efficiency.’

Primed conversion can be performed simultaneously as well as with a temporal delay using red (see Mohr et al., 2017, *Ang Chem In Ed*) or near-infrared (NIR) (Dempsey et al. 2015, *Nat Meth*). Hence, please correct the wording and refer also to Mohr et al., 2017, *Ang Chem In Ed* as follow:

‘To bypass the use of highly energetic UV light in the green-to-red PCFP variants, two-photon photoconversion or primed conversion (i.e. dual laser illumination of blue and red/near-infrared (NIR) light) have been used with different levels of efficiency.’

Further, in its current form, the authors’ chosen modification does not inform the interested reader that primed conversion can achieve a very high conversion efficiency (up to 80-fold, see Dempsey et al. 2015, *Nat Meth* and Mohr et al., 2017, *Ang Chem In Ed*) whereas two-photon photoconversion displays a very minor conversion efficiency (less than 2-fold, see Dempsey et al. 2015, *Nat Meth*), which renders this methodology not a practical option. I do not insist on defining this difference, yet I encourage the authors to at least mention in their discussion that in their particular case conversion efficiencies could be much higher comparable to values achieved by primed conversion, echoing my previous comments.

Following the suggestion of the reviewer, we have modified the text to better clarify the differences in the photoconversion of green-to-red FP with the different strategies. The text is now:

To bypass the use of highly energetic UV light, two-photon photoconversion^{16,17} or primed conversion¹⁸ i.e. the combined illumination of blue and red/near-infrared (NIR) light have been used. However, the former approach suffers from low yield, while the latter still requires visible spectra illumination, relatively high red-light intensities and slow kinetics¹⁸.

We cite [18] Dempsey et al 2015 *Nat. Meth.*, as a good resource for the reader to both be introduced to primed conversion and compare this strategy to alternative photoconversion wavelengths.

Figures

Figure 1b - The colour yellow is barely visible. Please opt for another contrast fitting for PSmOrange.

We updated the figure accordingly, using a darker color for PSmOrange.

Figure 1e - The maximum green-to-red photoconversion using primed conversion is still 10-fold higher than the values reported for the most optimal procedure of miRFP photoconversion selected by the authors.

To be sincere and transparent to the interested reader, the reviewer suggests emphasising the conversion yield limitation in the discussion as mentioned above.

As explained more in detail in the first response we have followed this suggestion and expanded the Discussion accordingly to be transparent to the reader.

Figure 2b – Please include labels for immediate identification.

We updated the figure accordingly.

Figure 2c (right) - Why does the red channel increase – should the value not decrease, as the pool of red FP is reduced?

There is a degree of spectral bleed-through present between the two species of the investigated miRFP, meaning that in the red channel we will have a part of the signal coming from the blue-shifted species. While this bleed-through is small, in Fig. 2c (Fig. 2g in the revised manuscript) the overall intensity of the bleed-through is not taken into account, only the relative increase in signal in the different lifetime components. Since the long-lifetime component mainly represents the blue-shifted species (as can be seen from Figure 2d), we can understand that the 4-fold increase in signal in the long-lifetime component in the red channel is caused by the small bleed-through present. Instead, the overall strong decrease of the signal in the red channel is represented by decrease of the short-lifetime component.

Figure 4 - To provide a convincing case for in vivo axial confinement authors are advised to use cytoplasmic label and photoconvert one single cell in vivo rather than using a nuclear label that is mosaically expressed.

The suggestion of the reviewer about the axial confinement of photoconversion is not only important for the live-cell applicability of this photoconversion, but it also gives us further insight on the photophysical mechanism behind the photoconversion at different wavelengths.

To get an answer not convoluted with any biological system used, and to compare the axial confinement at different wavelengths of photoconversion (such as 405 nm and 775 nm), we simplified the system and considered a thick pad of bacteria expressing miRFP713 (Fig. R1). We chose miRFP713 for its higher fold of conversion among the studied miRFP, but we strongly believe that the observation can be extended to all other miRFP studied. We photoconverted an area of $20 \times 20 \mu\text{m}^2$ at a depth of $15 \mu\text{m}$ in the sample, using 775 nm femtosecond pulsed light (Fig. R1a) and 405 nm continuous wave light (Fig. R1b), respectively. The energy of the 405 and 775 nm light has been kept below saturation to avoid artifacts, as characterized in the power-dependent photoconversion curves in Fig. R1d and R1e, respectively. Using the 775 nm illumination shows a clear axial confinement, that is absent with the 405 nm illumination (Fig. R1f). A similar result as that for the femtosecond 775 nm light has been found also for illumination with a

Figure R1. Optical sectioning for different photoconversion wavelengths. *E. coli* expressing miRFP713 illuminated with different photoconversion-inducing illumination sources: 775 nm (120 fs, 80 MHz) (a), 405 nm (CW) (b), and 775 nm (550 ps, 40 MHz) (c). (a–c) The photoconverted area is at $15 \mu\text{m}$ from the cover glass, and in the images the blue-shifted channel after photoconversion is reported for the xy plane at $z = 15 \mu\text{m}$ (up) and the xz projection (down). (d–e) Photoconversion yield for the blue-shifted channel at increasing laser power of the photoconverting-inducing illumination at 775 nm (120 fs, 80 MHz) (d) or 405 nm (CW) (e). The vertical dotted lines specify the energy of the experiment in panels (a–b). It is important to note that these images have been acquired on a scanning system, therefore the effective energy per pixel is convoluted with the scanning. Each reported data point is the mean, and the shaded area is the SD, of the photoconversion yield for the bacteria enclosed in an area of $15 \times 15 \mu\text{m}^2$. (f) Intensity profile along the z-axis for the area enclosed in the dotted lines of panel a-to-c. All experiments have been repeated twice for each of the conditions.

picosecond 775 nm laser source (Fig. R1c). This experiment confirms the hypothesis of a non-linear process of photoconversion for the 775 nm light with respect to the linear photoconversion for the 405 nm light.

The data have been incorporated in Figure 1 (Fig. 1g) to give a better description of the mechanism and the characteristics that are expected in the application.

Further, to rule out any photodamage the authors are advised to provide datasets where the photoconverted cells successfully undergo division. No need to do lineage tracing.

We thank the reviewer for pointing out the importance of assessing photodamage when introducing photocontrollable FPs, given the intrinsic requirement for an additional illumination step. We further assessed this in two different ways by comparing the energy used in our studies with the standards reported in the literature and by experimentally investigating photodamage for our specific application in zebrafish imaging, both showing the overall relatively low phototoxicity risks of applying the photoconversion process of miRFPs.

We used three modalities of illumination to induce photoconversion and their energy requirement are within the ranges that have been used and characterized in the literature for cell division. Waldchen et al, 2016 *Plos One*, doi: 10.1038/srep15348 extensively characterized the photoresistance of transfected cells upon irradiation at different wavelength using cell division as the assessment parameter. They characterized how the phototoxicity increases with decreasing wavelengths, identifying the limit of $\sim 50 \text{ J/cm}^2$ for 405 nm irradiation and $\sim 1 \text{ kJ/cm}^2$ when moving toward the NIR (their experimental setting stop at 640 nm). In our photoconversion studies we are within the limit defined by Waldchen et al, therefore we can expect a low impact of phototoxicity in our study with the ability of tuning the phototoxicity effect by choosing the most NIR shifted wavelengths for photoconversion. Nevertheless, careful considerations always need to be done when assessing phototoxicity, not only on the wavelength but also cell line, phase of the cell and illumination conditions.

Photoconversion light	Energy at saturation	Irradiance limit
UV/Vis continuous wave: 405 nm	$\sim 5 \text{ J/cm}^2$	$\sim 50 \text{ J/cm}^2$
NIR picosecond light: 775 nm	200 J/cm^2	$\sim 1 \text{ kJ/cm}^2$
NIR femtosecond light: 670 – 800 nm	4 J/cm^2	-

With a pulse width of 550–700 ps, illumination typically used in STED microscopy can still be included in the consideration of Waldchen et al. Typical levels of 775 nm light for STED microscopy can reach hundreds of kJ/cm^2 . Additionally, several considerations have previously been made on the compatibility of STED microscopy with live-cell imaging (Kilian et al. *Nat Method*, 2018, doi: 10.1038/s41592-018-0145-5) and we believe they are out of the scope of our work. However, it is important to note that the energy of 775 nm (at 550–700 ps) required for the photoconversion of miRFPs is $\sim 0.2 \text{ kJ/cm}^2$, while the energy of the 775 nm light required to achieve STED imaging with miRFPs is $\sim 50 \text{ kJ/cm}^2$ (Matlashov et al 2018 doi:10.1038/s41467-019-13897-6).

The peak values reached in femtosecond pulsed laser light can be order of magnitude higher than the average value (that is the one we commonly report), therefore it is fair to compare the energy required by two-photon photoconversion with what is commonly reported in the two-photon microscopy field. In the literature, power values ranging between 1 and 50 mW are reported, and the photoconversion that we report has its peak at around 4 mW at the back aperture for a dwell time of 3.14 μs .

The measured power density used for photoconversion of the miRFPs and other photocontrollable FPs are reported in Supplementary Note 5 as a reference for the topic of phototoxicity.

More specifically for the provided application on zebrafish embryos, we decided to test whether the photoconversion at the presented power density would induce any damage to the tissue (Fig. R2). Recording a transmitted image before and after the photoconversion reveals no differences: the tissue as well as the blood flow are intact. The blood flow is visible in the confocal image as a movement-induced blurring artifact, enclosed in the dotted orange lines in Fig. R2, and also in Supplementary Fig. S20 as longitudinal stipe artifacts linked to the movement along the axis parallel to the fish orientation.

Figure R2. Photodamage assessment in zebrafish after 775 nm light photoconversion. Transmission and fluorescence images of a mosaic expression of H2B-miRFP720 in zebrafish. The power density used was 1 kJ/cm² and the photoconverted area was $\sim 15 \times 15 \mu\text{m}^2$, corresponding to a value close to saturation of the photoconversion efficiency.

S2 - The colour yellow is barely visible. Please opt for another contrast fitting for PSmOrange.

We have updated the figure accordingly.

S8 – figure legends: Please replace ‘matains’ with ‘maintains’.

We have corrected the typo in the legend.

Reviewer #2

The manuscript reports on the photo-convertibility of fluorescent proteins (FP) to shorter wavelengths upon irradiation with near-infrared light. The studied FPs are called miRFP (as in monomeric, infrared red-fluorescent protein) and were originally developed a few years ago by one of the PIs (Shcherbakova et al, Nat Comm. 2016) and subsequently shown to be useful for live-cell super-resolution microscopy (Matlashov et al, Nat Comm 2020) by the two PIs of the present study.

Extending the color palette of FPs into the infrared spectral range is an important endeavor for live-cell imaging in tissue, not only to increase the number fluorescence signals that can be spectrally separated but also to exploit the fact that biological tissue can be imaged better at longer wavelengths where autofluorescence, absorption and scattering tend to be reduced.

What's new in this study is the observation that these miRFP proteins undergo a blue-shift in their spectral properties upon exposure to infrared light. This is a curious and counter-intuitive effect, which is potentially interesting, as most photo-conversions involve a red-shift to longer wavelengths.

We thank the reviewer for acknowledging the novelty of the presented photoconversion mechanism and the importance of photoactivation strategies for live-cell imaging.

In general, light-controlable spectral shifts can be a very useful feature for pulse-chase type experiments, but they may also be a bug that complicates quantitative measurements, if the fluorescence intensities or lifetimes cannot be fully accounted for.

It's a bit unclear if this may be the case here or not, because for instance the wavelength used for STED imaging by itself triggers the photo-conversion, convolving the imaging with the photo-conversion.

As underlined by the Reviewer, a fluorescent protein with photocontrollable behavior can be efficiently translated into a tool for live-cell applications only if its photophysical behavior is fully outlined and predictable.

As for this family of miRFPs and its use in STED imaging, we agree with the reviewer that the combination of STED microscopy and photoconversion can be misunderstood and requires a more in-depth explanation. Therefore, we have extended the consideration in a note (Supplementary Note 4), where we clearly explain the image formation process during STED imaging and how the photoconversion and STED are effectively sequential processes. Furthermore, a graphical explanation and energy considerations have been integrated in Fig. 5 of the main text and in the paragraph dedicated to the STED imaging in the main text.

In STED microscopy, a red-shifted donut-shaped beam (here 775 nm) is spatially overlapped to the excitation beam (here 640 nm) to suppress the fluorescence emission from the molecules when applying the beams at its periphery. Given the scanning nature of this nanoscopy technique, a point in space will experience the light of half of the spatial intensity distribution of the donut beam before being imaged/recorded (i.e., being at the center of the donut beam and therefore in the subdiffracted confined volume and allowed to emit fluorescence, Fig. R3a). If we calculate the energy delivered/accumulated to that point in space by the donut beam during the scanning, we can see that the threshold for photoconversion is exceeded long before the point is centered on the crest of the STED beam (Fig. R3b), let alone the central zero of the STED beam where it is allowed to emit. Thus, photoconversion will, with a high probability, occur long before the image formation of a specific molecule at the center of the STED beam. This is confirmed by the majority of the fluorescence signal being present in the blue-shifted channel (quantified in Fig. 5c of the main text).

Overall, it is not possible to not drive the photoconversion during a STED imaging, but by tuning the detection bandwidth the phenomenon can be accounted for or integrated in the same emission window, maintaining the possibility to quantitatively follow the signal overtime.

Figure R3. Photoconversion induced by the STED beam. (a) Representation of the STED beam at 775 nm during the scanning process of the image acquisition. A region of the sample will be exposed to all the light of one crest of the STED beam light pattern before ending up in the center of the donut where the fluorescence photons can be registered. The donut is calculated for an intensity of 775 nm light at the back aperture of 15 mW. (b) Building up of the energy level (blue line) based on the previously described scanning of the STED beam (the profile of which is reported in yellow line). The dashed grey line is the energy threshold for the photoconversion to reach saturation ($\sim 1 \text{ kJ/cm}^2$).

I have these two general criticisms:

- The benefit of the blue-shifted photo-convertibility remains a bit theoretical. If the authors could demonstrate more clearly its advantages over existing FP imaging strategies, it would increase the impact of the study.

The NIR spectral window, which is specific to miRFPs, makes it a unique tool for photoconversion experiments. To follow up on the criticism of the reviewer we experimentally demonstrated how those characteristics translate into new capabilities for imaging experiments. We highlighted two specific aspects: optical sectioning and multiplexed photoconversion experiments.

- The non-linear nature of the photoconversion at $>700 \text{ nm}$ translates in optical sectioning. Both fluorescence and photoconversion of miRFPs are in the far-red-to-NIR region of the electromagnetic spectrum, where scattering and absorption for biological tissue are minimized, allowing deeper imaging penetration and relatively low phototoxicity. Many publications already report applications and development of these probes for imaging purposes *in vivo* (Matlashov et al 2020, *Nat Comm*, doi: 10.1038/s41467-019-13897-6). If we add photoconversion to the context of highly three-dimensional systems typical of *in vivo* studies, selectivity in the optical axis is desirable. The non-linearity of the photoconversion at 775 nm illumination provides such optical sectioning, similarly to two-photon imaging (Fig. R1a and R1c). In contrast, the photoconversion at 405 nm, as a linear process, does not provide any selectivity along the optical axis (Fig. R1b).

This is another point of advantage with respect to currently available PCFPs, where the photoconversion is generally linear and attempts of achieving photoconversion with two-photon processes often results in a lower photoconversion efficiency with respect to the contrast obtained with a one-photon process (Dempsey et al, *Nat Meth*, 2015 doi:10.1038/nmeth.3405). Primed conversion (combination of blue and red/near-IR light) with a double-step excitation can provide some degree of optical sectioning, however some limitations remain. First, the use of blue light does not provide the same sample accessibility compared to illumination entirely in the NIR range. Second, this combined illumination modality requires high energy ($\sim 10 \text{ kJ/cm}^2$), which limits the temporal accessibility of the photoconversion due to the longer illumination needed.

- Proof-of-principle for multiplexed photoconversion strategies. In the previous version of the manuscript, we reported how the miRFP photoconversion could enrich the level of information achievable by photolabeling studies through multiplexing. A novel imaging strategy was presented to enable photoconversion of miRFPs and green-to-red GFP-like PCFPs in a compatible way. In terms of applications, we envisioned two possible scenarios, which were reported in Supplementary Fig. S19 (now Supplementary Fig. S21 and Fig. 3): spatiotemporally coupled and spatially/temporally uncoupled photoconversion of the same or multiple species of proteins. We experimentally tested that we can photoconvert two distinct proteins, LAMP1 (lysosomal-associated membrane protein 1) in lysosomes and PTS1 (peroxisomal targeting signal 1) in peroxisomes, and

Figure R4. Multiplexed photoconversion strategies for green-to-red and NIR-to-far-red PCFPs. (a) Emission spectra for Dendra2 and miRFP713 before (solid) and after (dashed) photoconversion. (b) UV/vis wavelength. The normalized photoconversion yield for miRFP713 and Dendra2 is compared over the same energy range of 405 nm light. (c) NIR wavelength. The energy dependence of the normalized photoconversion yield for miRFP at 775 nm (550 ps, 40 MHz) is compared to the primed conversion for Dendra2 at fixed 488 nm (0.13 kW/cm²) and increasing 775 nm light. The comparisons in (b, c) have been performed on a layer of *E. coli* expressing miRFP713 or Dendra2. Each data point is the mean and SD for bacteria enclosed in a field of view of $\sim 15 \times 15 \mu\text{m}^2$ (approximately 100 bacteria). (d) Spatiotemporally uncoupled photoconversion. Using photoconversion wavelengths unique for the PCFPs, like pulsed 775 nm for miRFP and low energy of 405 nm light for the green-to-red PCFPs, two different spatial locations and structures can be highlighted in the cells. (e) Spatiotemporally coupled photoconversion. The spatially localized illumination with 405 nm light at a high energy level photoconverts both PCFPs in the same area.

follow their trafficking independently at the sub-cellular level. This was only possible by independently photoconverting the PCFPs mEos and miRFP fused to PTS1 and LAMP1 respectively. Even if out of the scope of this work, we can imagine using this new capability to photo-label organelles such as mitochondria or lysosomes independently in specific compartments of polarized cells like neurons, such as the growth cone, and follow their dynamics.

To further demonstrate the benefits of the blue-shift photoconversion process we experimentally demonstrated also the spatially decoupled photoconversion strategy, investigating and comparing the energy requirements of the PCFPs to combine (Fig. R4).

In the previous version of the manuscript, we hypothesized that we could separate the photoconversion by using a selective wavelength for the photoconversion of each PCFP. In particular, 775 nm light for miRFP720 and primed conversion (488 + >640 nm) for Dendra2. This resulted to be unfeasible since the >640 nm light needed in primed conversion was an order of magnitude higher than what was needed for the miRFP, therefore inducing both the miRFP720 and Dendra2 photoconversion simultaneously. However, an analysis of the energy dependencies of the two PCFPs at different wavelengths reveal that illumination intensity can be used as a way of selection between different PCFPs (Fig. R4b–c). Dendra2 is extremely reactive to 405 nm but less so to 775 nm (when combined to a fixed 488 nm for primed conversion), while miRFP720 has a higher reactivity to 775 nm light than Dendra2. We are aware that this behavior might be different for different green-to-red PCFPs, as reported in extensive literature on primed conversion (Turkowsky et al, *Angew Chem*, 2017; doi: 10.1002/anie.201702870), nevertheless this demonstrates that given the knowledge on the power dependency of a combination of prospective PCFPs, a selective combination of photoconversion wavelengths and illumination intensities can be defined.

We perform a proof-of-principle experiment for this recording strategy on a bacterial layer composed of a mix of *E.coli* expressing either miRFP720 or Dendra2. We could selectively photoconvert either Dendra2 or miRFP713 by using 405 nm (1.5 mJ/cm²) and 775 nm (1.4 kJ/cm², pulse width 550 nm, Fig. R4c). Alternatively, a spatiotemporally coupled photoconversion can be achieved using 775 nm (16 kJ/cm², pulse width 550 nm) with 488 nm (4 mJ/cm², Fig. R4d).

The possibility to uncouple the photoconversion can be useful for marking the same species of proteins but photolabeled at different time-points or the same protein specie located in different part of a tissue which would not be possible with only one PCFP.

- The paper is not so well written or organized, jumping straight into technical details in the Introduction and Discussion sections, whereas a broader view of the context, motivation and significance of the study would help the reader judge its merits.

We apology for the lack of clarity. We have formatted and generally overhauled the text in the new version of the manuscript to more clearly structure the context and motivation, methodology, and findings.

Reviewer #3

Report to 393188_Pennacchietti et al.

This study presents properties of five derivatives of bacterial phytochromes tailored to minimal size, but still maintaining their capability of photoconversion and fluorescence emission. Fused proteins were generated with other proteins characteristic for various organelles and cellular compartments. The major intention on the molecular level was a variation in absorption and emission properties highlighting here the switch from the parental state to a blue-shifted photoproduct, in contrast to most GFP-derivatives that are mentioned for comparison.

Taking together the work in living cells, the application of various microscopic methods, and the data handling, all this is performed and presented with the well-known great expertise of this ensemble of authors. No critics here.

We thank the reviewer for the acknowledgement.

The treatment of the basics in phy photochemistry (abs and fluorescence), however, shows flaws and logical inconsistencies and needs thorough revision (see details further down). As a general comment, many sections of the paper can only be understood after detailed study of the SI material. To the understanding of this reviewer, SI, however, should only give details to the specialists and should be part of the main body of information.

Following up on the reviewer suggestions we have added the information to the main text of the manuscript; the reader should now be able to understand the concept, experiments and key aspects while reading only the main text. In particular, we enriched the Introduction with a more in-depth presentation of the miRFP family and the overall context of the work.

In general, the manuscript demonstrates that in many cases, also here, scholars live in a bubble of their own community. Many arguments, results, and comments cannot be understood without reading the cited literature. Acronyms and abbreviations are not explained. Nowhere in the manuscript is the origin of these miRFPs (species name?) mentioned. Also, changes by mutagenesis (amino acid positions and exchanges from/to) that yielded the generated five RFPs are not presented.

We apologize for the lack of clarity in the origin of the miRFP family as well as the previous usage of the same protein family. We have now given more space to the introduction of those variants and the convention about their nomenclature.

As a summary, the family of miRFPs comes from engineering efforts toward optimized NIR fluorescent protein tags able to combine high brightness with monomeric state and therefore live-cell compatibility, ranging from tissue to sub-cellular and nanoscale imaging. The five specific variants that we have characterized in the study come specifically from three publications:

- miRFP713: Matlashov et al, *Nat Comm*, 2020, doi: 10.1038/s41467-019-13897-6.
- miRFP720: Shcherbakova et al 2018, *Nat Chem Biol*, doi: 10.1038/s41589-018-0044-1.
- miRFP670, miRFP703, miRFP709: Shcherbakova et al, *Nat Comm*, 2016, doi: 10.1038/ncomms12405.

A cartoon of the 3D crystal structure would be more than helpful. It is assumed that the naming of the compound, example 'miRFP670' identifies the absorption maximum. Detailed information can only be extracted from the SI-table.

We have now moved the Supplementary Table into the Main text, and we also added the 3D structure of a typical miRFP as Supplementary Figure 1.

Comments to selected paragraphs

1. 60/61

The presentation of bacterial phytochromes focussing on the here employed proteins should contain an additional information, as such that some of these bacterial proteins are so-called ‘bathy’-phys (this difference is to be mentioned and cited) highlighting the advantage that the parental state is the one absorbing at longer wavelengths. So, this phenomenon is well known in the community and not a novelty.

Bacterial phytochromes (BphP) are generally characterized by a photoswitching mechanism between a red absorbing (Pr, maximum at ~ 700 nm) and far-red/near-infrared absorbing (Pfr, maximum at ~ 750 nm) state (Fig R5a). In a prototypical BphP, the Pr state is the dark-adapted state and the Pfr state can only be reached by photoactivation. From the Pfr state the system reverts to the Pr state either by photoactivation or by thermal relaxation. Instead, for the “bathy”-phytochromes the Pfr state is the dark-adapted state, but a similar switching mechanism between the two states is present.

Given the above definitions, the mechanism that we identify and describe in the paper does not correspond to a bathy-phytochrome: even if the spectral photoconversion has a direction that agrees with a bathy-phytochrome, i.e. a blue-shift, the spectral properties and the nature of the two states differ from the reported definition. Specifically, the miRFP family of proteins have been engineered to stabilize the Pr state and hinder the Pr→Pfr transition by stabilizing D-ring positioning in the Pr state by introduced hydrogen bonds with its immediate protein backbone. This is confirmed by looking at the absorption and emission spectra of the miRFPs, which are typical of a Pr state of canonical BphPs, as well as the crystal structure of the proteins (Zhu et al 2015, *Sci Rep*, doi: 10.1038/srep12840).

In the panorama of reported photoswitching in BphPs, the photoconversion of miRFPs share some similarity with the photoswitching of *RpBphP3*, where the light-induced isomerization of the chromophore converts it from the Pr form to a form absorbing at shorter wavelengths (Pnr, for near-red, maximum at ~ 640 nm, Fig. R5b)(Giraud et al., *J Biol Chem*, 2005, doi: 10.1074/jbc.M506890200; Yang et al., *Structure*, 2015, doi: 10.1016/j.str.2015.04.022). However, the miRFP photoconversion maintains a fundamental difference with respect to the *RpBphP3*, and that is the irreversibility of the photoconversion (Fig. R5c).

Fig. R5. Simplified photoswitching scheme for the majority of BphPs (a) and for the specific case of *RpBphP3* (b). The circular arrows highlight how the photoisomerization of the ring D proceeds by multiple intermediates of which only the first is light-induced. The lack of extensive characterization for *RpBphP3* does not allow to specify key steps of the photoswitching, but a great similarity with other BphPs is hypothesized (Giraud et al., 2005). The dashed lines go from the dark-adapted state to the photoproduct for the reported definition.

Nevertheless, this similarity allows formulating a hypothesis on the miRFP photoconversion mechanism, which could provide mutagenesis strategies for optimization of the photoconversion.

1. 63

Here, an improvement of the fluorescence is mentioned as 20-fold. This value sounds impressive, however, a principal drawback of many – including those used here – FPs of various origin is their extremely low genuine fluorescence efficiency. This holds true also in this manuscript, as an absolute parameter of the fluorescence (always shown as normalized to 1.0) is never given. Use of a well characterized reference compound would have been most welcomed.

In the section mentioned by the reviewer we were citing previous work that characterized PAiRFP (Piatkevich et al 2013, *Nat Comm*, doi: 10.1038/ncomms3153), which is an example, despite instable in the converted form, of photoactivatable mechanism in NIR FPs and a point of reference for us.

The miRFPs reported in this work are the result of previous engineering efforts, which screens for brightness and compatibility with advanced microscopy (Matlashov et al 2020, *Nat Comm*, doi: 10.1038/s41467-019-13897-6; Shcherbakova et al 2018, *Nat Chem Biol*, doi: 10.1038/s41589-018-0044-1; Shcherbakova et al 2016, *Nat Comm*, doi: 10.1038/ncomms12405). In the following table, which is part of their original publications, a detailed characterization of miRFP brightness and two comparable proteins is provided.

Monomeric NIR FPs	Parental BphP	Ex, nm	Em, nm	ϵ , M ⁻¹ cm ⁻¹	Φ , %	Molecular brightness vs miRFP720, %	Effective brightness in HeLa cells vs miRFP720, %	Ref
mIFP	BrBphP	683	705	65900	6.9	76.1	13.9	[1]
SNIFP	DrBphP	697	720	149000	2.2	54.8	n.a.	[2]
miRFP670	RpBphP1	642	670	87400	14.0	204.7	67.9	[3]
miRFP703	RpBphP1	674	703	90900	8.6	130.8	49.8	[3]
miRFP709	RpBphP1	683	709	78400	5.4	79.8	29.0	[3]
miRFP713	RpBphP2	690	713	99000	7.0	115.9	94.9	[4]
miRFP720	RpBphP2	702	720	98000	6.1	100	100	[5]

Ref: [1] Yu et al. 2015 doi: 10.1038/nmeth.3447. [2] Kamper et al 2018 doi: 10.1038/s41467-018-07246-2. [3] Shcherbakova et al 2016, doi: 10.1038/ncomms12405. [4] Matlashov et al 2020, doi: 10.1038/s41467-019-13897-6. [5] Shcherbakova et al 2018, doi: 10.1038/s41589-018-0044-1.

Additionally, in line with the previous general comment of the reviewer, we have clarified the origin of the studied miRFP proteins in the Introduction section of the manuscript.

The decision of presenting the fluorescence as normalized to one (as in Fig. 1b, 3a, and 4a of the revised manuscript) is to allow easier spectral comparison between different proteins, where the absorption and emission spectra rather than brightness are the main parameters of the reported data. For the rest of the figures, the main reported quantification is the photoconversion ratio or fold of conversion, seeing as this is the main aim of the current work.

In this context, several of the presented emission spectra are suffering from a low S/N ratio pointing to either a low expression of these proteins or a very low fluorescence efficiency.

The low signal-to-noise of the emission and excitation spectra is to be attributed to the modality of recording rather than the absolute brightness of the studied miRFPs. The spectra are recorded with the spectral detection of a microscope and not with a traditional spectrophotometer. This method unfortunately suffers from lower signal-to-noise ratio; however, it gave us the possibility to directly relate the spectral change in the sample of interest and under imaging conditions (exactly the same laser source as used in the applications).

The authors claim a ,high sensitivity to day light‘, yet, the quantum yield for fluorescence – as said above – is marginal. This reads like a contradiction.

The “high sensitivity to daylight” in the sentence referred to PAiRFP and not miRFP. We have now changed and expanded the Introduction and more clearly separated the two groups of proteins to avoid possible misunderstanding.

PAiRFPs are here reported as the closest example of photoactivatable fluorescent proteins engineered from BphPs that enabled bioimaging application (Piatkevich et al 2013, *Nat Comm*, doi: 10.1038/ncomms3153). The PAiRFP is a bathy phytochrome, whose photoactivation mechanism can be leveraged in photolabeling and contrast enhancement. However, this tool suffers from two main drawbacks: (1) instability of the photoactive form that slowly reverts to the dark-adapted non-fluorescent state, and (2) extreme reactivity to light, reducing the effective contrast enhancement achievable unless optimally kept in the dark.

1. 74

The rationale of this paragraph at the end of the introduction does not easily disclose to the readers as it leaves them somehow confused. In the beginning the authors emphasize the advantages of the miRFPs as being extremely red shifted advantaging over autofluorescence and other obstacles and allowing more easy multiplexing by extending the wavelength range for applications. So, why a blue shift?

We apologize for the confusion, indeed the main strength of miRFP photoconversion is that both forms, converted and not, feature absorption and emission spectra that are red shifted compared to currently available photoconvertible FPs from the GFP family (emission peak at ~ 510 nm/converted at ~ 610 nm).

The emission spectra of the ground/non-converted form peaks at ~ 700 nm, while the photoconverted form peaks at ~ 650 nm, which is indeed 50 nm blue shifted but still in the proximity of the “transparent” window, i.e., a wavelength range of the visual spectrum at ~ 650–900 nm that is relatively free of cell-inherent absorbers and therefore relevant for bioimaging of animal cells.

1. 85

The authors might add in brackets the maxima for absorption and emission (although the parameters are given in the SI material). This would make further discussion more easy, see 1. 87 (ca. 53 nm blue shift: from where to where is this blue shift?)

To better clarify the spectral shift for the different miRFP variants and not confine the information in the supplementary material, we have now moved the Table to the main text.

1. 93/94

Determining the yield of conversion: this reviewer is not convinced that the ratio of fluorescence is a good measure for this photochemical process, except the authors have determined independently the fluorescence

quantum efficiencies of both states (that surely are different!!). If this has been taken into account, it should be mentioned.

In choosing the assessment method for the photoconversion we have adhered to previously reported studies (see as an example Dempsey et al, *Nat. Commun.*, doi: 10.1038/nmeth.3405). For further considerations on the pitfalls and strengths of this metric we have strengthened Supplementary Note 1.

Compared to the GFP-like PCFPs, the spectral shift of miRFPs is smaller and therefore crosstalk might interfere with the estimation. This effect will specifically affect the measurement as:

- (1) A non-zero initial fluorescent signal for the blue-shifted channel, which can result in an overestimation of the yield of conversion.
- (2) An additional fluorescence signal for the red channel after photoconversion, due to the tail of the emission spectra of the converted form.

We agree with the reviewer that the ratio metric is not precisely reflecting the absolute rate of photoconversion at the single molecule level, especially because the quantum efficiencies of the different species are not taken into consideration in the calculations. However, the metric is reliable for comparative studies such as the characterization and optimization of the photoconversion process at different illumination wavelengths, illumination energy, and during thermal recovery, as in our scope.

1. 96-99

This seems to be trivial, as the conversion efficiency (if the absorption band is homogeneous) follows absorption band intensity = oscillator strength = probability to catch a photon.

We have now changed the sentence and more clearly explained the photophysical mechanism behind the photoconversion. This explanation is now under the Result section titled: *Characterization of the miRFP NIR-to-far-red photoconversion*.

Figure 1d informs both on the non-linear nature of the photoconversion, with a quadratic dependency on the energy for every wavelength, and on the spectral window in which it is possible to drive the photoconversion. Behind the photoconversion in the NIR there is a two-photon absorption (2PA). The efficiency of 2PA is resonantly enhanced in the spectral region of 710–810 nm due to the nearby one-photon transition of the Q-band, an effect which is widely reported for most tetrapyrrole (Drobizhev et al, *Chem Phys Lett*, 2002, doi:10.1016/S0009-2614(02)00206-3; Drobizhev et al. 2011, *Nat Meth*, doi: 10.1038/nmeth.1596). Therefore, the behavior of the dependency on the energy at each wavelength is the result of the balance between multiple possible transitions, each of which will have a specific efficiency in different spectral windows. In the energy diagram reported below we summarize some of the transitions that play a role in the NIR region.

Pathways that promote the photoconversion will be: one photon absorption combined with the excited state absorption (ESA) and 2PA. Pathways that will compete with the photoconversion will be: stimulated

emission and, since we are investigating an immobilized layer of bacteria, bleaching. The balance between the two will define the maximum yield of photoconversion reachable at the different wavelengths of illumination and the speed of the process.

We have now updated Fig. 1c (and more in detail in Supplementary Fig. 6c) to reflect this complexity and clarified the description of the mechanism in the main text.

l. 117

This paragraph outlining the spectral properties of the Soret is surely correct, however, own experiences in phy spectroscopy have told that this band is of low use for activation phy proteins for (i) a lower efficiency (correctly cited), and (ii) the shift of the Soret band upon conversion is so small that selective excitation seems to be impossible.

Further, the absorption is placed in the region of autofluorescence caused by flavins, if the experiment goes to living cells. The authors might consider to minimize this paragraph in length. This aspect is picked up again in l. 134. So, it might be worth to present the spectra (abs, fluo) for the Soret band.

The use of the Soret band for excitation only discloses later when the same wavelength is used for two different photolabels with same excitation wavelength range but differently tagged proteins.

The investigation in the UV/vis range is important for two main reasons:

- The complete characterization of the NIR-to-far-red photoconversion. The side-by-side analysis of the photoconversion in the NIR and UV/vis shows that the photoconversion results from an excited state of the mRFP proteins that can be reached either by one photon absorption at 405 nm or by two-photon absorption in the 700 nm range.
- The applicability in imaging. UV/vis is a very common wavelength range both for use with photoactivatable FPs and practically present in commercially available microscopes. Hence, unveiling the energy dependence of the NIR-to-far-red photoconversion at 405 nm illumination is fundamental for any multiplexing strategy of photoconversion and for providing a fair comparison between PCFPs. On the other hand, the possibility to use this wavelength, despite being less compatible with certain live-cell applications, broadens the applicability of the probe and means that it can be implemented in different microscopy architectures.

The photoproduct reached by illumination in the two spectral regions is the same, therefore in the study we have used both in the live-cell experiments.

l. 207

The advantage of not damaging DNA by 700 nm light is apparent and trivial. DNA absorbs at 260 nm (maximum), and even excitation of the Soret band around 400 nm would not be harmful (if wavelength selection would be performed properly).

In the study of this NIR-to-far-red photoconversion we have explored a big range of energies and wavelengths. The cited studies investigate and discuss the problem of light induced phototoxicity when high energies are used in non-linear imaging, such as two-photon fluorescence microscopy. In the paper of Post et al, *FEBS Letter*, 2005, doi: 10.1016/j.febslet.2004.11.092 the authors specifically investigate how the different modality of photoconversion of PAGFP (both 1-photon and 2-photon) in *Drosophila* embryo would have influenced the biological system, by tracking nuclear viability. They conclude that only a shift to 2-photon photoconversion at less than 1500 GW/cm² for 820 nm could maintain the viability in whole mount embryos. Our illumination is around 1–2 MW/cm², therefore well below the limit of viability identified by the authors.

Considering that there is a femtosecond pulsed laser of the same kind used in two-photon fluorescence microscopy among the different laser sources we are using, we aimed to provide a reference to access photodamage.

Furthermore, we included a direct comparison of the photoconversion energy required by miRFP in respect to state of art photoconversion mechanism of PCFP, which can be used to assess photodamage and phototoxicity. We have summarized this in the Supplementary Note 5.

1. 210

This paragraph required several-fold reading to understand the message. Also, as at first glance some of the arguments appear to contradict generally accepted phy results.

This paragraph is in the Discussion part of the manuscript and obviously should be considered as the discussion of possible/hypothetic mechanism(s) of the blue-shifted (hypsochromic) miRFP photoconversion. To elucidate exact mechanism of the photoconversion requires crystallization and determination of structure of the chromophore and its immediate environment of the photoconverted protein species, as well as rigorous mass-spectrometry investigation of the isolated photoconverted chromophore. This is a separate large project, which is beyond the scope of this paper that reports, for the first time, the unique phenomenon of the NIR-induced fluorescence hypsochromic shift in bacteriophytochrome-derived NIR FPs.

There are several aspects to address:

(i) why does the excitation not induce isomerization of the D-ring? If mentioned in the text above, the reason for this statement is not clearly outlined.

Because in miRFPs the Pr state is stabilized by engineered hydrogen bonds with the chromophore's ring D and side chains of the mutated residues in its immediate environment.

(ii) Where is the result that neighboring residues move in response to the ring movements? There is literature to this conformational change from other phys, but here, is there a crystal structure or mutagenesis experimental evidence (for second bond formation)? In addition, how is the 'transient bond formation' concluded?

Possible movement of the residues around the chromophore, as well as transient bond formation (e.g., hydrogen bonds) between the chromophore and its environment, in response to the photoconverting light are just the hypotheses that are proposed in the Discussion part of the manuscript.

(iii) Why shows the denatured protein a higher mobility in SDS-PAGE? Proteins under these conditions are denatured and adapt a spheric, oblong-oval conformation that not necessarily is disturbed by an additional bond (1. 219).

The increase of the electrophoretic mobility on SDS-PAGE gels of denatured miRFPs, resulting from the formation of additional covalent bond between the biliverdin chromophore and protein polypeptide, have been shown by Verkhusha and Lagarias groups in the past (e.g., Baloban et al, *Chemical Science*, 2017, doi: 10.1039/c7sc00855d)

(iv) 'Chromophore protonation': Detailed studies have shown that phys carry the chromophore in the protonated state in both states parental and photoproduct. This feature has been intensely discussed in the

phy literature, and also deprotonation/reprotonation is known, e.g., meta-R II intermediate in plant phy that still carry the chromophore protonated in both states. Also deprotonation in few CBCRs have been reported, still with intense discussion in the community. Thus, the comment that the chromophore is protonated seems to be in agreement with common knowledge and should not be taken as a special exception. Also, the authors should be aware that bilin protonation in general causes a BATHOCHROMIC shift. If the authors wish to keep the protonation argument they should make this more clear by a detailed discussion.

We do not make an exception about possible protonation of the protoconverted chromophore in miRFPs. We have revised this place in the text.

Indeed, photoconversion to Pfr state, accompanied by chromophore protonation, induces the bathochromic shift in absorbance. However, in miRFPs the chromophore is stabilized in the Pr state, and its protonation may cause its hypsochromic shift observed experimentally in this paper. Notably, in another type of FPs, such as multicolor FPs of the GFP-like protein family, the chromophore protonation results in the hypsochromic shifts (e.g., Shcherbakova al, *Current Opinion in Chemical Biology*, 2014, doi: 10.1016/j.cbpa.2014.04.010).

l. 373/4

The authors record here fluorescence. Thus ,fluorescence‘ should be added: ... and fluorescence was recorded in ...

We have corrected the text accordingly.

l. 375, change tense (characterize): ... to test and to characterize ...

The text has been updated.

l. 384, add comma: .. dependency, and lifetime

The text has been updated.

l. 386-388: this sounds confusing, as ,line average was tuned to reach different ..., EXCEPT (highlighted by reviewer) for ... 405 nm light, the others were set to a 405 nm light intensity of‘

So, what (which wavelength) was kept and which was tuned?

We have now divided the sentence into two separate ones. The reported value 4.8 J/cm^2 is the energy of 405 nm illumination used in the experiments where a fixed-power 405 nm illumination was used. These are the experiments in which we wanted to not see the energy dependence of the photoconversion process, but rather other characteristics like the change in lifetime, application in live cells, oxygen dependence and thermal recovery.

l. 391: ,... accounts for ... of 3..‘ 3 what?

We completed the sentence to report the energy of photoconversion.

l. 403 and following

This reads somehow confusing. The protein is denatured and this reviewer assumes it is no longer soluble, but will be found in the precipitate. If then the ,debris‘ is removed, also denatured protein might be lost. The authors should clarify this point.

Low-speed centrifugation (~1000 g) for a limited 5 min time was used to remove just extremely large protein aggregates and perhaps remaining after Ni-NTA purification agarose beads to avoid their sticking at the beginning of the running lanes of the SDS-PAGE.

Supplementary material

Table S1

It is assumed that ,excitation‘ maxima are absorption maxima, and should better be named as such.

The values reported in the table are in fact excitation maxima, not absorption maxima, as the recorded spectra are excitation spectra.

Notably, the lambda values in absorption and emission (Stokes shifts) become smaller the longer the absorption/emission wavelength are. This diminishes to some extent the proposed advantage of the novel compounds. Further, as this manuscript presents far red-shifted compounds the information on changes in the Soret band is incomplete, but is not helpful in a discussion of potential applications, the more so, as the nm-shift of the Soret bands in abs and emission is marginally small and do practically do not allow selective excitation and detection.

Supplementary Table 1 (now moved in the main text) summarizes the spectral shift induced by the photoconversion for each of the miRFP variants that we have studied at the different photoconversion wavelengths, both in the Soret band and in the Q band. We believe that the label of the table were misleading so we change them to be more adherent to what they report on. The new labels are: “PC yield for $\lambda^{PC, Q \text{ band}}$ “ and “PC yield for $\lambda^{PC, S \text{ band}}$ “.

For imaging purposes, the more attractive variants are the more red-shifted ones, and indeed most of the bioimaging examples we have reported are either with the miRFP720 or miRFP713 variants. From the previous characterization of the miRFP family as NIR FP tags, there is an additional parameter beside the emission wavelength, that makes miRFP720 and miRFP713 more attractive for imaging in live-cell applications: the higher effective brightness (molecular and cellular brightness) with respect to the other variants (see table reported in the previous comments or the original source, Matlashov et al 2020, *Nat Comm*, doi: 10.1038/s41467-019-13897-6).

Nevertheless, it is important to report the extent of this phenomenon also for the other variants, identifying the NIR-to-far-red PC as a common mechanism in the miRFP family.

We would like to underline that the photoconversion we observed is not reversible like the Pr/Pfr switch of canonical BphPs, therefore we do not expect to use an eventual shift in the Soret band as a way to selectively promote a forward and backward reaction.

Comment to note and fig N1. The authors show examples highlighting the principle for cases of large A/E shifts. Correct, however, they use as example (panel a) an absorption of 500 +/- nm that is not given for any of their compounds.

We believe that the NIR-to-far-red photoconversion mechanism that we introduce in the study differs, in many photochemical characteristics, to the one currently reported among fluorescent proteins. Therefore, the note specifically compares: GFP-like PCFPs with a red-shift (previously reported) and NIR PCFPs with a blue-shift (introduced here). In the figure we take as an example the green-to-red photoconversion of

mEos3.2 (a) and the NIR-to-far-red photoconversion of miRFP720 (b). This is the reason why the wavelengths of panel a in Fig. N1 are in the 500 nm region.

Further, both cases red- or blue shifts of the photoproduct does not escape the dilemma: panel a (for a large shift of 100 nm!) still has contributions of exciting also the photoproduct, and even if this can be minimized, there might appear a FRET process falsifying the result. This is even worse in panel b, where excitation of the parental state generates a significant amount of photoproduct, again causing false positive results. The authors address this point in their note, but clearly, precise manipulation of the data is required and may bear difficulties.

The FRET process has a range of action between 1–10 nm. Hence, both for monomeric fluorescent proteins freely diffusing in the cytoplasm of bacteria and labelled to structures such as histone, vimentin or lysosomes this effect can be disregarded as the common distance between FPs is larger.

Fig N2. Why do the authors excite at 405 nm? Apparently, this is the Soret band, but why not escaping autofluorescence of the cells and using a longer wavelength?

It is important to note that 405 nm is only used as a photoconversion wavelength and not for fluorescence excitation. The photons emitted during 405 nm illumination are not recorded, which free the experiment from any potential 405 nm induced autofluorescence.

Figure R6. Comparison of lifetime and spectral changes between photoconversion triggered by 405 nm (a) and 775 nm (b). The photoconversion is reported both spectrally (middle column) and in fluorescence lifetime, through the phasor plot obtain by summing up the multiple spectral windows between 640 and 740 nm. The photoconversion is done on *E. coli* expressing miRFP720.

We also characterized the fluorescence lifetime response of the photoconversion. We have compared the spectral and fluorescence lifetime characteristics upon 405 nm and 775 nm photoconversion and have found comparable behavior. As such, since the equivalency of the two wavelengths to induce the photoconversion is in this case ensured, the result is independent from it.

Reviewer #4

The paper is interesting and innovative, but has also several weaknesses as will be pointed out below. The major concerns with the manuscript are that

- 1) The main aim of the paper is unclear.
- 2) Statistics and information about reproducibility are lacking
- 3) Many of the conclusions drawn are discussed at a superficial level

Key results

The manuscript contains novel concepts. Particularly the approach of employing photoconversion to enable intracellular diffusion studies in combination with STED; however, the key finding is not clearly defined. Is it the discovery of the blue shift photoconversion of the biliverdin chromophore using NIR excitation that is the primary results, or is it the technical realization of a sophisticated approach combining photoconversion in a more general sense? Based on the title the blue-shifted photoconversion using NIR is highlighted as the discovery, but as the manuscript also presents data in which 405 nm excitation is utilized for photoconversion, as well as another fluorophore which is green-to-red shifted (rather than NIR-to-far-red shifted), this is confusing. Furthermore, a mechanistic understanding of the “discovery” of blue shift in the specific fluorophore is not adequately substantiated, which is a weakness.

Following the reviewers’ suggestions, we have now reorganized the text to achieve a clearer structure and better deliver the presented findings.

At the core of the paper is the discovery of the novel photoconversion mechanism in miRFPs. This directly translates to novel live-cell applications in the NIR spectral window, which not only complements the current available PCFPs tools of the GFP-family but add new spatially and temporally multiplexed possibilities.

Additionally, the non-linear nature of the process is pivotal for accurate spatial confinement, especially axially, where it can be used to select precisely single layer of cells or even specific compartment at the sub-cellular level.

From this consideration, the proof-of-principle of the photoconversion multiplexing emerges, where this novel photoconversion mechanism can be combined with that of green-to-red PCFPs to enable multiple photolabeling experiments.

We indeed added the data about 405 nm photoconversion since it is the standard in current tools and therefore important for direct and fair comparisons. Additionally, it enables multiplexed photoconversion experiments, which can be useful for specific applications yet easy to implement. In fact, the 405 nm is common in commercial setups and typically available in imaging facilities. Therefore, having the possibility to test the photoconversion in any microscope is to our view a positive aspect.

Validity

Although the data presented in each figure is analyzed and interpreted thoroughly, the overall robustness and validity of the approach is difficult to assess since statistics and reproducibility is not commented. This is a general weakness throughout the manuscript.

We apology for the lack of clarity on the statistics and reproducibility, we do believe this aspect to be of fundamental importance and we have improved it in the following ways:

- 1) All the experimental parameters for each figure as well as the data are collected in Supplementary Table 1.

- 2) We have adequately commented on the statistical information that was missing in the legend of each figure as well as in the *Statistics and Reproducibility* statement in the Material and Methods section.

The photoconversion mechanism has been investigated for different illumination conditions (wavelengths, pulse widths, repetition rates) showing the same behaviour in two biological systems, both bacteria and eukaryotic cells (Fig. 1–2, Supplementary Fig. S2–S14). For each imaging strategy we report in the paper at least two representative examples, one in the Main Text (Fig. 4, multiplexing; Fig. 5, combination with STED imaging; and Fig. 6, *in vivo* imaging) and one in the Supplementary Material (Supplementary Fig. S16, multiplexing; Supplementary Fig. S18, combination with STED imaging; and Supplementary Fig. S20, *in vivo* imaging).

Significance

As pointed out above, the approach employing photoconversion to enable intracellular diffusion studies in combination with STED is of interest for the biophotonics community, as well as the possibility to combine the approach of photoconversion at multiple wavelengths enabling multiplexing; however this is not pointed out and supported as the main finding. Instead, several different sub-goals are highlighted as the main conclusions. Many of these are demonstrated as proof-of-concept, and discussed briefly as will be more specifically criticized below.

Thus it is advised that the manuscript is revised to limit the scope and thereby be more clear about the main findings and/or discovery. The revision should preferably include statistical analyses and elaborate on the theoretical anchoring to support the conclusions.

We appreciate the critical feedback and we re-structure the text highlighting the main claims.

A detailed answer is provided to the comment below.

Critical review of the conclusions drawn

Report on a NIR-to-far-red photoconversion phenomenon.

The manuscript contains data acquired from E.coli bacteria expressing miRFP709 and analogue mutants supporting this claim; however, the underlying mechanism being the phenomenon is not discussed at a satisfactory detail.

Neither of the mutants demonstrate Q-band emission (and therefor also unlikely absorption) of NIR at 775 nm, which is the excitation wavelength chosen for the photoconversion. It might be speculated that at this wavelength using 120 fs pulsed laser light at quite high powers (4 MW/cm²) non-linear excitation routes (targeting the Soret-band?) might come into play; however, nothing is stated about this. Instead, it is vaguely stated that “Likely, the blue-shift of the photoconverted form is caused by disturbance of the π -conjugated system in the BV chromophore”. If, for example, two-photon excitation is expected to perturb the π -conjugated system, this should be further supported and related to the Soret-band absorption. Furthermore, it would have been appropriate to include two-photon excitation and cross sections data to support the mechanistic claims. In fact, the term two-photon excitation is just briefly mentioned in the discussion section and not explained in the context of data, which is a weakness.

We apology for the lack of clarity in the explanation of the photophysical mechanism behind miRFP photoconversion. In the revised manuscript we added new experiments and explanations to characterize the phenomenon and support the claim of its non-linear nature.

The experimental evidence collected suggests that the photoconversion proceeds from an excited state that can be reached either by irradiation in the Soret band (i.e. 405 nm), or at wavelengths > 670 nm (including 775 nm).

In the Soret band the photoconversion is linear suggesting a one-photon absorption (1PA). For illumination at wavelengths > 670 nm photoconversion is non-linear, specifically quadratic with the light intensity. The photoconversion is therefore the result of a two-photon absorption (2PA). The efficiency of 2PA is resonantly enhanced in the spectral region of 710 – 810 nm due to the nearby one-photon transition of the Q-band, effect widely reported for most tetrapyrrole (Drobizhev et al, *Chem Phys Lett*, 2002, doi:10.1016/S0009-2614(02)00206-3; Drobizhev et al. 2011, *Nat Meth*, doi: 10.1038/nmeth.1596). Absolute 2PA cross-section have also been measured for iRFP proteins, miRFP ancestors, confirming a similar enhanced 2PA in the Soret region, in this study identified in the 890 – 950 nm spectral region (Piatkevich et al, *Biophysical Journal*, 2017, doi: 10.1016/j.bpj.2017.09.007).

When measuring the action spectra of the photoconversion we report on the final product induced by the 2PA. Therefore, the peak position will be strongly influenced by the Q-transition, being blue-shifted with respect to the pure 2PA cross-section because of the resonance enhancement. The complexity of this interaction is reflected in the different efficiency and power dependencies reported in Fig. 1d at different wavelengths of excitation moving from a spectral region of maximal overlap with the Q-band to the more NIR wavelengths. Respect to the simplified diagram reported in Fig. 1c, more processes need to be added in the energy diagram (we report a more complete energy diagram below as well as in Supplementary Fig. 6c). In particular when on the 1PA region of the Q-band excited state absorption represent a positive drive for the photoconversion, although the possible saturation of the S₁(Q) define the competition of stimulated emission and bleaching that push down the photoconversion yield. Moving above 720 nm the process is mainly driven by 2PA, probably with lower efficiency (lower absolute photoconversion yield) but less affected by bleaching that now come only from the S₂ state (slower decrease of the photoconversion yield).

In the NIR we have also explored different illumination modalities, namely femtosecond and picosecond lasers. Considering the 775 nm wavelength, with a pulse width of 120 fs saturation of the photoconversion is reached at 4 J/cm², while at 550 ps the mean energy needed is of 200 J/cm². The factor of difference between these two illuminations is of ~ 50 times, in line with what expected to achieve the same number of absorbed photons per fluorophore per pulse, n_a , that we assumed can trigger the photoconversion.

To formally compare the two excitations, we can express the number of absorbed photon per fluorophore per pulse as:

$$n_a = \frac{p_{avg}^2 \delta_{2PA}}{\tau_p f_p} \left(\frac{NA^2}{2\hbar c \lambda} \right)^2$$

where is the, p_{avg} is the average power, δ_{2PA} the 2PA cross-section, τ_p the pulse width, f_p the pulse repetition rate, NA the numerical aperture of the objective, λ the excitation wavelength, \hbar the reduced Planck constant, c the speed of light (Denk et al, 1990, *Science*, doi: 10.1126/science.2321027). Comparing the same fluorescent protein, miRFP720, in microscope of equal numerical aperture ($NA = 1.4$), the expected ratio between the powers to achieve the same number of absorbed photons is simply linked to the characteristics of the excitation:

$$\frac{p_{avg,1}}{p_{avg,2}} = \frac{f_{p,1}}{f_{p,2}} \sqrt{\frac{\tau_{p,1}}{\tau_{p,2}}}$$

that would give a theoretical ratio of ~34, in line with the experimental value.

In the discussion section (line 210-228), there is a paragraph more focused on the mechanistic understanding; however, it is difficult to connect this paragraph to the actual data presented. The paragraph brings up quite specific information about the protein structure, without connection to actual data. Bringing back to what is pointed out above, the key question still concerns how the excitation using NIR can induce this photoconversion effect. Thus it is advised that the mechanistic discussion is merged and integrated with the data commentary section.

To elucidate a mechanism of the photoconversion requires crystallization and determination of structure of the chromophore and its immediate environment of the photoconverted protein species, as well as rigorous mass-spectrometry investigation of the isolated photoconverted chromophore. This is a separate large project, which is beyond the scope of this paper that reports, for the first time, the unique phenomenon of the NIR-induced fluorescence hypsochromic shift in bacteriophytochrome-derived NIR FPs.

Here, in the Discussion part, we can just reasonably hypothesize about possible photoconversion mechanism(s), based on the limited biochemical analysis (e.g., Suppl Figure 22; new numbering) and on the published data on the chromophore behavior in bacteriophytochrome-derived NIR FPs. Going in more details in our hypothesis will result in pure speculation, which is not appropriate for the research paper.

Anyway, we have followed the suggestion of the Reviewer to separate the photophysical considerations about the photoconversion phenomenon and the possible/hypothetical molecular mechanism(s) of photoconversion itself.

Another confusion is that both NIR induced as well as 405 nm induced photoconversion are explored in the paper, but their presentations are not clearly separated and explained. For example, the photoconversion taking place in HeLa cells (Note in materials and methods cells are stated to be U2OS?) seem to be restricted to 405 nm induction. Also the reported lifetime shift from the photoconversion seem to rely on 405 nm irradiation. This mixing up and lack of systematic analysis is confusing to the reader and needs to be clarified.

Ultimately, irradiation with 405 nm and >700 nm give rise to the same photoproduct (Fig. R6). This is confirmed both by the spectral and lifetime profiles recorded after photoconversion-inducing illumination with the two wavelengths.

This is the reason why in the applications we have used the two wavelengths equally, choosing the one that matches the strategy of photoconversion that we wanted to explore. We understand that this have introduced confusion in the text, therefore we have (1) collected all the acquisition and sample parameters behind the images and data presented in the figures in Supplementary Table 1, (2) separated the two thematic blocks: photophysical description of the photoconversion process and applications of the photoconversion, and (3) clarified the impact of the chosen photoconversion-inducing wavelength and illumination source for the generation of the final photoproduct by comparing side-by-side the spectral and fluorescence lifetime behavior in Supplementary Fig. 6d-e.

The observed photoconversion is further claimed based on the observation of a shift in fluorescence lifetime. Data supporting the observation of increase of fluorescence lifetime is presented for photoconversion using 405 nm excitation (and not supported for the NIR induced photoconversion?). It is unclear for what reason the lifetime measurements are included. Is it to support the understanding of the underlying mechanism, or more as technical means to improve imaging contrast? Stated in the manuscript

is a sentence “The photoconversion introduces an unbalance in the protein population by shifting it towards the blue shifted form with a longer lifetime”. What is the rationale for drawing this conclusion, and what is meant by “unbalance in protein population”? Instead it would be interesting to elaborate the discussion about the origin of the observed shift in lifetime from a more fundamental level. Does the longer lifetime relate to a shift of the natural lifetime of the fluorophore due to the hypsochromic shift, or can it be speculated that the non-radiative decay routes of the chromophore-protein complex is affected?

Fluorescent lifetime is an important parameter that both reports on the photoconversion mechanism and provides an additional dimension that can be used for imaging applications, for example to separate with higher accuracy than the sole spectral shift, the converted and not-converted form. We have restructured the text to better divide the considerations that belong to one or the other motivation in the Result section that discusses the fluorescent lifetime recordings. Here we want to clarify those findings/considerations:

- The fluorescence lifetime measurements reveal an initial heterogeneity of the miRFPs, before photoconversion. Indeed, the time-domain can provide insight into heterogeneity that cannot be achieved through conventional steady-state fluorescence measurements, which can only provide an average view. In particular, spectrally resolved fluorescence lifetime imaging enables exploration of the wavelength-dependence of the relative contribution of two or multiple fluorophore/species. In the phasor plot, the mix of two components results in the fluorescence lifetime moving along a straight line. At the extremes of this line are the conditions of maximal contribution of the species that is predominant in that wavelength window, while in the in-between spectral region the lifetime will be the linear mix of them (Hanley Q. S., *J. R. Soc. Interface*, 2009, doi: 10.1098/rsif.2008.0393.focus).

Overall, the mix observed in fluorescence lifetime measurements reveals a degree of ground state heterogeneity, where at least two different conformations are coexisting at equilibrium. The photoconverted species has lifetime and spectral characteristics similar to one of the two populations, however it is impossible from just this parameter to know if the photoconverted form is the same form already present at thermal equilibrium (Tang et al., *Chem Rev*, 2021, doi: 10.1021/acs.chemrev.1c00194). Further studies correlating the observation of fluorescence lifetime with ultrafast spectroscopy will allow to clarify the detail of this heterogeneity.

The sentence highlighted by the reviewer is grounded in this context. We wanted to highlight how before photoconversion there is a mix of two populations of the miRFP: one with a faster lifetime and one with a slower lifetime. After photoconversion the slower lifetime population is predominant. We have made this sentence more explicit and added references that allow the reader to better follow the reasoning of our analysis.

This spectral dependency of the unconverted form suggests an initial heterogeneity of miRFP720 in the equilibrium state that converges in the slower form upon conversion³⁶ (Fig. 2c and Supplementary Note 2).

[36] Hanley Q. S., *J. R. Soc. Interface*, 2009, doi: 10.1098/rsif.2008.0393.focus

- Knowing and characterizing the fluorescence lifetime upon photoconversion is of invaluable help when fluorescence intensity is not a fully reliable readout for imaging. Such a condition is common for *in vivo* imaging, where the detected fluorescence intensities can be low and/or highly contested with a strong background. Furthermore, as we demonstrate in Fig. 2d-g, the knowledge of the fluorescence lifetime can be used to further enhance the contrast of the imaging after photoconversion, since the isolation of the two components allows to correct for bleed-through between the two signals.

The “prolonged stability” is promoted in the discussion as an advancement; however, the present data is not discussed in the context of state-of-the-art in the field. What is the expected stability of the photoconversion technologies known today? This needs to be stated in order to support the statement.

The “prolonged stability” refers to the thermal stability of the photoconversion, where we see no return to the original near-IR form over an extended period of 12 hours (Supplementary Fig. 9). We have modified the term in the text accordingly for clarity.

The need for a close monitoring of this parameter is linked to the photoactive nature of bacteriophytochromes, where the Pr/Pfr switching can also proceed thermally with a variety of recovery rates (Fig. R5). Two major examples of Bphys that are meaningful as comparisons of the identified blue-shift of miRFPs are: the far-red-light photoactivatable near-IR PAiRFP, and RpBphP3 for which a Pr→Pnr hypsochromic shift has also been reported.

PAiRFP1 and PAiRFP2 are bacteriophytochrome-based photoactivatable FPs, engineered from a bathy BphP from *Agrobacterium tumefaciens* C58 called AtBphP2. As a bathy BphP, at thermal equilibrium the protein is in the Pfr form, where no significant fluorescence is recorded. After irradiation with 660 nm the fluorescent Pr state is reached. The photoactivatable behavior of PAiRFP1 and PAiRFP2 has allowed not only *in vivo* photolabeling, but also an enhancement in the signal-to-noise ratio for *in vivo* imaging. However, for these proteins, the photoactivation does not involve any chemical modification of the chromophore (except for protonation/deprotonation and isomerization) and thus the dark relaxation brings the system back with a half time of ~1 h (PAiRFP1) and ~4 h (PAiRFP2). This reset of the system imposes limitations on the applications compatible with these PAFPs.

Instead for RpBphP3, the Pr→Pnr hypsochromic shift has a high degree of similarity with the photoconversion that we have reported for what concerns the spectral characteristics. Both proteins start from a state excitable around 670 nm, identified as the Pr state, to then photoconvert upon irradiation with ~700 nm light toward the blue-side of the spectrum, ~ 650 nm. The thermic stability is exactly what distinguishes the photoconversion of miRFPs and RpBphP3. In the latter case the process is indeed reversible and Giraud et al have demonstrated that the system can be brought back to the Pr state either by illumination with 645 nm or by dark relaxation (Giraud et al., *J Biol Chem*, 2005, doi: 10.1074/jbc.M506890200; Yang et al., *Structure*, 2015, doi: 10.1016/j.str.2015.04.022). For miRFPs, we have observed no return to the Pr state neither by illumination nor dark relaxation, aspects that suggest that a different molecular mechanism is behind the here investigated photoconversion.

High spatiotemporal precision

It is demonstrated that the concept of photoconversion of the miRFP is compatible with STED microscopy, known to have superior spatial precision compared to e.g. confocal microscopy. However, it is unclear what is referred to in the context of temporal precision? This should be clarified, or the claim removed.

With temporal precision we wanted to highlight the compatibility of the photoconversion mechanism with the imaging time dictated by the microscopy technique under use, i.e. STED nanoscopy. More in general, the energy needed to photoconvert miRFPs to saturation (therefore at a good signal-to-noise ratio in the blue-shifted channel) is 0.2 kJ/cm² and 5 J/cm² for 775 and 405 nm illumination respectively. These energies can be delivered in a single point in time or space, where we can compress the dwell time down to a minimum, i.e. >3 μs. In the combination with STED imaging, where also the spatial shape of the beam plays an important role, this means that the photoconversion can be driven during the recording of the STED image itself, utilizing the depletion beam as a simultaneous photoconversion beam. We have performed calculations of the energy relationship between photoconversion and STED image acquisition in Supplementary Note 4 and also added the key aspects of this in Fig 5a–b.

More in general, the energy threshold for an efficient photoconversion is an important parameter for the applicability of photoconversion, especially in live-cell studies. The energy requirement for photoconversion translates in a minimum time of applied photoconversion-inducing illumination, and when the energy requirement is high the photoconversion can impose a rate-limiting step to the accessible dynamics.

Study dynamic processes in living cells

Photoconversion is reported to be demonstrated in mammalian cells. The manuscript includes 3-5 images from single HeLa(or U2OS?) cells, using fusion to different cellular proteins; however, statistics are missing. In order to claim successful photoconversion in mammalian cells, it needs to be stated what cell types the claim is restricted to and something about the success rate of the method. Was it a one time experiment, or could it be repeated?

We have now reported the specification on the statistical analysis in a dedicated paragraph of the material and method section as well as in the figure legend of the presented data. To enhance reproducibility and allow an easier comparison of the experimental conditions for each experiment in Supplementary Table 1 we collected and specified the cell type used for each example together with the image recording parameters.

For what concerns the success rate of the method, it is important to notice that for representative images in which we wanted to demonstrate the photoconversion, either in combination with other photoconvertible probes (Fig. 4 of the main text) or for *in vivo* application (fig. 5 and 6 of the main text), we have used the energy level identified by the calibration of photoconversion, a calibration performed either on bacteria or directly in mammalian cells on a day-to-day basis to have control over the fluctuation of the laser power and the different image acquisition settings (i.e. most importantly dwell time and pixel size). Using these calibrated energy levels, the photoconversion is ensured to take place with certainty. For the live cell imaging data, the reported experiments out of which the images are presented are representative of at least triplicates.

Data from a single multiplexing photolabeling experiment is demonstrated as proof-of-concept in living cells (Figure 3, cell type not stated in legend). It is claimed that “using this method, subsets of lysosomes and peroxisomes targeted with miRFP720 and mEos2 respectively were photolabeled and tracked over time”. Statistics are lacking. Furthermore, the biological implications of the findings in the experiments are not discussed. Is the finding realistic/unexpected? Is the method limited to this cell type, this organelle? What alternative methods is this approach complementary to?

Optical labeling provides spatial and temporal information through site-specific photoconversion, being it at the subcellular level or in a whole organism or tissue. Expanding this labeling strategy to a second optical label would expand its dimensionality, enabling the correlation in time or in space of multiple proteins or cells of interest.

The miRFP family, especially in its more red-shifted variants (miRFP713 and miRFP720), represents a unique complementary label for photoconversion multiplexing, with the possibility to be paired with the most numerous groups of PCFPs, the green-to-red PCFPs. Indeed, the GFP-like proteins are mostly spectrally overlapping. The blue-shift reported and engineered for dyes (addressed in some publications as “photoblueing”, Helmerich et al, *Nat Meth*, 2021, doi:10.1038/s41592-021-01061-2) have some point of overlap with the miRFP proteins, namely being in the NIR part of the spectrum and efficiently photoconvertible with two-photon excitation. Nevertheless, the photoconversion that has been reported for the NIR dyes is of greater magnitude respect to the miRFP variants, bringing the photoconverted specie to emit in the 500 – 600 nm region, therefore severely overlapping with the red-shifted form of the green-to-red PCFPs (Kwon et al, *Adv Sci*, 2016 doi:10.1038/srep23866 and Saladin et al, *Angew Chem Int Ed*, 2023 doi:10.1002/anie.202215085).

We have now commented on the statistical repetitions for the reported experiments. In Fig. 4 (previously fig. 2) is reported a representative image of the spatially coupled photoconversion multiplexing and another example is presented in Supplementary Fig. 16 (previously Supplementary Fig. 14). The success of the photoconversion, once we set the energy at the calibrated power to achieve saturated photoconversion for

both proteins (Fig. 4c), it is consistent for different cells. The main aspect that enters in the measurements, potentially damaging the imaging strategy, is the level of transfection for the two plasmids. For all our experiments we transiently transfect the cells and image them after 24 – 48 h. For two separate plasmids their ratio of expression is not linked and often an unbalance in the level of expression for the two proteins can happen. Therefore, the main obstacle to a successful experiment is normally the signal-to-noise level of the double transfection. Strategies can be implemented to optimize the level of expression of both proteins, although the current setting does not preclude the demonstration of the strategy in live cells with the target on highly dynamic organelles.

In Figure 4 g-h, data are presented from zebrafish embryo. It is stated that “single cells could be photolabeled without any sign of photodamage, opening new possibilities for tracking cell dynamics in vivo.” How was the viability and lack of photodamage assessed/validated in order to make this claim.

To assess the phototoxicity in the specific case of the Zebrafish embryos we have focused on the persistence of the blood flow, that appear as stripes in brightfield imaging (Fig. R2) or in fluorescence image also after 3 hours (Fig. 6 and Supplementary Fig. 19), and the absence of any lesions to the tissue caused by the photoconverting light. This assessment is now added and presented in Supplementary Fig. 20.

In line with the comments of the other reviewer about phototoxicity we decided to also add a dedicated note (Supplementary Note 5), where we contextualize the energy used for miRFP photoconversion with the one reported in the literature. The energy level needed are indeed inside the range of irradiance typical of other PCFPs and more in general super resolution imaging (Waldchen et al, *Sci Rep*, 2015, doi: 10.1038/srep15348, Alvelid et al. *Current Opinion in Biomedical Engineering*, 2019, doi: 10.1016/j.cobme.2019.09.009). This assessment confirms that the miRFP photoconversion process can be driven with very low risks of phototoxicity in both mammalian cellular and *in vivo* measurements.

Background of Photoswitchable fluorescent proteins

The general concept of photoswitchable fluorescent proteins is introduced in context of two specific references (Ref 1 and 2), restricted to co-authors of the present manuscript. It can be questioned if this is a true representation of the state-of-the art in the field. Photoswitching and photoconversion is of broad general interest e.g. in the fields of photopharmacology and optogenetics, and should preferably be introduced at a more general level.

With the first two references highlighted by the reviewer we wanted to present the possible mechanisms of photoswitching in terms of their chemical characteristics (ref 1: Shcherbakova et al, *Current Opinion in Cellular Biology*, 2014, doi: 10.1016/j.cbpa.2014.04.010) and in the context of what specific applications requires from such mechanisms to utilize them (ref 2: Shcherbakova et al, *Annual Review of Biophysics*, 2014, doi: 10.1146/annurev-biophys-051013-022836). To expand this collection bringing also other view in defining the topic of photoswitching in fluorescent proteins, we have now added the recent publication from the Nienhaus' group (Nienhaus et al, *Method App Fluoresc*, 2022, doi: 10.1088/2050-6120/ac7d3f).

In the very first few lines we want to set the boundary in which the manuscript belongs, and we believe that opening too much to incorporate the role of photoswitching in photopharmacology and optogenetics would be misleading to the reader. We recognize that photoswitching is a very broad field but the photoconversion we aim to describe here is the one of a fluorescent tag without an effector of a molecular function directly linked to it.

Additional comments

In the abstract, several claims are made at a more general level. Therefore, the abstract is not conveying in what specific way the work is advancing the knowledge in the field. The abstract should be rewritten to better reflect the scientific work, the findings made, and the advances of understanding.

We have changed the Abstract to follow the suggestions of the reviewer, in order to be more specific in what way this work represents new advancements. In short, with this work we are introducing a new photoconversion mechanism entirely driven in the NIR spectral window, the wavelength that induce the photoconversion are >700 nm and the emission and absorption spectral shift of the not-converted and converted species are NIR-to-far-red. The spectra and the kinetics of the photoconversion is advantageous for live cell and *in vivo* imaging application as it allows novel multiplexed experiments.

Figure 2. Are data from HeLa cells or U2OS as stated in materials and methods?

We apology for the mistake and have corrected the cell type for each figure, and also provided this information in each figure legend as well as in Supplementary Table 1.

Figure 4 a-f show data from a (single?) cell, while fig 4g-h demonstrate data from zebra fish larvae. It is confusing to combine these data in the same figure.

We have now separated the panels to two separate figures, Fig 5 and Fig 6, with the aim of more clearly separating the application areas for the reader. Fig 5 now presents the photoconversion in combination with STED microscopy, and Fig 6 presents the *in vivo* experiments from Zebrafish larvae.

The manuscript lacks a concluding paragraph that summarizes the main findings and how the paper actually advances the knowledge in the field.

Together with the reorganization of the text we have restructured also the concluding paragraph.

It would be advised to present an overview of what different types of organisms are investigated in the manuscript and for what purpose: *E.coli*, HeLa cells (or U2OS?), and zebra fish larvae. Particularly useful would have been to motivate the choice of model organisms in the introduction, in connection to biological relevance and stat of the art in the field.

The choice to express and to photoconvert the miRFP in these biological systems was done to highlight the general applicability of the phenomenon of photoconversion rather than carrying out one specific and in-depth biological investigation, which is out of the scope of this work.

We demonstrated that miRFP can be successfully expressed and photoconverted in very different systems, highlighting the robustness of the probe at different maturation condition and imaging condition. We opted for *E. coli* as example of prokaryote as very much used in bacterial research and provided a confine example of activation which can inspire new imaging experiments in bacteria films.

We chose HeLa as a commonly used model system for eukaryotic sub-cellular imaging studies. Here, we provided proof-of-principle experiments to track distinct organelles within the same cell, but a selected spatial and temporal start provided by photoconversion of two probes simultaneously.

We chose Zebrafish as an example of *in vivo* bio-imaging of a multi-cellular organism.

To help clarify the cell type and organisms considered we have collected this information in the added table of the supplementary material (Supplementary Table 1) and added this information wherever it was missing.

aPoint-by-point responses to reviewers of the manuscript NCOMMS-22-41203:

Blue-shift photoconversion of near-infrared fluorescent proteins for labeling and tracking in living cells and organisms

Francesca Pennacchietti, Jonatan Alvelid, Rodrigo A. Morales, Martina Damenti, Dirk Ollech, Olena S. Oliinyk, Daria M. Shcherbakova, Eduardo J. Villablanca, Vladislav V. Verkhusha, and Ilaria Testa

We appreciate the comments and the suggestions of all four Reviewers. All the inputs helped to improve both the structure and clarity of the manuscript.

In the revised version we performed new experiments to provide additional insight into the photophysical mechanism behind miRFP photoconversion, a new multiplexed photoconversion application, a direct comparison to primed conversion, and further assessment of phototoxicity. With the current work we want to report, for the first time, the unique phenomenon of the NIR-induced fluorescence hypsochromic shift in bacteriophytochrome-derived NIR FPs, as well as to demonstrate the impact of this novel photoconversion mechanism for bioimaging, where it enables several novel applications in live cells and living animals. The text now clearly outlines this perspective. We have expanded the Introduction section to provide a deeper context both of existing photoconversion mechanisms and the origin of the miRFP family. The Results section has been restructured to first provide the photophysical characterization, and then present the newly enabled imaging strategies, namely photoconversion multiplexing, combination with STED microscopy, and *in vivo* imaging.

Below we provide a point-by-point response to the comments of each Reviewers. The comments of the Reviewers are written in blue and our responses in black.

Reviewer comments

Reviewer #1

Overall

The manuscript has been much improved and streamlined since the last time I reviewed it. The authors have invested much thought into considering cross-talk/bleed-through effects for the conversion analysis which is commendable.

We thank the reviewer for his/her positive evaluation of our work.

Importantly though, the maximum conversion yield reported is rather low. In comparison to green-to-red photoconversion that can achieve up to 80-fold changes using dual-laser illumination via primed conversion (see Dempsey et al. 2015, Nat Meth) or to a lesser degree double digit fold changes using 405nm illumination due to various photochemical effects (see Thédié et al, J. Phys. Chem. Lett. and De Zitter et al., 2019, Nat Meth,). Hence, green-to-red photoconversion using primed conversion is still 10-fold higher than the values reported for the most optimal procedure of miRFP photoconversion selected by the authors. To be sincere and transparent to the interested reader, the reviewer suggests emphasising the conversion yield limitation in the discussion. The authors are also encouraged to mention that it is likely that further protein engineering efforts can e.g. convey higher photostability to the miRFP photoconversion versions as previously accomplished for green-to-red photoconvertible fluorescent proteins (see Mohr et al. 2017, Angew Chem, Turkowyd et al. 2017, Angew Chem) which will likely increase the conversion rate.

Following the reviewer's suggestion, we have expanded the discussion and provided new experiments to clarify the applicability of the miRFP photoconversion mechanism compared to previous PCFPs.

The maximum conversion yield is indeed an important parameter, but the exact value can differ depending on the used metric, and from sample to sample and instrument to instrument. In our study, as in the studies mentioned by the reviewer wherever reported in the supplementary materials (i.e., Dempsey et al. 2015, Nat. Methods, doi: 10.1038/nmeth.3405; Mohr et al. 2017, Angew. Chem., doi: 10.1002/anie.201706121; Turkowyd et al. 2017, Angew. Chem., doi: 10.1002/anie.201702870), the fold of conversion is calculated as the ratio between the fluorescence intensity after and before photoconversion in the detection channel of the photoconverted form, i.e., $\text{Fluo}^{\text{after-PC}} / \text{Fluo}^{\text{before-PC}}$. Using this metric in a direct comparison, the presence of any level of spectral crosstalk between the two forms and the presence of any blue-shifted species at equilibrium define the $\text{Fluo}^{\text{before-PC}} > 0$, as is the case for the miRFPs. Instead, for most green-to-red PCFPs, the excitation of the photoconverted red-shifted species does not cross the absorption spectra of the greenshifted species and at equilibrium all the population is in the non-photoconverted red-shifted form. Therefore, the fluorescence emission of the photoconverted red-shifted species before photoconversion is very close to zero, $\text{Fluo}^{\text{before-PC}} \sim 0$. With a value close to zero as the denominator in the ratio, the metric becomes extremely dependent on the background intensities of the detectors in use and on the acquisition settings (excitation power, dwell time, detection bandwidth etc.) for the photoconverted channel, $\text{Fluo}^{\text{afterPC}}$. We have discussed these considerations in Supplementary Note 1.

We also discuss how future miRFP engineering efforts can be directed to the optimization of the photoconversion mechanism in the Discussion section:

The photoconversion yield of miRFPs is currently lower than the one reported for GFP-like PCFPs. Nevertheless, future tailored engineering efforts have the potential to further optimize the miRFPs toward higher photoconversion yield, firstly by minimizing crosstalk and heterogeneity in the equilibrium state.

Furthermore, we envision that the use of the phasor approach that helped us identify the presence of some heterogeneity in the equilibrium state, where an amount of blue-shifted species is already present, would be helpful in guiding the mutagenesis. These efforts would be focused on both minimizing the equilibrium presence of the blue-shifted species, as well as increasing the spectral shift between the converted and nonconverted form.

To conclude, even if the calculated photoconversion yields of the current miRFP variants are an order of magnitude lower than green-to-red PCFPs, we believe that this NIR-to-far-red photoconversion mechanism can be a valuable tool due to the NIR spectral region, axial confinement, and lower illumination doses for conversion compared to primed conversion.

The reviewer is pleased to see the added discussion section.

I congratulate the authors for the FLIM data inclusion and their prominent placement into the main text. This important addition renders the comparison more robust given that photoconversion yields are currently rather low. The interested user will be very eager to use this alternative approach, as it can be conducted in a commercially available system as the authors demonstrated. I am though interested to know why the lifetime is several folds shorter than for other FPs?

We thank the reviewer for acknowledging the FLIM data and motivating us with her/his previous comments to further investigate the lifetime dimension.

Fluorescence lifetime for most FPs of the GFP family is in the range of 2.3–3.5 ns, with few exceptions in the extended interval of 0.7–5 ns (Mamontova et al., *Sci. Rep.*, 2018. doi: 10.1038/s41598-018-31687-w). On the contrary, for bacteriophytochromes fluorescence lifetimes in the sub-nanosecond domain are common, and miRFPs are in line with this general behavior. The short lifetime is generally linked to a greater instability of the chromophore in the pocket and the nature of the hydrogen bond network that results in multiple non-radiative decays to the ground state (Hontani et al., *Sci. Rep.*, 2016 doi: 0.1038/srep37362).

More specifically, the lifetimes in the miRFP family show a correlation with the emission spectra, with longer lifetimes, ~ 1.1 ns, for blue-shifted miRFPs and shorter lifetimes, ~ 0.7 ns, for red-shifted miRFPs (Rice et al., *Cancer Res*, 2015, doi: 10.1158/0008-5472.CAN-14-3001). This dependency is also shown in the reported measurement, Fig. 2a of the main text. The longer lifetime is associated with an increased rigidity of ring A for blue-shifted miRFPs (i.e., miRFP670), where two cysteines covalently bond the chromophore instead of one. This reduces the mobility, affecting the π -electron conjugated system, and minimizes the dispersion (Hontani et al., *Sci. Rep.*, 2016, doi: 10.1038/srep37362). Another factor that has been demonstrated to influence the fluorescence lifetime of this type of near-infrared FP is the hydrogen bond network of the chromophore, specifically demonstrated in iRFP713 (Zhu et al., *Sci. Rep.*, 2015 doi: 10.1038/srep12840). Indeed, the hydrogen bond network is a regulator of the overall flexibility of the chromophore.

Relying on those previous studies, the observed increase in lifetime upon photoconversion can be linked to a stabilization of the chromophore, either by the formation of an additional covalent bond to the biliverdin, as we observe in the Zn^{2+} assay (Supplementary Fig. 22), and/or by a rearrangement of the hydrogen bond network.

Excellent!

Here are my remaining comments to the revised manuscript Introduction

‘These are characterized by high brightness and photostability in both states.’

Technically not correct - the pre-versions are often inferior to most non-modifiable FPs. Recent efforts have improved the photostability of the green versions by several folds (see Mohr et al., 2017, *Ang Chem Int Ed*). This deserves a reference.

We updated the sentence in the paper to enclose the evolutionary effort that has been done in the green-tored PCFPs. The updated version is:

These have been engineered to achieve high brightness, contrast of photoconversion and photostability in both states^{10,11}.

Where the citations are [10] Turkowyd et al *Angew Chem* 2017 doi: 10.1002/anie.201702870, and [11] Mohr et al *Angew Chem* 2017 doi: 10.1002/ange.201706121, both key in the recent development of the green-to-red PCFPs.

Excellent!

‘To bypass the use of highly energetic UV light in the green-to-red PCFP variants, two-photon photoconversion^{13,14} or simultaneous blue and near-infrared (NIR) light conversion¹⁵ have been used with different levels of efficiency.’

Primed conversion can be performed simultaneously as well as with a temporal delay using red (see Mohr et al., 2017, *Ang Chem In Ed*) or near-infrared (NIR) (Dempsey et al. 2015, *Nat Meth*). Hence, please correct the wording and refer also to Mohr et al., 2017, *Ang Chem In Ed* as follow:

‘To bypass the use of highly energetic UV light in the green-to-red PCFP variants, two-photon photoconversion or primed conversion (i.e. dual laser illumination of blue and red/near-infrared (NIR) light) have been used with different levels of efficiency.’

Further, in its current form, the authors’ chosen modification does not inform the interested reader that primed conversion can achieve a very high conversion efficiency (up to 80-fold, see Dempsey et al. 2015, *Nat Meth* and Mohr et al., 2017, *Ang Chem In Ed*) whereas two-photon photoconversion displays a very minor conversion efficiency (less than 2-fold, see Dempsey et al. 2015, *Nat Meth*), which renders this

methodology not a practical option. I do not insist on defining this difference, yet I encourage the authors to at least mention in their discussion that in their particular case conversion efficiencies could be much higher comparable to values achieved by primed conversion, echoing my previous comments.

Following the suggestion of the reviewer, we have modified the text to better clarify the differences in the photoconversion of green-to-red FP with the different strategies. The text is now:

To bypass the use of highly energetic UV light, two-photon photoconversion^{16,17} or primed conversion¹⁸ i.e. the combined illumination of blue and red/near-infrared (NIR) light have been used. However, the former approach suffers from low yield, while the latter still requires visible spectra illumination, relatively high red-light intensities and slow kinetics¹⁸.

We cite [18] Dempsey et al 2015 *Nat. Meth*, as a good resource for the reader to both be introduced to primed conversion and compare this strategy to alternative photoconversion wavelengths.

The reviewer acknowledges that the authors have now opted to specify in some detail the limitations of the previous approaches. The reviewer notices that the authors listed 'slow kinetics' as a limiting factor for reference [18]. The reviewer has carefully studied this work and cannot find any study that refers to a process characterised by slow kinetics. Can the authors specify what they mean, as this limitation is mentioned in the rebuttal and in the main manuscript and can be easily misinterpreted in their chosen word context.

Figures

Figure 1b - The colour yellow is barely visible. Please opt for another contrast fitting for PSmOrange.

We updated the figure accordingly, using a darker color for PSmOrange.

Excellent!

Figure 1e - The maximum green-to-red photoconversion using primed conversion is still 10-fold higher than the values reported for the most optimal procedure of miRFP photoconversion selected by the authors.

To be sincere and transparent to the interested reader, the reviewer suggests emphasising the conversion yield limitation in the discussion as mentioned above.

As explained more in detail in the first response we have followed this suggestion and expanded the Discussion accordingly to be transparent to the reader.

Figure 2b – Please include labels for immediate identification.

We updated the figure accordingly.

Thank you!

Figure 2c (right) - Why does the red channel increase – should the value not decrease, as the pool of red FP is reduced?

There is a degree of spectral bleed-through present between the two species of the investigated miRFP, meaning that in the red channel we will have a part of the signal coming from the blue-shifted species. While this bleed-through is small, in Fig. 2c (Fig. 2g in the revised manuscript) the overall intensity of the bleed-through is not taken into account, only the relative increase in signal in the different lifetime components. Since the long-lifetime component mainly represents the blue-shifted species (as can be seen from Figure 2d), we can understand that the 4-fold increase in signal in the long-lifetime component in the

red channel is caused by the small bleed-through present. Instead, the overall strong decrease of the signal in the red channel is represented by decrease of the short-lifetime component.

Thank you!

Figure 4 - To provide a convincing case for *in vivo* axial confinement authors are advised to use cytoplasmic label and photoconvert one single cell *in vivo* rather than using a nuclear label that is mosaically expressed.

The suggestion of the reviewer about the axial confinement of photoconversion is not only important for the live-cell applicability of this photoconversion, but it also gives us further insight on the photophysical mechanism behind the photoconversion at different wavelengths.

To get an answer not convoluted with any biological system used, and to compare the axial confinement at different wavelengths of photoconversion (such as 405 nm and 775 nm), we simplified the system and considered a thick pad of bacteria expressing miRFP713 (Fig. R1). We chose miRFP713 for its higher fold of conversion among the studied miRFP, but we strongly believe that the observation can be extended to all other miRFP studied. We photoconverted an area of $20 \times 20 \mu\text{m}^2$ at a depth of $15 \mu\text{m}$ in the sample, using 775 nm femtosecond pulsed light (Fig. R1a) and 405 nm continuous wave light (Fig. R1b), respectively. The energy of the 405 and 775 nm light has been kept below saturation to avoid artifacts, as characterized in the power-dependent photoconversion curves in Fig. R1d and R1e, respectively. Using the 775 nm illumination shows a clear axial confinement, that is absent with the 405 nm illumination (Fig. R1f). A similar result as that for the femtosecond 775 nm light has been found also for illumination with a

Figure R1. Optical sectioning for different photoconversion wavelengths. *E. coli* expressing miRFP713 illuminated with different photoconversion-inducing illumination sources: 775 nm (120 fs, 80 MHz) (a), 405 nm (CW) (b), and 775 nm (550 ps, 40 MHz) (c). (a–c) The photoconverted area is at 15 μm from the cover glass, and in the images the blue-shifted channel after photoconversion is reported for the xy plane at $z = 15 \mu\text{m}$ (up) and the xz projection (down). (d–e) Photoconversion yield for the blueshifted channel at increasing laser power of the photoconverting-inducing illumination at 775 nm (120 fs, 80 MHz) (d) or 405 nm (CW) (e). The vertical dotted lines specify the energy of the experiment in panels (a–b). It is important to note that these images have been acquired on a scanning system, therefore the effective energy per pixel is convoluted with the scanning. Each reported data point is the mean, and the shaded area is the SD, of the photoconversion yield for the bacteria enclosed in an area of $15 \times 15 \mu\text{m}^2$. (f) Intensity profile along the z-axis for the area enclosed in the dotted lines of panel a-to-c. All experiments have been repeated twice for each of the conditions.

picosecond 775 nm laser source (Fig. R1c). This experiment confirms the hypothesis of a non-linear process of photoconversion for the 775 nm light with respect to the linear photoconversion for the 405 nm light.

The data have been incorporated in Figure 1 (Fig. 1g) to give a better description of the mechanism and the characteristics that are expected in the application.

The non-linear process of photoconversion using a picosecond 775nm laser source – albeit not presented in an *in vivo* context - is convincing. It is though intriguing that the 405nm photoconversion is not visibly evident below the focal plane at 15 μm depth as indicated in panel f). Can the authors comment on it? Further, the authors are encouraged to also include the photoconversion yield for experimental panel (c) to ensure an objective comparison with d) and e).

Further, to rule out any photodamage the authors are advised to provide datasets where the photoconverted cells successfully undergo division. No need to do lineage tracing.

We thank the reviewer for pointing out the importance of assessing photodamage when introducing photocontrollable FPs, given the intrinsic requirement for an additional illumination step. We further assessed this in two different ways by comparing the energy used in our studies with the standards reported in the literature and by experimentally investigating photodamage for our specific application in zebrafish imaging, both showing the overall relatively low phototoxicity risks of applying the photoconversion process of miRFPs.

We used three modalities of illumination to induce photoconversion and their energy requirement are within the ranges that have been used and characterized in the literature for cell division. Waldchen et al, 2016 *Plos One*, doi: 10.1038/srep15348 extensively characterized the photoresistance of transfected cells upon irradiation at different wavelength using cell division as the assessment parameter. They characterized how the phototoxicity increases with decreasing wavelengths, identifying the limit of $\sim 50 \text{ J/cm}^2$ for 405 nm irradiation and $\sim 1 \text{ kJ/cm}^2$ when moving toward the NIR (their experimental setting stop at 640 nm). In our photoconversion studies we are within the limit defined by Waldchen et al, therefore we can expect a low impact of phototoxicity in our study with the ability of tuning the phototoxicity effect by choosing the most NIR shifted wavelengths for photoconversion. Nevertheless, careful considerations always need to be done when assessing phototoxicity, not only on the wavelength but also cell line, phase of the cell and illumination conditions.

Photoconversion light	Energy at saturation	Irradiance limit
UV/Vis continuous wave: 405 nm	$\sim 5 \text{ J/cm}^2$	$\sim 50 \text{ J/cm}^2$
NIR picosecond light: 775 nm	200 J/cm^2	$\sim 1 \text{ kJ/cm}^2$

NCOMMS-22-41203		
NIR femtosecond light: 670 – 800 nm	4 J/cm ²	-

With a pulse width of 550–700 ps, illumination typically used in STED microscopy can still be included in the consideration of Waldchen et al. Typical levels of 775 nm light for STED microscopy can reach hundreds of kJ/cm². Additionally, several considerations have previously been made on the compatibility of STED microscopy with live-cell imaging (Kilian et al. Nat Method, 2018, doi: 10.1038/s41592-0180145-5) and we believe they are out of the scope of our work. However, it is important to note that the energy of 775 nm (at 550–700 ps) required for the photoconversion of miRFPs is ~0.2 kJ/cm², while the energy of the 775 nm light required to achieve STED imaging with miRFPs is ~50 kJ/cm² (Matlashov et al 2018 doi:10.1038/s41467-019-13897-6).

The peak values reached in femtosecond pulsed laser light can be order of magnitude higher than the average value (that is the one we commonly report), therefore it is fair to compare the energy required by two-photon photoconversion with what is commonly reported in the two-photon microscopy field. In the literature, power values ranging between 1 and 50 mW are reported, and the photoconversion that we report has its peak at around 4 mW at the back aperture for a dwell time of 3.14 μs.

The measured power density used for photoconversion of the miRFPs and other photocontrollable FPs are reported in Supplementary Note 5 as a reference for the topic of phototoxicity.

More specifically for the provided application on zebrafish embryos, we decided to test whether the photoconversion at the presented power density would induce any damage to the tissue (Fig. R2). Recording a transmitted image before and after the photoconversion reveals no differences: the tissue as well as the blood flow are intact. The blood flow is visible in the confocal image as a movement-induced blurring artifact, enclosed in the dotted orange lines in Fig. R2, and also in Supplementary Fig. S20 as longitudinal stipe artifacts linked to the movement along the axis parallel to the fish orientation.

Figure R2. Photodamage assessment in zebrafish after 775 nm light photoconversion. Transmission and fluorescence images of a mosaic expression of H2B-miRFP720 in zebrafish. The power density used was 1 kJ/cm² and the photoconverted area was $\sim 15 \times 15 \mu\text{m}^2$, corresponding to a value close to saturation of the photoconversion efficiency.

The reviewer acknowledges the referenced work from the Sauer lab that has been chosen by the authors as selected benchmark, which though differs in the experimental context (in particular, *in vitro* in Waldchen et al, 2016 vs *in vivo* in this study). The Sauer lab's work emphasises that their reported results strongly suggest that examination for phototoxic effects should not be limited to phenotypic assessment immediately after imaging as the authors opted to present in Figure R2. To significantly strengthen the presented work and increase the appeal of the presented work for the *in vivo* imaging community, the reviewer strongly recommends performing a direct assessment of cell division upon photoconversion.

The reviewer thanks the authors for sharing the power calculations. While the total dose of light an experimenter puts into the cell/tissue is important, what is far more important for the well-being is the instantaneous power delivered to the cell/tissue. The authors are therefore encouraged to also include in their comparison the instantaneous peak intensity values for each modality that can faithfully reflect the diverse power values when using pulsed versus continuous wave illumination.

S2 - The colour yellow is barely visible. Please opt for another contrast fitting for PSmOrange.

We have updated the figure accordingly.

Thank you!

S8 – figure legends: Please replace ‘matains’ with ‘maintains’.

We have corrected the typo in the legend.

Thank you!

Additional comment:

Figure 3

The reviewer believes that the normalisation of achievable photoconversion yields to illumination intensity with other photoconversion modalities in b) and c) is problematic when put in direct comparison to miRFP. The reader will get the impression that the achievable conversion yields are comparable to miRFP. The authors are advised to include the photoconversion yield raw values for all cases to ensure a fair comparison (see comments for Figure R1).

Can the authors confirm that the 775nm light in 3c is based on pulsed illumination for miRFP and continuous wave illumination for primed conversion of Dendra2?

The reviewer has a hard time to understand e) based on the limited figure legend. Can the authors explain how/where the 405nm light was used.

Reviewer #2

The manuscript reports on the photo-convertibility of fluorescent proteins (FP) to shorter wavelengths upon irradiation with near-infrared light. The studied FPs are called miRFP (as in monomeric, infrared redfluorescent protein) and were originally developed a few years ago by one of the PIs (Shcherbakova et al, Nat Comm. 2016) and subsequently shown to be useful for live-cell super-resolution microscopy (Matlashov et al, Nat Comm 2020) by the two PIs of the present study.

Extending the color palette of FPs into the infrared spectral range is an important endeavor for live-cell imaging in tissue, not only to increase the number fluorescence signals that can be spectrally separated but also to exploit the fact that biological tissue can be imaged better at longer wavelengths where autofluorescence, absorption and scattering tend to be reduced.

What's new in this study is the observation that these miRFP proteins undergo a blue-shift in their spectral properties upon exposure to infrared light. This is a curious and counter-intuitive effect, which is potentially interesting, as most photo-conversions involve a red-shift to longer wavelengths.

We thank the reviewer for acknowledging the novelty of the presented photoconversion mechanism and the importance of photoactivation strategies for live-cell imaging.

In general, light-controlable spectral shifts can be a very useful feature for pulse-chase type experiments, but they may also be a bug that complicates quantitative measurements, if the fluorescence intensities or lifetimes cannot be fully accounted for.

It's a bit unclear if this may be the case here or not, because for instance the wavelength used for STED imaging by itself triggers the photo-conversion, convolving the imaging with the photo-conversion.

As underlined by the Reviewer, a fluorescent protein with photocontrollable behavior can be efficiently translated into a tool for live-cell applications only if its photophysical behavior is fully outlined and predictable.

As for this family of miRFPs and its use in STED imaging, we agree with the reviewer that the combination of STED microscopy and photoconversion can be misunderstood and requires a more in-depth explanation. Therefore, we have extended the consideration in a note (Supplementary Note 4), where we clearly explain the image formation process during STED imaging and how the photoconversion and STED are effectively sequential processes. Furthermore, a graphical explanation and energy considerations have been integrated in Fig. 5 of the main text and in the paragraph dedicated to the STED imaging in the main text.

In STED microscopy, a red-shifted donut-shaped beam (here 775 nm) is spatially overlapped to the excitation beam (here 640 nm) to suppress the fluorescence emission from the molecules when applying the beams at its periphery. Given the scanning nature of this nanoscopy technique, a point in space will experience the light of half of the spatial intensity distribution of the donut beam before being imaged/recorded (i.e., being at the center of the donut beam and therefore in the subdiffracted confined volume and allowed to emit fluorescence, Fig. R3a). If we calculate the energy delivered/accumulated to that point in space by the donut beam during the scanning, we can see that the threshold for photoconversion is exceeded long before the point is centered on the crest of the STED beam (Fig. R3b), let alone the central zero of the STED beam where it is allowed to emit. Thus, photoconversion will, with a high probability, occur long before the image formation of a specific molecule at the center of the STED beam. This is confirmed by the majority of the fluorescence signal being present in the blue-shifted channel (quantified in Fig. 5c of the main text).

Overall, it is not possible to not drive the photoconversion during a STED imaging, but by tuning the detection bandwidth the phenomenon can be accounted for or integrated in the same emission window, maintaining the possibility to quantitatively follow the signal overtime.

Figure R3. Photoconversion induced by the STED beam. (a) Representation of the STED beam at 775 nm during the scanning process of the image acquisition. A region of the sample will be exposed to all the light of one crest of the STED beam light pattern before ending up in the center of the donut where the fluorescence photons can be registered. The donut is calculated for an intensity of 775 nm light at the back aperture of 15 mW. (b) Building up of the energy level (blue line) based on the previously described scanning of the STED beam (the profile of which is reported in yellow line). The dashed grey line is the energy threshold for the photoconversion to reach saturation ($\sim 1 \text{ kW/cm}^2$).

I have these two general criticisms:

- The benefit of the blue-shifted photo-convertibility remains a bit theoretical. If the authors could demonstrate more clearly its advantages over existing FP imaging strategies, it would increase the impact of the study.

The NIR spectral window, which is specific to miRFPs, makes it a unique tool for photoconversion experiments. To follow up on the criticism of the reviewer we experimentally demonstrated how those characteristics translate into new capabilities for imaging experiments. We highlighted two specific aspects: optical sectioning and multiplexed photoconversion experiments.

- The non-linear nature of the photoconversion at >700 nm translates in optical sectioning. Both fluorescence and photoconversion of miRFPs are in the far-red-to-NIR region of the electromagnetic spectrum, where scattering and absorption for biological tissue are minimized, allowing deeper imaging penetration and relatively low phototoxicity. Many publications already report applications and development of these probes for imaging purposes *in vivo* (Matlashov et al 2020, *Nat Comm*, doi: 10.1038/s41467-019-13897-6). If we add photoconversion to the context of highly three-dimensional systems typical of *in vivo* studies, selectivity in the optical axis is desirable. The non-linearity of the photoconversion at 775 nm illumination provides such optical sectioning, similarly to two-photon imaging (Fig. R1a and R1c). In contrast, the photoconversion at 405 nm, as a linear process, does not provide any selectivity along the optical axis (Fig. R1b).

This is another point of advantage with respect to currently available PCFPs, where the photoconversion is generally linear and attempts of achieving photoconversion with two-photon processes often results in a lower photoconversion efficiency with respect to the contrast obtained with a one-photon process (Dempsey et al, *Nat Meth*, 2015 doi:10.1038/nmeth.3405). Primed conversion (combination of blue and red/near-IR light) with a double-step excitation can provide some degree of optical sectioning, however some limitations remain. First, the use of blue light does not provide the same sample accessibility compared to illumination entirely in the NIR range. Second, this combined illumination modality requires high energy (~ 10 kJ/cm²), which limits the temporal accessibility of the photoconversion due to the longer illumination needed.

- Proof-of-principle for multiplexing photoconversion strategies. In the previous version of the manuscript, we reported how the miRFP photoconversion could enrich the level of information achievable by photolabeling studies through multiplexing. A novel imaging strategy was presented to enable photoconversion of miRFPs and green-to-red GFP-like PCFPs in a compatible way. In terms of applications, we envisioned two possible scenarios, which were reported in Supplementary Fig. S19 (now Supplementary Fig. S21 and Fig. 3): spatiotemporally coupled and spatially/temporally uncoupled photoconversion of the same or multiple species of proteins. We experimentally tested that we can photoconvert two distinct proteins, LAMP1 (lysosomal-associated membrane protein 1) in lysosomes and PTS1 (peroxisomal targeting signal 1) in peroxisomes, and

Figure R4. Multiplexed photoconversion strategies for green-to-red and NIR-to-far-red PCFPs. (a) Emission spectra for Dendra2 and miRFP713 before (solid) and after (dashed) photoconversion. (b) UV/vis wavelength. The normalized photoconversion yield for miRFP713 and Dendra2 is compared over the same energy range of 405 nm light. (c) NIR wavelength. The energy dependence of the normalized photoconversion yield for miRFP at 775 nm (550 ps, 40 MHz) is compared to the primed conversion for Dendra2 at fixed 488 nm (0.13 kW/cm²) and increasing 775 nm light. The comparisons in (b, c) have been performed on a layer of *E. coli* expressing miRFP713 or Dendra2. Each data point is the mean and SD for bacteria enclosed in a field of view of $\sim 15 \times 15 \mu\text{m}^2$ (approximately 100 bacteria). (d) Spatiotemporally uncoupled photoconversion. Using photoconversion wavelengths unique for the PCFPs, like pulsed 775 nm for miRFP and low energy of 405 nm light for the green-to-red PCFPs, two different spatial locations and structures can be highlighted in the cells. (e) Spatiotemporally coupled photoconversion. The spatially localized illumination with 405 nm light at a high energy level photoconverts both PCFPs in the same area.

follow their trafficking independently at the sub-cellular level. This was only possible by independently photoconverting the PCFPs mEos and miRFP fused to PTS1 and LAMP1 respectively. Even if out of the scope of this work, we can imagine using this new capability to photo-label organelles such as mitochondria or lysosomes independently in specific compartments of polarized cells like neurons, such as the growth cone, and follow their dynamics.

To further demonstrate the benefits of the blue-shift photoconversion process we experimentally demonstrated also the spatially decoupled photoconversion strategy, investigating and comparing the energy requirements of the PCFPs to combine (Fig. R4).

In the previous version of the manuscript, we hypothesized that we could separate the photoconversion by using a selective wavelength for the photoconversion of each PCFP. In particular, 775 nm light for

miRFP720 and primed conversion (488 + >640 nm) for Dendra2. This resulted to be unfeasible since the >640 nm light needed in primed conversion was an order of magnitude higher than what was needed for the miRFP, therefore inducing both the miRFP720 and Dendra2 photoconversion simultaneously. However, an analysis of the energy dependencies of the two PCFPs at different wavelengths reveal that illumination intensity can be used as a way of selection between different PCFPs (Fig. R4b–c). Dendra2 is extremely reactive to 405 nm but less so to 775 nm (when combined to a fixed 488 nm for primed conversion), while miRFP720 has a higher reactivity to 775 nm light than Dendra2. We are aware that this behavior might be different for different green-to-red PCFPs, as reported in extensive literature on primed conversion (Turkowsky et al, *Angew Chem*, 2017; doi: 10.1002/anie.201702870), nevertheless this demonstrates that given the knowledge on the power dependency of a combination of prospective PCFPs, a selective combination of photoconversion wavelengths and illumination intensities can be defined.

We perform a proof-of-principle experiment for this recording strategy on a bacterial layer composed of a mix of *E.coli* expressing either miRFP720 or Dendra2. We could selectively photoconvert either Dendra2 or miRFP713 by using 405 nm (1.5 mJ/cm²) and 775 nm (1.4 kJ/cm², pulse width 550 nm, Fig. R4c). Alternatively, a spatiotemporally coupled photoconversion can be achieved using 775 nm (16 kJ/cm², pulse width 550 nm) with 488 nm (4 mJ/cm², Fig. R4d).

The possibility to uncouple the photoconversion can be useful for marking the same species of proteins but photolabeled at different time-points or the same protein specie located in different part of a tissue which would not be possible with only one PCFP.

- The paper is not so well written or organized, jumping straight into technical details in the Introduction and Discussion sections, whereas a broader view of the context, motivation and significance of the study would help the reader judge its merits.

We apology for the lack of clarity. We have formatted and generally overhauled the text in the new version of the manuscript to more clearly structure the context and motivation, methodology, and findings.

Reviewer #3

Report to 393188_Pennacchietti et al.

This study presents properties of five derivatives of bacterial phytochromes taylorred to minimal size, but still maintaining their capability of photoconversion and fluorescence emission. Fused proteins were generated with other proteins characteristic for various organelles and cellular compartments. The major intention on the molecular level was a variation in absorption and emission properties highlighting here the switch from the parental state to a blue-shifted photoproduct, in contrast to most GFP-derivatives that are mentioned for comparison.

Taking together the work in living cells, the application of various microscopic methods, and the data handling, all this is performed and presented with the well-known great expertise of this ensemble of authors. No critics here.

We thank the reviewer for the acknowledgement.

The treatment of the basics in phy photochemistry (abs and fluorescence), however, shows flaws and logical inconsistencies and needs thorough revision (see details further down). As a general comment, many sections of the paper can only be understood after detailed study of the SI material. To the understanding of this reviewer, SI, however, should only give details to the specialists and should be part of the main body of information.

Following up on the reviewer suggestions we have added the information to the main text of the manuscript; the reader should now be able to understand the concept, experiments and key aspects while reading only the main text. In particular, we enriched the Introduction with a more in-depth presentation of the miRFP family and the overall context of the work.

In general, the manuscript demonstrates that in many cases, also here, scholars live in a bubble of their own community. Many arguments, results, and comments cannot be understood without reading the cited literature. Acronyms and abbreviations are not explained. Nowhere in the manuscript is the origin of these miRFPs (species name?) mentioned. Also, changes by mutagenesis (amino acid positions and exchanges from/to) that yielded the generated five RFPs are not presented.

We apology for the lack of clarity in the origin of the miRFP family as well as the previous usage of the same protein family. We have now given more space to the introduction of those variants and the convention about their nomenclature.

As a summary, the family of miRFPs comes from engineering efforts toward optimized NIR fluorescent protein tags able to combine high brightness with monomeric state and therefore live-cell compatibility, ranging from tissue to sub-cellular and nanoscale imaging. The five specific variants that we have characterized in the study come specifically from three publications:

- miRFP713: Matlashov et al, *Nat Comm*, 2020, doi: 10.1038/s41467-019-13897-6.
- miRFP720: Shcherbakova et al 2018, *Nat Chem Biol*, doi: 10.1038/s41589-018-0044-1.
- miRFP670, miRFP703, miRFP709: Shcherbakova et al, *Nat Comm*, 2016, doi: 10.1038/ncomms12405.

A cartoon of the 3D crystal structure would be more than helpful. It is assumed that the naming of the compound, example ,miRFP670‘ identifies the absorption maximum. Detailed information can only be extracted from the SI-table.

We have now moved the Supplementary Table into the Main text, and we also added the 3D structure of a typical miRFP as Supplementary Figure 1.

Comments to selected paragraphs

1. 60/61

The presentation of bacterial phytochromes focussing on the here employed proteins should contain an additional information, as such that some of these bacterial proteins are so-called ,bathy‘-phys (this difference is to be mentioned and cited) highlighting the advantage that the parental state is the one absorbing at longer wavelengths. So, this phenomenon is well known in the community and not a novelty.

Bacterial phytochromes (BphP) are generally characterized by a photoswitching mechanism between a red absorbing (Pr, maximum at ~ 700 nm) and far-red/near-infrared absorbing (Pfr, maximum at ~ 750 nm)

state (Fig R5a). In a prototypical BphP, the Pr state is the dark-adapted state and the Pfr state can only be reached by photoactivation. From the Pfr state the system reverts to the Pr state either by photoactivation or by thermal relaxation. Instead, for the “bathy”-phytochromes the Pfr state is the dark-adapted state, but a similar switching mechanism between the two states is present.

Given the above definitions, the mechanism that we identify and describe in the paper does not correspond to a bathy-phytochrome: even if the spectral photoconversion has a direction that agrees with a bathyphytochrome, i.e. a blue-shift, the spectral properties and the nature of the two states differ from the reported definition. Specifically, the miRFP family of proteins have been engineered to stabilize the Pr state and hinder the Pr→Pfr transition by stabilizing D-ring positioning in the Pr state by introduced hydrogen bonds with its immediate protein backbone. This is confirmed by looking at the absorption and emission spectra of the miRFPs, which are typical of a Pr state of canonical BphPs, as well as the crystal structure of the proteins (Zhu et al 2015, *Sci Rep*, doi: 10.1038/srep12840).

In the panorama of reported photoswitching in BphPs, the photoconversion of miRFPs share some similarity with the photoswitching of RpBphP3, where the light-induced isomerization of the chromophore converts it from the Pr form to a form absorbing at shorter wavelengths (Pnr, for near-red, maximum at ~ 640 nm, Fig. R5b)(Giraud et al., *J Biol Chem*,2005, doi: 10.1074/jbc.M506890200; Yang et al., *Structure*, 2015, doi: 10.1016/j.str.2015.04.022). However, the miRFP photoconversion maintains a fundamental difference with respect to the RpBphP3, and that is the irreversibility of the photoconversion (Fig. R5c).

Fig. R5. Simplified photoswitching scheme for the majority of BphPs (a) and for the specific case of RpBphP3 (b). The circular arrows highlight how the photoisomerization of the ring D proceeds by multiple intermediates of which only the first is light-induced. The lack of extensive characterization for RpBphP3 does not allow to specify key steps of the photoswitching, but a great similarity with other BphPs is hypothesized (Giraud et al., 2005). The dashed lines go from the dark-adapted state to the photoproduct for the reported definition.

Nevertheless, this similarity allows formulating a hypothesis on the miRFP photoconversion mechanism, which could provide mutagenesis strategies for optimization of the photoconversion.

1. 63

Here, an improvement of the fluorescence is mentioned as 20-fold. This value sounds impressive, however, a principal drawback of many – including those used here – FPs of various origin is their extremely low genuine fluorescence efficiency. This holds true also in this manuscript, as an absolute parameter of the fluorescence (always shown as normalized to 1.0) is never given. Use of a well characterized reference compound would have been most welcomed.

In the section mentioned by the reviewer we were citing previous work that characterized PAiRFP (Piatkevich et al 2013, *Nat Comm*, doi: 10.1038/ncomms3153), which is an example, despite instable in the converted form, of photoactivatable mechanism in NIR FPs and a point of reference for us.

The miRFPs reported in this work are the result of previous engineering efforts, which screens for brightness and compatibility with advanced microscopy (Matlashov et al 2020, *Nat Comm*, doi: 10.1038/s41467-019-13897-6; Shcherbakova et al 2018, *Nat Chem Biol*, doi: 10.1038/s41589-018-0044-1; Shcherbakova et al 2016, *Nat Comm*, doi: 10.1038/ncomms12405). In the following table, which is part of their original publications, a detailed characterization of miRFP brightness and two comparable proteins is provided.

Monomeric NIR FPs	Parental BphP	Ex, nm	Em, nm	ϵ , M ⁻¹ cm ⁻¹	Φ , %	Molecular brightness vs miRFP720, %	Effective brightness vs HeLa cells miRFP720, %	in Ref
miRFP	BrBphP	683	705	65900	6.9	76.1	13.9	[1]
SNIFP	DrBphP	697	720	149000	2.2	54.8	n.a.	[2]
miRFP670	RpBphP1	642	670	87400	14.0	204.7	67.9	[3]
miRFP703	RpBphP1	674	703	90900	8.6	130.8	49.8	[3]
miRFP709	RpBphP1	683	709	78400	5.4	79.8	29.0	[3]
miRFP713	RpBphP2	690	713	99000	7.0	115.9	94.9	[4]
miRFP720	RpBphP2	702	720	98000	6.1	100	100	[5]

Ref: [1] Yu et al. 2015 doi: 10.1038/nmeth.3447. [2] Kamper et al 2018 doi: 10.1038/s41467-018-07246-2. [3] Shcherbakova et al 2016, doi: 10.1038/ncomms12405. [4] Matlashov et al 2020, doi: 10.1038/s41467-019-13897-6. [5] Shcherbakova et al 2018, doi: 10.1038/s41589-018-0044-1.

Additionally, in line with the previous general comment of the reviewer, we have clarified the origin of the studied miRFP proteins in the Introduction section of the manuscript.

The decision of presenting the fluorescence as normalized to one (as in Fig. 1b, 3a, and 4a of the revised manuscript) is to allow easier spectral comparison between different proteins, where the absorption and emission spectra rather than brightness are the main parameters of the reported data. For the rest of the figures, the main reported quantification is the photoconversion ratio or fold of conversion, seeing as this is the main aim of the current work.

In this context, several of the presented emission spectra are suffering from a low S/N ratio pointing to either a low expression of these proteins or a very low fluorescence efficiency.

The low signal-to-noise of the emission and excitation spectra is to be attributed to the modality of recording rather than the absolute brightness of the studied miRFPs. The spectra are recorded with the spectral detection of a microscope and not with a traditional spectrophotometer. This method unfortunately suffers from lower signal-to-noise ratio; however, it gave us the possibility to directly relate the spectral change in the sample of interest and under imaging conditions (exactly the same laser source as used in the applications).

The authors claim a 'high sensitivity to day light', yet, the quantum yield for fluorescence – as said above – is marginal. This reads like a contradiction.

The “high sensitivity to daylight” in the sentence referred to PAiRFP and not miRFP. We have now changed and expanded the Introduction and more clearly separated the two groups of proteins to avoid possible misunderstanding.

PAiRFPs are here reported as the closest example of photoactivatable fluorescent proteins engineered from BphPs that enabled bioimaging application (Piatkevich et al 2013, *Nat Comm*, doi: 10.1038/ncomms3153). The PAiRFP is a bathy phytochrome, whose photoactivation mechanism can be leveraged in photolabeling and contrast enhancement. However, this tool suffers from two main drawbacks: (1) instability of the photoactive form that slowly reverts to the dark-adapted non-fluorescent state, and (2) extreme reactivity to light, reducing the effective contrast enhancement achievable unless optimally kept in the dark.

1. 74

The rationale of this paragraph at the end of the introduction does not easily disclose to the readers as it leaves them somehow confused. In the beginning the authors emphasize the advantages of the miRFPs as being extremely red shifted advantaging over autofluorescence and other obstacles and allowing more easy multiplexing by extending the wavelength range for applications. So, why a blue shift?

We apologize for the confusion, indeed the main strength of miRFP photoconversion is that both forms, converted and not, feature absorption and emission spectra that are red shifted compared to currently available photoconvertible FPs from the GFP family (emission peak at ~ 510 nm/converted at ~ 610 nm).

The emission spectra of the ground/non-converted form peaks at ~ 700 nm, while the photoconverted form peaks at ~ 650 nm, which is indeed 50 nm blue shifted but still in the proximity of the “transparent” window, i.e., a wavelength range of the visual spectrum at ~ 650–900 nm that is relatively free of cell-inherent absorbers and therefore relevant for bioimaging of animal cells.

1. 85

The authors might add in brackets the maxima for absorption and emission (although the parameters are given in the SI material). This would make further discussion more easy, see l. 87 (ca. 53 nm blue shift: from where to where is this blue shift?)

To better clarify the spectral shift for the different miRFP variants and not confine the information in the supplementary material, we have now moved the Table to the main text.

1. 93/94

Determining the yield of conversion: this reviewer is not convinced that the ratio of fluorescence is a good measure for this photochemical process, except the authors have determined independently the fluorescence quantum efficiencies of both states (that surely are different!!). If this has been taken into account, it should be mentioned.

In choosing the assessment method for the photoconversion we have adhered to previously reported studies (see as an example Dempsey et al, *Nat. Commun.*, doi: 10.1038/nmeth.3405). For further considerations on the pitfalls and strengths of this metric we have strengthened Supplementary Note 1.

Compared to the GFP-like PCFPs, the spectral shift of miRFPs is smaller and therefore crosstalk might interfere with the estimation. This effect will specifically affect the measurement as:

- (1) A non-zero initial fluorescent signal for the blue-shifted channel, which can result in an overestimation of the yield of conversion.

- (2) An additional fluorescence signal for the red channel after photoconversion, due to the tail of the emission spectra of the converted form.

We agree with the reviewer that the ratio metric is not precisely reflecting the absolute rate of photoconversion at the single molecule level, especially because the quantum efficiencies of the different species are not taken into consideration in the calculations. However, the metric is reliable for comparative studies such as the characterization and optimization of the photoconversion process at different illumination wavelengths, illumination energy, and during thermal recovery, as in our scope.

1. 96-99

This seems to be trivial, as the conversion efficiency (if the absorption band is homogeneous) follows the absorption band intensity = oscillator strength = probability to catch a photon.

We have now changed the sentence and more clearly explained the photophysical mechanism behind the photoconversion. This explanation is now under the Result section titled: *Characterization of the miRFP NIR-to-far-red photoconversion*.

Figure 1d informs both on the non-linear nature of the photoconversion, with a quadratic dependency on the energy for every wavelength, and on the spectral window in which it is possible to drive the photoconversion. Behind the photoconversion in the NIR there is a two-photon absorption (2PA). The efficiency of 2PA is resonantly enhanced in the spectral region of 710–810 nm due to the nearby one-photon transition of the Q-band, an effect which is widely reported for most tetrapyrrole (Drobizhev et al, *Chem Phys Lett*, 2002, doi:10.1016/S0009-2614(02)00206-3; Drobizhev et al. 2011, *Nat Meth*, doi: 10.1038/nmeth.1596). Therefore, the behavior of the dependency on the energy at each wavelength is the result of the balance between multiple possible transitions, each of which will have a specific efficiency in different spectral windows. In the energy diagram reported below we summarize some of the transitions that play a role in the NIR region.

Pathways that promote the photoconversion will be: one photon absorption combined with the excited state absorption (ESA) and 2PA. Pathways that will compete with the photoconversion will be: stimulated emission and, since we are investigating an immobilized layer of bacteria, bleaching. The balance between the two will define the maximum yield of photoconversion reachable at the different wavelengths of illumination and the speed of the process.

We have now updated Fig. 1c (and more in detail in Supplementary Fig. 6c) to reflect this complexity and clarified the description of the mechanism in the main text.

1. 117

This paragraph outlining the spectral properties of the Soret is surely correct, however, own experiences in phy spectroscopy have told that this band is of low use for activation phy proteins for (i) a lower efficiency (correctly cited), and (ii) the shift of the Soret band upon conversion is so small that selective excitation seems to be impossible.

Further, the absorption is placed in the region of autofluorescence caused by flavins, if the experiment goes to living cells. The authors might consider to minimize this paragraph in length. This aspect is picked up again in l. 134. So, it might be worth to present the spectra (abs, fluo) for the Soret band.

The use of the Soret band for excitation only discloses later when the same wavelength is used for two different photolabels with same excitation wavelength range but differently tagged proteins.

The investigation in the UV/vis range is important for two main reasons:

- The complete characterization of the NIR-to-far-red photoconversion. The side-by-side analysis of the photoconversion in the NIR and UV/vis shows that the photoconversion results from an excited state of the miRFP proteins that can be reached either by one photon absorption at 405 nm or by two-photon absorption in the 700 nm range.
- The applicability in imaging. UV/vis is a very common wavelength range both for use with photoactivatable FPs and practically present in commercially available microscopes. Hence, unveiling the energy dependence of the NIR-to-far-red photoconversion at 405 nm illumination is fundamental for any multiplexing strategy of photoconversion and for providing a fair comparison between PCFPs. On the other hand, the possibility to use this wavelength, despite being less compatible with certain live-cell applications, broadens the applicability of the probe and means that it can be implemented in different microscopy architectures.

The photoproduct reached by illumination in the two spectral regions is the same, therefore in the study we have used both in the live-cell experiments.

1. 207

The advantage of not damaging DNA by 700 nm light is apparent and trivial. DNA absorbs at 260 nm (maximum), and even excitation of the Soret band around 400 nm would not be harmful (if wavelength selection would be performed properly).

In the study of this NIR-to-far-red photoconversion we have explored a big range of energies and wavelengths. The cited studies investigate and discuss the problem of light induced phototoxicity when high energies are used in non-linear imaging, such as two-photon fluorescence microscopy. In the paper of Post et al, *FEBS Letter*, 2005, doi: 10.1016/j.febslet.2004.11.092 the authors specifically investigate how the different modality of photoconversion of PAGFP (both 1-photon and 2-photon) in *Drosophila* embryo would have influenced the biological system, by tracking nuclear viability. They conclude that only a shift to 2-photon photoconversion at less than 1500 GW/cm² for 820 nm could maintain the viability in whole mount embryos. Our illumination is around 1–2 MW/cm², therefore well below the limit of viability identified by the authors.

Considering that there is a femtosecond pulsed laser of the same kind used in two-photon fluorescence microscopy among the different laser sources we are using, we aimed to provide a reference to access photodamage.

Furthermore, we included a direct comparison of the photoconversion energy required by miRFP in respect to state of art photoconversion mechanism of PCFP, which can be used to assess photodamage and phototoxicity. We have summarized this in the Supplementary Note 5.

1. 210

This paragraph required several-fold reading to understand the message. Also, as at first glance some of the arguments appear to contradict generally accepted phy results.

This paragraph is in the Discussion part of the manuscript and obviously should be considered as the discussion of possible/hypothetic mechanism(s) of the blue-shifted (hypsochromic) miRFP photoconversion. To elucidate exact mechanism of the photoconversion requires crystallization and determination of structure of the chromophore and its immediate environment of the photoconverted protein species, as well as rigorous mass-spectrometry investigation of the isolated photoconverted chromophore. This is a separate large project, which is beyond the scope of this paper that reports, for the first time, the unique phenomenon of the NIR-induced fluorescence hypsochromic shift in bacteriophytochrome-derived NIR FPs.

There are several aspects to address:

(i) why does the excitation not induce isomerization of the D-ring? If mentioned in the text above, the reason for this statement is not clearly outlined.

Because in miRFPs the Pr state is stabilized by engineered hydrogen bonds with the chromophore's ring D and side chains of the mutated residues in its immediate environment.

(ii) Where is the result that neighboring residues move in response to the ring movements? There is literature to this conformational change from other phys, but here, is there a crystal structure or mutagenesis experimental evidence (for second bond formation)? In addition, how is the 'transient bond formation' concluded?

Possible movement of the residues around the chromophore, as well as transient bond formation (e.g., hydrogen bonds) between the chromophore and its environment, in response to the photoconverting light are just the hypotheses that are proposed in the Discussion part of the manuscript.

(iii) Why shows the denatured protein a higher mobility in SDS-PAGE? Proteins under these conditions are denatured and adopt a spheric, oblong-oval conformation that not necessarily is disturbed by an additional bond (l. 219).

The increase of the electrophoretic mobility on SDS-PAGE gels of denatured miRFPs, resulting from the formation of additional covalent bond between the biliverdin chromophore and protein polypeptide, have been shown by Verkhusha and Lagarias groups in the past (e.g., Baloban et al, *Chemical Science*, 2017, doi: 10.1039/c7sc00855d)

(iv) 'Chromophore protonation': Detailed studies have shown that phys carry the chromophore in the protonated state in both states parental and photoproduct. This feature has been intensely discussed in the phy literature, and also deprotonation/reprotonation is known, e.g., meta-R II intermediate in plant phys that still carry the chromophore protonated in both states. Also deprotonation in few CBCRs have been reported, still with intense discussion in the community. Thus, the comment that the chromophore is protonated seems to be in agreement with common knowledge and should not be taken as a special exception. Also, the authors should be aware that bilin protonation in general causes a BATHOCHROMIC shift. If the authors wish to keep the protonation argument they should make this more clear by a detailed discussion.

We do not make an exception about possible protonation of the protoconverted chromophore in miRFPs. We have revised this place in the text.

Indeed, photoconversion to Pfr state, accompanied by chromophore protonation, induces the bathochromic shift in absorbance. However, in miRFPs the chromophore is stabilized in the Pr state, and its protonation may cause its hypsochromic shift observed experimentally in this paper. Notably, in another type of FPs, such as multicolor FPs of the GFP-like protein family, the chromophore protonation results in the hypsochromic shifts (e.g., Shcherbakova al, *Current Opinion in Chemical Biology*, 2014, doi: 10.1016/j.cbpa.2014.04.010).

l. 373/4

The authors record here fluorescence. Thus ,fluorescence‘ should be added: ... and fluorescence was recorded in ...

We have corrected the text accordingly.

l. 375, change tense (characterize): ... to test and to characterize ... The text has been updated.

l. 384, add comma: .. dependency, and lifetime The text has been updated.

l. 386-388: this sounds confusing, as ,line average was tuned to reach different ..., EXCEPT (highlighted by reviewer) for ... 405 nm light, the others were set to a 405 nm light intensity of‘ So, what (which wavelength) was kept and which was tuned?

We have now divided the sentence into two separate ones. The reported value 4.8 J/cm^2 is the energy of 405 nm illumination used in the experiments where a fixed-power 405 nm illumination was used. These are the experiments in which we wanted to not see the energy dependence of the photoconversion process, but rather other characteristics like the change in lifetime, application in live cells, oxygen dependence and thermal recovery.

l. 391: ,... accounts for ... of 3..‘ 3 what?

We completed the sentence to report the energy of photoconversion.

l. 403 and following

This reads somehow confusing. The protein is denatured and this reviewer assumes it is no longer soluble, but will be found in the precipitate. If then the ,debris‘ is removed, also denatured protein might be lost. The authors should clarify this point.

Low-speed centrifugation (~1000 g) for a limited 5 min time was used to remove just extremely large protein aggregates and perhaps remaining after Ni-NTA purification agarose beads to avoid their sticking at the beginning of the running lanes of the SDS-PAGE.

Supplementary material

Table S1

It is assumed that ‘excitation’ maxima are absorption maxima, and should better be named as such.

The values reported in the table are in fact excitation maxima, not absorption maxima, as the recorded spectra are excitation spectra.

Notably, the lambda values in absorption and emission (Stokes shifts) become smaller the longer the absorption/emission wavelength are. This diminishes to some extent the proposed advantage of the novel compounds. Further, as this manuscript presents far red-shifted compounds the information on changes in the Soret band is incomplete, but is not helpful in a discussion of potential applications, the more so, as the nm-shift of the Soret bands in absorption and emission is marginally small and do practically do not allow selective excitation and detection.

Supplementary Table 1 (now moved in the main text) summarizes the spectral shift induced by the photoconversion for each of the miRFP variants that we have studied at the different photoconversion wavelengths, both in the Soret band and in the Q band. We believe that the label of the table were misleading so we change them to be more adherent to what they report on. The new labels are: “PC yield for $\lambda^{\text{PC, Q band}}$ ” and “PC yield for $\lambda^{\text{PC, S band}}$ ”.

For imaging purposes, the more attractive variants are the more red-shifted ones, and indeed most of the bioimaging examples we have reported are either with the miRFP720 or miRFP713 variants. From the previous characterization of the miRFP family as NIR FP tags, there is an additional parameter beside the emission wavelength, that makes miRFP720 and miRFP713 more attractive for imaging in live-cell applications: the higher effective brightness (molecular and cellular brightness) with respect to the other variants (see table reported in the previous comments or the original source, Matlashov et al 2020, *Nat Comm*, doi: 10.1038/s41467-019-13897-6).

Nevertheless, it is important to report the extent of this phenomenon also for the other variants, identifying the NIR-to-far-red PC as a common mechanism in the miRFP family.

We would like to underline that the photoconversion we observed is not reversible like the Pr/Pfr switch of canonical BphPs, therefore we do not expect to use an eventual shift in the Soret band as a way to selectively promote a forward and backward reaction.

Comment to note and fig N1. The authors show examples highlighting the principle for cases of large A/E shifts. Correct, however, they use as example (panel a) an absorption of 500 +/- nm that is not given for any of their compounds.

We believe that the NIR-to-far-red photoconversion mechanism that we introduce in the study differs, in many photochemical characteristics, to the one currently reported among fluorescent proteins. Therefore, the note specifically compares: GFP-like PCFPs with a red-shift (previously reported) and NIR PCFPs with a blue-shift (introduced here). In the figure we take as an example the green-to-red photoconversion of mEos3.2 (a) and the NIR-to-far-red photoconversion of miRFP720 (b). This is the reason why the wavelengths of panel a in Fig. N1 are in the 500 nm region.

Further, both cases red- or blue shifts of the photoproduct does not escape the dilemma: panel a (for a large shift of 100 nm!) still has contributions of exciting also the photoproduct, and even if this can be minimized, there might appear a FRET process falsifying the result. This is even worse in panel b, where excitation of

the parental state generates a significant amount of photoproduct, again causing false positive results. The authors address this point in their note, but clearly, precise manipulation of the data is required and may bear difficulties.

The FRET process has a range of action between 1–10 nm. Hence, both for monomeric fluorescent proteins freely diffusing in the cytoplasm of bacteria and labelled to structures such as histone, vimentin or lysosomes this effect can be disregarded as the common distance between FPs is larger.

Fig N2. Why do the authors excite at 405 nm? Apparently, this is the Soret band, but why not escaping autofluorescence of the cells and using a longer wavelength?

It is important to note that 405 nm is only used as a photoconversion wavelength and not for fluorescence excitation. The photons emitted during 405 nm illumination are not recorded, which free the experiment from any potential 405 nm induced autofluorescence.

Figure R6. Comparison of lifetime and spectral changes between photoconversion triggered by 405 nm (a) and 775 nm (b). The photoconversion is reported both spectrally (middle column) and in fluorescence lifetime, through the phasor plot obtain by summing up the multiple spectral windows between 640 and 740 nm. The photoconversion is done on *E. coli* expressing miRFP720.

We also characterized the fluorescence lifetime response of the photoconversion. We have compared the spectral and fluorescence lifetime characteristics upon 405 nm and 775 nm photoconversion and have found comparable behavior. As such, since the equivalency of the two wavelengths to induce the photoconversion is in this case ensured, the result is independent from it.

Reviewer #4

The paper is interesting and innovative, but has also several weaknesses as will be pointed out below. The major concerns with the manuscript are that

- 1) The main aim of the paper is unclear.
- 2) Statistics and information about reproducibility are lacking
- 3) Many of the conclusions drawn are discussed at a superficial level

Key results

The manuscript contains novel concepts. Particularly the approach of employing photoconversion to enable intracellular diffusion studies in combination with STED; however, the key finding is not clearly defined. Is it the discovery of the blue shift photoconversion of the biliverdin chromophore using NIR excitation that is the primary results, or is it the technical realization of a sophisticated approach combining photoconversion in a more general sense? Based on the title the blue-shifted photoconversion using NIR is highlighted as the discovery, but as the manuscript also presents data in which 405 nm excitation is utilized for photoconversion, as well as another fluorophore which is green-to-red shifted (rather than NIR-to-farred shifted), this is confusing. Furthermore, a mechanistic understanding of the “discovery” of blue shift in the specific fluorophore is not adequately substantiated, which is a weakness.

Following the reviewers’ suggestions, we have now reorganized the text to achieve a clearer structure and better deliver the presented findings.

At the core of the paper is the discovery of the novel photoconversion mechanism in miRFPs. This directly translates to novel live-cell applications in the NIR spectral window, which not only complements the current available PCFPs tools of the GFP-family but add new spatially and temporally multiplexed possibilities.

Additionally, the non-linear nature of the process is pivotal for accurate spatial confinement, especially axially, where it can be used to select precisely single layer of cells or even specific compartment at the sub-cellular level.

From this consideration, the proof-of-principle of the photoconversion multiplexing emerges, where this novel photoconversion mechanism can be combined with that of green-to-red PCFPs to enable multiple photolabeling experiments.

We indeed added the data about 405 nm photoconversion since it is the standard in current tools and therefore important for direct and fair comparisons. Additionally, it enables multiplexed photoconversion experiments, which can be useful for specific applications yet easy to implement. In fact, the 405 nm is common in commercial setups and typically available in imaging facilities. Therefore, having the possibility to test the photoconversion in any microscope is to our view a positive aspect.

Validity

Although the data presented in each figure is analyzed and interpreted thoroughly, the overall robustness and validity of the approach is difficult to assess since statistics and reproducibility is not commented. This is a general weakness throughout the manuscript.

We apology for the lack of clarity on the statistics and reproducibility, we do believe this aspect to be of fundamental importance and we have improved it in the following ways:

- 1) All the experimental parameters for each figure as well as the data are collected in Supplementary Table 1.

- 2) We have adequately commented on the statistical information that was missing in the legend of each figure as well as in the *Statistics and Reproducibility* statement in the Material and Methods section.

The photoconversion mechanism has been investigated for different illumination conditions (wavelengths, pulse widths, repetition rates) showing the same behaviour in two biological systems, both bacteria and eukaryotic cells (Fig. 1–2, Supplementary Fig. S2–S14). For each imaging strategy we report in the paper at least two representative examples, one in the Main Text (Fig. 4, multiplexing; Fig. 5, combination with STED imaging; and Fig. 6, *in vivo* imaging) and one in the Supplementary Material (Supplementary Fig. S16, multiplexing; Supplementary Fig. S18, combination with STED imaging; and Supplementary Fig. S20, *in vivo* imaging).

Significance

As pointed out above, the approach employing photoconversion to enable intracellular diffusion studies in combination with STED is of interest for the biophotonics community, as well as the possibility to combine the approach of photoconversion at multiple wavelengths enabling multiplexing; however this is not pointed out and supported as the main finding. Instead, several different sub-goals are highlighted as the main conclusions. Many of these are demonstrated as proof-of-concept, and discussed briefly as will be more specifically criticized below.

Thus it is advised that the manuscript is revised to limit the scope and thereby be more clear about the main findings and/or discovery. The revision should preferably include statistical analyses and elaborate on the theoretical anchoring to support the conclusions.

We appreciate the critical feedback and we re-structure the text highlighting the main claims.

A detailed answer is provided to the comment below.

Critical review of the conclusions drawn

Report on a NIR-to-far-red photoconversion phenomenon.

The manuscript contains data acquired from *E.coli* bacteria expressing miRFP709 and analogue mutants supporting this claim; however, the underlying mechanism being the phenomenon is not discussed at a satisfactory detail.

Neither of the mutants demonstrate Q-band emission (and therefor also unlikely absorption) of NIR at 775 nm, which is the excitation wavelength chosen for the photoconversion. It might be speculated that at this wavelength using 120 fs pulsed laser light at quite high powers (4 MW/cm²) non-linear excitation routes (targeting the Soret-band?) might come into play; however, nothing is stated about this. Instead, it is vaguely stated that “Likely, the blue-shift of the photoconverted form is caused by disturbance of the π -conjugated system in the BV chromophore”. If, for example, two-photon excitation is expected to perturb the π conjugated system, this should be further supported and related to the Soret-band absorption. Furthermore, it would have been appropriate to include two-photon excitation and cross sections data to support the mechanistic claims. In fact, the term two-photon excitation is just briefly mentioned in the discussion section and not explained in the context of data, which is a weakness.

We apology for the lack of clarity in the explanation of the photophysical mechanism behind miRFP photoconversion. In the revised manuscript we added new experiments and explanations to characterize the phenomenon and support the claim of its non-linear nature.

The experimental evidence collected suggests that the photoconversion proceeds from an excited state that can be reached either by irradiation in the Soret band (i.e. 405 nm), or at wavelengths > 670 nm (including 775 nm).

In the Soret band the photoconversion is linear suggesting a one-photon absorption (1PA). For illumination at wavelengths > 670 nm photoconversion is non-linear, specifically quadratic with the light intensity. The photoconversion is therefore the result of a two-photon absorption (2PA). The efficiency of 2PA is resonantly enhanced in the spectral region of 710 – 810 nm due to the nearby one-photon transition of the Q-band, effect widely reported for most tetrapyrrole (Drobizhev et al, *Chem Phys Lett*, 2002, doi:10.1016/S0009-2614(02)00206-3; Drobizhev et al. 2011, *Nat Meth*, doi: 10.1038/nmeth.1596). Absolute 2PA cross-section have also been measured for iRFP proteins, miRFP ancestors, confirming a similar enhanced 2PA in the Soret region, in this study identified in the 890 – 950 nm spectral region (Piatkevich et al, *Biophysical Journal*, 2017, doi: 10.1016/j.bpj.2017.09.007).

When measuring the action spectra of the photoconversion we report on the final product induced by the 2PA. Therefore, the peak position will be strongly influenced by the Q-transition, being blue-shifted with respect to the pure 2PA cross-section because of the resonance enhancement. The complexity of this interaction is reflected in the different efficiency and power dependencies reported in Fig. 1d at different wavelengths of excitation moving from a spectral region of maximal overlap with the Q-band to the more NIR wavelengths. Respect to the simplified diagram reported in Fig. 1c, more processes need to be added in the energy diagram (we report a more complete energy diagram below as well as in Supplementary Fig. 6c). In particular when on the 1PA region of the Q-band excited state absorption represent a positive drive for the photoconversion, although the possible saturation of the S₁(Q) define the competition of stimulated emission and bleaching that push down the photoconversion yield. Moving above 720 nm the process is mainly driven by 2PA, probably with lower efficiency (lower absolute photoconversion yield) but less affected by bleaching that now come only from the S₂ state (slower decrease of the photoconversion yield).

In the NIR we have also explored different illumination modalities, namely femtosecond and picosecond lasers. Considering the 775 nm wavelength, with a pulse width of 120 fs saturation of the photoconversion is reached at 4 J/cm², while at 550 ps the mean energy needed is of 200 J/cm². The factor of difference between these two illuminations is of ~ 50 times, in line with what expected to achieve the same number of absorbed photons per fluorophore per pulse, n_{aa} , that we assumed can trigger the photoconversion.

To formally compare the two excitations, we can express the number of absorbed photon per fluorophore per pulse as:

$$n_{aa} = \frac{ppaaaaaa2 \delta\delta2PPPP NNNN2}{\tau\tau_{pp} f_{pp} 2\hbar cccc} \quad 2$$

where $\delta\delta_{2PPP}$ is the 2PA cross-section, $\tau\tau_{pp}$ the pulse width, ff_{pp} the pulse repetition rate, NNN the numerical aperture of the objective, cc the excitation wavelength, \hbar the reduced Planck constant, cc the speed of light (Denk et al, 1990, *Science*, doi: 10.1126/science.2321027). Comparing the same fluorescent protein, miRFP720, in microscope of equal numerical aperture (NA = 1.4), the expected ratio between the powers to achieve the same number of absorbed photons is simply linked to the characteristics of the excitation:

$$\frac{pp_{aaaaaa,1} ff_{pp,1} \tau\tau_{pp,1}}{pp_{aaaaaa,2} ff_{pp,2} \tau\tau_{pp,2}}$$

that would give a theoretical ratio of ~34, in line with the experimental value.

In the discussion section (line 210-228), there is a paragraph more focused on the mechanistic understanding; however, it is difficult to connect this paragraph to the actual data presented. The paragraph brings up quite specific information about the protein structure, without connection to actual data. Bringing back to what is pointed out above, the key question still concerns how the excitation using NIR can induce this photoconversion effect. Thus it is advised that the mechanistic discussion is merged and integrated with the data commentary section.

To elucidate a mechanism of the photoconversion requires crystallization and determination of structure of the chromophore and its immediate environment of the photoconverted protein species, as well as rigorous mass-spectrometry investigation of the isolated photoconverted chromophore. This is a separate large project, which is beyond the scope of this paper that reports, for the first time, the unique phenomenon of the NIR-induced fluorescence hypsochromic shift in bacteriophytochrome-derived NIR FPs.

Here, in the Discussion part, we can just reasonably hypothesize about possible photoconversion mechanism(s), based on the limited biochemical analysis (e.g., Suppl Figure 22; new numbering) and on the published data on the chromophore behavior in bacteriophytochrome-derived NIR FPs. Going in more details in our hypothesis will result in pure speculation, which is not appropriate for the research paper.

Anyway, we have followed the suggestion of the Reviewer to separate the photophysical considerations about the photoconversion phenomenon and the possible/hypothetical molecular mechanism(s) of photoconversion itself.

Another confusion is that both NIR induced as well as 405 nm induced photoconversion are explored in the paper, but their presentations are not clearly separated and explained. For example, the photoconversion taking place in HeLa cells (Note in materials and methods cells are stated to be U2OS?) seem to be restricted to 405 nm induction. Also the reported lifetime shift from the photoconversion seem to rely on 405 nm irradiation. This mixing up and lack of systematic analysis is confusing to the reader and needs to be clarified.

Ultimately, irradiation with 405 nm and >700 nm give rise to the same photoproduct (Fig. R6). This is confirmed both by the spectral and lifetime profiles recorded after photoconversion-inducing illumination with the two wavelengths.

This is the reason why in the applications we have used the two wavelengths equally, choosing the one that matches the strategy of photoconversion that we wanted to explore. We understand that this have introduced confusion in the text, therefore we have (1) collected all the acquisition and sample parameters behind the images and data presented in the figures in Supplementary Table 1, (2) separated the two thematic blocks: photophysical description of the photoconversion process and applications of the photoconversion, and (3) clarified the impact of the chosen photoconversion-inducing wavelength and illumination source for the

generation of the final photoproduct by comparing side-by-side the spectral and fluorescence lifetime behavior in Supplementary Fig. 6d-e.

The observed photoconversion is further claimed based on the observation of a shift in fluorescence lifetime. Data supporting the observation of increase of fluorescence lifetime is presented for photoconversion using 405 nm excitation (and not supported for the NIR induced photoconversion?). It is unclear for what reason the lifetime measurements are included. Is it to support the understanding of the underlying mechanism, or more as technical means to improve imaging contrast? Stated in the manuscript is a sentence “The photoconversion introduces an unbalance in the protein population by shifting it towards the blue shifted form with a longer lifetime”. What is the rationale for drawing this conclusion, and what is meant by “unbalance in protein population”? Instead it would be interesting to elaborate the discussion about the origin of the observed shift in lifetime from a more fundamental level. Does the longer lifetime relate to a shift of the natural lifetime of the fluorophore due to the hypsochromic shift, or can it be speculated that the non-radiative decay routes of the chromophore-protein complex is affected?

Fluorescent lifetime is an important parameter that both reports on the photoconversion mechanism and provides an additional dimension that can be used for imaging applications, for example to separate with higher accuracy than the sole spectral shift, the converted and not-converted form. We have restructured the text to better divide the considerations that belong to one or the other motivation in the Result section that discusses the fluorescent lifetime recordings. Here we want to clarify those findings/considerations:

- The fluorescence lifetime measurements reveal an initial heterogeneity of the miRFPs, before photoconversion. Indeed, the time-domain can provide insight into heterogeneity that cannot be achieved through conventional steady-state fluorescence measurements, which can only provide an average view. In particular, spectrally resolved fluorescence lifetime imaging enables exploration of the wavelength-dependence of the relative contribution of two or multiple fluorophore/species. In the phasor plot, the mix of two components results in the fluorescence lifetime moving along a straight line. At the extremes of this line are the conditions of maximal contribution of the species that is predominant in that wavelength window, while in the in-between spectral region the lifetime will be the linear mix of them (Hanley Q. S., *J. R. Soc. Interface*, 2009, doi: 10.1098/rsif.2008.0393.focus).

Overall, the mix observed in fluorescence lifetime measurements reveals a degree of ground state heterogeneity, where at least two different conformations are coexisting at equilibrium. The photoconverted species has lifetime and spectral characteristics similar to one of the two populations, however it is impossible from just this parameter to know if the photoconverted form is the same form already present at thermal equilibrium (Tang et al., *Chem Rev*, 2021, doi: 10.1021/acs.chemrev.1c00194).

Further studies correlating the observation of fluorescence lifetime with ultrafast spectroscopy will allow to clarify the detail of this heterogeneity.

The sentence highlighted by the reviewer is grounded in this context. We wanted to highlight how before photoconversion there is a mix of two populations of the miRFP: one with a faster lifetime and one with a slower lifetime. After photoconversion the slower lifetime population is predominant. We have made this sentence more explicit and added references that allow the reader to better follow the reasoning of our analysis.

This spectral dependency of the unconverted form suggests an initial heterogeneity of miRFP720 in the equilibrium state that converges in the slower form upon conversion³⁶ (Fig. 2c and Supplementary Note 2).

[36] Hanley Q. S., *J. R. Soc. Interface*, 2009, doi: 10.1098/rsif.2008.0393.focus

- Knowing and characterizing the fluorescence lifetime upon photoconversion is of invaluable help when fluorescence intensity is not a fully reliable readout for imaging. Such a condition is common

for *in vivo* imaging, where the detected fluorescence intensities can be low and/or highly contested with a strong background. Furthermore, as we demonstrate in Fig. 2d-g, the knowledge of the fluorescence lifetime can be used to further enhance the contrast of the imaging after photoconversion, since the isolation of the two components allows to correct for bleed-through between the two signals.

The “prolonged stability” is promoted in the discussion as an advancement; however, the present data is not discussed in the context of state-of-the-art in the field. What is the expected stability of the photoconversion technologies known today? This needs to be stated in order to support the statement.

The “prolonged stability” refers to the thermal stability of the photoconversion, where we see no return to the original near-IR form over an extended period of 12 hours (Supplementary Fig. 9). We have modified the term in the text accordingly for clarity.

The need for a close monitoring of this parameter is linked to the photoactive nature of bacteriophytochromes, where the Pr/Pfr switching can also proceed thermally with a variety of recovery rates (Fig. R5). Two major examples of Bphys that are meaningful as comparisons of the identified blueshift of miRFPs are: the far-red-light photoactivatable near-IR PAiRFP, and RpBphP3 for which a Pr→Pnr hypsochromic shift has also been reported.

PAiRFP1 and PAiRFP2 are bacteriophytochrome-based photoactivatable FPs, engineered from a bathy BphP from *Agrobacterium tumefaciens* C58 called AtBphP2. As a bathy BphP, at thermal equilibrium the protein is in the Pfr form, where no significant fluorescence is recorded. After irradiation with 660 nm the fluorescent Pr state is reached. The photoactivatable behavior of PAiRFP1 and PAiRFP2 has allowed not only *in vivo* photolabeling, but also an enhancement in the signal-to-noise ratio for *in vivo* imaging. However, for these proteins, the photoactivation does not involve any chemical modification of the chromophore (except for protonation/deprotonation and isomerization) and thus the dark relaxation brings the system back with a half time of ~1 h (PAiRFP1) and ~4 h (PAiRFP2). This reset of the system imposes limitations on the applications compatible with these PAFPs.

Instead for RpBphP3, the Pr→Pnr hypsochromic shift has a high degree of similarity with the photoconversion that we have reported for what concerns the spectral characteristics. Both proteins start from a state excitable around 670 nm, identified as the Pr state, to then photoconvert upon irradiation with ~700 nm light toward the blue-side of the spectrum, ~ 650 nm. The thermic stability is exactly what distinguishes the photoconversion of miRFPs and RpBphP3. In the latter case the process is indeed reversible and Giraud et al have demonstrated that the system can be brought back to the Pr state either by illumination with 645 nm or by dark relaxation (Giraud et al., *J Biol Chem*, 2005, doi: 10.1074/jbc.M506890200; Yang et al., *Structure*, 2015, doi: 10.1016/j.str.2015.04.022). For miRFPs, we have observed no return to the Pr state neither by illumination nor dark relaxation, aspects that suggest that a different molecular mechanism is behind the here investigated photoconversion.

High spatiotemporal precision

It is demonstrated that the concept of photoconversion of the miRFP is compatible with STED microscopy, known to have superior spatial precision compared to e.g. confocal microscopy. However, it is unclear what is referred to in the context of temporal precision? This should be clarified, or the claim removed.

With temporal precision we wanted to highlight the compatibility of the photoconversion mechanism with the imaging time dictated by the microscopy technique under use, i.e. STED nanoscopy. More in general, the energy needed to photoconvert miRFPs to saturation (therefore at a good signal-to-noise ratio in the blue-shifted channel) is 0.2 kJ/cm² and 5 J/cm² for 775 and 405 nm illumination respectively. These energies

can be delivered in a single point in time or space, where we can compress the dwell time down to a minimum, i.e. $>3 \mu\text{s}$. In the combination with STED imaging, where also the spatial shape of the beam plays an important role, this means that the photoconversion can be driven during the recording of the STED image itself, utilizing the depletion beam as a simultaneous photoconversion beam. We have performed calculations of the energy relationship between photoconversion and STED image acquisition in Supplementary Note 4 and also added the key aspects of this in Fig 5a–b.

More in general, the energy threshold for an efficient photoconversion is an important parameter for the applicability of photoconversion, especially in live-cell studies. The energy requirement for photoconversion translates in a minimum time of applied photoconversion-inducing illumination, and when the energy requirement is high the photoconversion can impose a rate-limiting step to the accessible dynamics.

Study dynamic processes in living cells

Photoconversion is reported to be demonstrated in mammalian cells. The manuscript includes 3-5 images from single HeLa(or U2OS?) cells, using fusion to different cellular proteins; however, statistics are missing. In order to claim successful photoconversion in mammalian cells, it needs to be stated what cell types the claim is restricted to and something about the success rate of the method. Was it a one time experiment, or could it be repeated?

We have now reported the specification on the statistical analysis in a dedicated paragraph of the material and method section as well as in the figure legend of the presented data. To enhance reproducibility and allow an easier comparison of the experimental conditions for each experiment in Supplementary Table 1 we collected and specified the cell type used for each example together with the image recording parameters.

For what concerns the success rate of the method, it is important to notice that for representative images in which we wanted to demonstrate the photoconversion, either in combination with other photoconvertible probes (Fig. 4 of the main text) or for *in vivo* application (fig. 5 and 6 of the main text), we have used the energy level identified by the calibration of photoconversion, a calibration performed either on bacteria or directly in mammalian cells on a day-to-day basis to have control over the fluctuation of the laser power and the different image acquisition settings (i.e. most importantly dwell time and pixel size). Using these calibrated energy levels, the photoconversion is ensured to take place with certainty. For the live cell imaging data, the reported experiments out of which the images are presented are representative of at least triplicates.

Data from a single multiplexing photolabeling experiment is demonstrated as proof-of-concept in living cells (Figure 3, cell type not stated in legend). It is claimed that “using this method, subsets of lysosomes and peroxisomes targeted with miRFP720 and mEos2 respectively were photolabeled and tracked over time”. Statistics are lacking. Furthermore, the biological implications of the findings in the experiments are not discussed. Is the finding realistic/unexpected? Is the method limited to this cell type, this organelle? What alternative methods is this approach complementary to?

Optical labeling provides spatial and temporal information through site-specific photoconversion, being it at the subcellular level or in a whole organism or tissue. Expanding this labeling strategy to a second optical label would expand its dimensionality, enabling the correlation in time or in space of multiple proteins or cells of interest.

The miRFP family, especially in its more red-shifted variants (miRFP713 and miRFP720), represents a unique complementary label for photoconversion multiplexing, with the possibility to be paired with the

most numerous groups of PCFPs, the green-to-red PCFPs. Indeed, the GFP-like proteins are mostly spectrally overlapping. The blue-shift reported and engineered for dyes (addressed in some publications as “photoblueing”, Helmerich et al, *Nat Meth*, 2021, doi:10.1038/s41592-021-01061-2) have some point of overlap with the miRFP proteins, namely being in the NIR part of the spectrum and efficiently photoconvertible with two-photon excitation. Nevertheless, the photoconversion that has been reported for the NIR dyes is of greater magnitude respect to the miRFP variants, bringing the photoconverted specie to emit in the 500 – 600 nm region, therefore severely overlapping with the red-shifted form of the green-to-red PCFPs (Kwon et al, *Adv Sci*, 2016 doi:10.1038/srep23866 and Saladin et al, *Angew Chem Int Ed*, 2023 doi:10.1002/anie.202215085).

We have now commented on the statistical repetitions for the reported experiments. In Fig. 4 (previously fig. 2) is reported a representative image of the spatially coupled photoconversion multiplexing and another example is presented in Supplementary Fig. 16 (previously Supplementary Fig. 14). The success of the photoconversion, once we set the energy at the calibrated power to achieve saturated photoconversion for both proteins (Fig. 4c), it is consistent for different cells. The main aspect that enters in the measurements, potentially damaging the imaging strategy, is the level of transfection for the two plasmids. For all our experiments we transiently transfect the cells and image them after 24 – 48 h. For two separate plasmids their ratio of expression is not linked and often an unbalance in the level of expression for the two proteins can happen. Therefore, the main obstacle to a successful experiment is normally the signal-to-noise level of the double transfection. Strategies can be implemented to optimize the level of expression of both proteins, although the current setting does not preclude the demonstration of the strategy in live cells with the target on highly dynamic organelles.

In Figure 4 g-h, data are presented from zebrafish embryo. It is stated that “single cells could be photolabeled without any sign of photodamage, opening new possibilities for tracking cell dynamics *in vivo*.” How was the viability and lack of photodamage assessed/validated in order to make this claim.

To assess the phototoxicity in the specific case of the Zebrafish embryos we have focused on the persistence of the blood flow, that appear as stripes in brightfield imaging (Fig. R2) or in fluorescence image also after 3 hours (Fig. 6 and Supplementary Fig. 19), and the absence of any lesions to the tissue caused by the photoconverting light. This assessment is now added and presented in Supplementary Fig. 20.

In line with the comments of the other reviewer about phototoxicity we decided to also add a dedicated note (Supplementary Note 5), where we contextualize the energy used for miRFP photoconversion with the one reported in the literature. The energy level needed are indeed inside the range of irradiance typical of other PCFPs and more in general super resolution imaging (Waldchen et al, *Sci Rep*, 2015, doi: 10.1038/srep15348, Alvelid et al. *Current Opinion in Biomedical Engineering*, 2019, doi: 10.1016/j.cobme.2019.09.009). This assessment confirms that the miRFP photoconversion process can be driven with very low risks of phototoxicity in both mammalian cellular and *in vivo* measurements.

Background of Photoswitchable fluorescent proteins

The general concept of photoswitchable fluorescent proteins is introduced in context of two specific references (Ref 1 and 2), restricted to co-authors of the present manuscript. It can be questioned if this is a true representation of the state-of-the art in the field. Photoswitching and photoconversion is of broad general interest e.g. in the fields of photopharmacology and optogenetics, and should preferably be introduced at a more general level.

With the first two references highlighted by the reviewer we wanted to present the possible mechanisms of photoswitching in terms of their chemical characteristics (ref 1: Shcherbakova et al, *Current Opinion in Cellular Biology*, 2014, doi: 10.1016/j.cbpa.2014.04.010) and in the context of what specific applications

requires from such mechanisms to utilize them (ref 2: Shcherbakova et al, *Annual Review of Biophysics*, 2014, doi: 10.1146/annurev-biophys-051013-022836). To expand this collection bringing also other view in defining the topic of photoswitching in fluorescent proteins, we have now added the recent publication from the Nienhaus' group (Nienhaus et al, *Method App Fluoresc*, 2022, doi: 10.1088/2050-6120/ac7d3f).

In the very first few lines we want to set the boundary in which the manuscript belongs, and we believe that opening too much to incorporate the role of photoswitching in photopharmacology and optogenetics would be misleading to the reader. We recognize that photoswitching is a very broad field but the photoconversion we aim to describe here is the one of a fluorescent tag without an effector of a molecular function directly linked to it.

Additional comments

In the abstract, several claims are made at a more general level. Therefore, the abstract is not conveying in what specific way the work is advancing the knowledge in the field. The abstract should be rewritten to better reflect the scientific work, the findings made, and the advances of understanding.

We have changed the Abstract to follow the suggestions of the reviewer, in order to be more specific in what way this work represents new advancements. In short, with this work we are introducing a new photoconversion mechanism entirely driven in the NIR spectral window, the wavelength that induce the photoconversion are >700 nm and the emission and absorption spectral shift of the not-converted and converted species are NIR-to-far-red. The spectra and the kinetics of the photoconversion is advantageous for live cell and *in vivo* imaging application as it allows novel multiplexed experiments.

Figure 2. Are data from HeLa cells or U2OS as stated in materials and methods?

We apology for the mistake and have corrected the cell type for each figure, and also provided this information in each figure legend as well as in Supplementary Table 1.

Figure 4 a-f show data from a (single?) cell, while fig 4g-h demonstrate data from zebra fish larvae. It is confusing to combine these data in the same figure.

We have now separated the panels to two separate figures, Fig 5 and Fig 6, with the aim of more clearly separating the application areas for the reader. Fig 5 now presents the photoconversion in combination with STED microscopy, and Fig 6 presents the *in vivo* experiments from Zebrafish larvae.

The manuscript lacks a concluding paragraph that summarizes the main findings and how the paper actually advances the knowledge in the field.

Together with the reorganization of the text we have restructured also the concluding paragraph.

It would be advised to present an overview of what different types of organisms are investigated in the manuscript and for what purpose: E.coli, HeLa cells (or U2OS?), and zebra fish larvae. Particularly useful would have been to motivate the choice of model organisms in the introduction, in connection to biological relevance and stat of the art in the field.

The choice to express and to photoconvert the miRFP in these biological systems was done to highlight the general applicability of the phenomenon of photoconversion rather than carrying out one specific and indepth biological investigation, which is out of the scope of this work.

We demonstrated that miRFP can be successfully expressed and photoconverted in very different systems, highlighting the robustness of the probe at different maturation condition and imaging condition. We opted for *E. coli* as example of prokaryote as very much used in bacterial research and provided a confine example of activation which can inspire new imaging experiments in bacteria films.

We chose HeLa as a commonly used model system for eukaryotic sub-cellular imaging studies. Here, we provided proof-of-principle experiments to track distinct organelles within the same cell, but a selected spatial and temporal start provided by photoconversion of two probes simultaneously.

We chose Zebrafish as an example of in vivo bio-imaging of a multi-cellular organism.

To help clarify the cell type and organisms considered we have collected this information in the added table of the supplementary material (Supplementary Table 1) and added this information wherever it was missing.

Reviewer #2 (Remarks to the Author)

The authors have satisfactorily addressed my criticism, I have no further comments and support publication of this interesting study.

Reviewer #3 (Remarks to the Author)

Report to 393188 revised version

General: The revised version of Pennacchietti et al (393188) originally submitted in 11/2022 has been improved to a good extent. The length of the rebuttal letter (31 pages), however, points to the flaws and partly imprecision of the original submission. In fact, reading through the reports of my three reviewer colleagues yields the same aspects in each report that make the manuscript disputable whether the paper is of sufficient interest for the wide readership of NCOMM.

The responses of the authors regarding the improvement of the fluorescence quantum yield of these novel compounds do not level off the overall relatively low efficiency (as this is a fact, it remains a challenge to find arguments for the employment of these compounds in optogenetic applications). A second field of critical discussion is the excitation of the Soret band of these compounds (exhibiting a fairly low oscillator strength), where the reviewers' reports indicated the small shift of these bands when the two states, lit and dark, are compared.

What remains is the peculiarity of these compounds that undergo a blue shift instead of the commonly found red shift of absorption in this class of compounds. Yet, to the understanding of this reviewer, an advantage in applications of these compounds does not become clear.

Overall, this reviewer is not convinced that the obtained results and the potential applications are advantageous over those from already introduced proteins, and thus this manuscript may not reach the broad interest of the NCOMMs readership. Yet, I will leave this decision to the editorial board.

Rebuttal to individual points raised by this reviewer: The underlying photochemistry (spectral parameters, quantitative description etc.) has been explained in greater detail yet, see above some of the 'general' comments).

Presentation of the specimen (abbreviation clarity, structural presentation of the compound(s) has been improved and makes the manuscript more readily understandable to the readership.

The term 'bathy' phytochromes is more a definition than a clear fact, so this point can be considered as sufficiently discussed in the rebuttal letter (although these compounds show a photochemistry identified for 'bathy' phys).

Comment on low S/N ratio is sufficiently well explained.

Determining the quantum yield of conversion (originally 1.93/94): the doubt that this method is reliable, seems to be not sufficiently well discussed in the rebuttal letter. There are a number of parameters to keep in mind, amongst other the finding that many of these proteins show heterogeneous absorption bands – easy to identify if the fluorescence excitation wavelength is scanned through the envelope of the absorption band. As has been reported positioning the excitation beam on the high energy shoulder, in the center, or on the low energy shoulder in some proteins yields excitation bands that vary in the wavelength of the maximal emission. Also, as has already been outlined in the report to the original submission, quantum yields of fluorescence for the two states most probably differs, and one may consider a partial spectral overlap of both states to a ‘contamination’. Still, this reviewer is convinced to go straight and measure spectrophotometrically the conversion. The employment of the here described method may serve as a relative measure in between these proteins.

Short comments: (i) Question why the D-ring does not convert: please clarify the manuscript better by highlighting the strong H-bonds.

(iv) The authors are correct here, there is no emphasis in their manuscript to claim an exception. These proteins ‘seem to follow the rule’. However, this sentence: ‘However, in miRFPs the

chromophore is stabilized in the Pr state, and its protonation may cause its hypsochromic shift observed experimentally in this paper’ seems to be illogic. This reviewer understands that the Pr state is already protonated. Then, there should be no further protonation causing a blue shift, and, in fact, the result that protonation in GFPs (of various variation) causes a blue shift does not hold here. The phy-community seems to be more convinced that the conjugation length, i.e., the extent

Reviewer #4 (Remarks to the Author)

The revised version of the manuscript has been substantially improved. The main aim of the paper is now much clearer. The central novelty lies in the concept photoconversion of the miRFPs family of fluorescent proteins in combination with super-resolution microscopy. Specifically, the possibility for blue shifted emission using non-linear excitation is of high interest in order to enable advanced intracellular diffusion studies. Both the underlying mechanisms as well as the demonstrations of potential applications are now supported. Information about methods, statistics and reproducibility is now more substantiated. For example, the table in supplementary gives a valuable overview of the experiments performed to support the claims. The conclusions drawn are discussed and problematized in a rigorous manner. To conclude, the authors have done a thorough work in answering and addressing the previous questions and concerns. I find the revised manuscript suitable for publication.

Revisions for the manuscript NCOMMS-22-41203A:

Blue-shift photoconversion of near-infrared fluorescent proteins for labeling and tracking in living cells and organisms

Francesca Pennacchietti, Jonatan Alvelid, Rodrigo A. Morales, Martina Damenti, Dirk Ollech, Olena S. Oliinyk, Daria M. Shcherbakova, Eduardo J. Villablanca, Vladislav V. Verkhusha, and Ilaria Testa

We thank all the Reviewers for acknowledging the improvement of the revised manuscript.

In this letter we further address the remaining concerns raised by Reviewer #1 and #3 regarding switching induce photo-toxicity checks and further clarification on switching kinetics. Below we provide a point-by-point response. The comments of the Reviewers are written in blue and our responses in black. The changes as compared to the last version of the manuscript are highlighted in yellow.

Reviewer comments

Reviewer #1

The reviewer acknowledges that the authors have now opted to specify in some detail the limitations of the previous approaches. The reviewer notices that the authors listed ‘slow kinetics’ as a limiting factor for reference [18]. The reviewer has carefully studied this work and cannot find any study that refers to a process characterised by slow kinetics. Can the authors specify what they mean, as this limitation is mentioned in the rebuttal and in the main manuscript and can be easily misinterpreted in their chosen word context.

The kinetics we are referring to is the dependency of the photoconversion yield to the light dose (at the wavelength of photoconversion), as described among other in Figure 3b-c. We did not intend “slow” as a limitation but rather as a description of the photoconversion mechanism driven by the primed conversion mechanism itself, therefore we have rephrased those sections of the main text (highlighted in yellow) to avoid any misleading interpretations.

The kinetics of photoconversion directly influence the applicability range of a photoconversion mechanism, defining its temporal boundaries. For example, if we consider a photolabeling experiment where the main aim is tracking, the speed with which a photoconverted species is formed (at a sufficient signal-to-noise ratio) will represent the temporal cutoff to the accessible dynamics of the investigated system. This parameter is therefore more or less influential depending on the biological application: while it is crucial for dynamics happening in the millisecond/second time scale (such as the presented organelle dynamics) it is less relevant for slower dynamics that can extend over several minutes, such as cell migration in tissue.

The non-linear process of photoconversion using a picosecond 775nm laser source – albeit not presented in an in vivo context - is convincing. It is though intriguing that the 405nm photoconversion is not visibly evident below the focal plane at 15µm depth as indicated in panel f). Can the authors comment on it? Further, the authors are encouraged to also include the photoconversion yield for experimental panel (c) to ensure an objective comparison with d) and e).

We thank the reviewer for the positive feedback on the experimental demonstration of the axial confinement for miRFP photoconversion, Following the Reviewer’s comment we updated Supplementary Figure S8 to include the photoconversion yield vs power also for the third condition tested, 775 nm with 700 ps pulse width and 40 MHz repetition frequency (i.e. the STED beam).

For the 405 nm in Supplementary Figure S8c, the axial profile is affected by the scattering of the 405 nm illumination, which contributes to a higher level of photoconversion in the first layer of the bacterial pellet with respect to the 775 nm illumination. Those two aspects (the linear response to photoconversion through 405 nm light and scattering by the first layer) are convoluted in the intensity

profile observed in the plot. Nevertheless, from the line profile a longer tail can be observed, extending beyond the plane of photoconversion to a greater extent with respect to the profile obtained when using either of the two NIR 775 nm illumination sources.

Supplementary Figure S8. Optical sectioning for different photoconversion wavelengths. *E. coli* expressing miRFP713 illuminated with different photoconversion-inducing illumination sources: 775 nm (120 fs, 80 MHz) (a), 405 nm (CW) (b), and 775 nm (700 ps, 40 MHz) (c). (a–c) The photoconverted area is at 15 μm from the cover glass, and in the images the blue-shifted channel after photoconversion is reported for the xy plane at $z = 15 \mu\text{m}$ (up) and the xz projection (down). (d–e) Photoconversion yield for the blue-shifted channel at increasing laser power of the photoconverting-inducing illumination at 775 nm (120 fs, 80 MHz) (d), 775 nm (700 ps, 40 MHz) (e) or 405 nm (CW) (f). The vertical dotted lines specify the energy of the experiment in panels (a–b). It is important to note that these images have been acquired on a scanning system, therefore the effective energy per pixel is convoluted with the scanning. Each reported data point is the mean, and the shaded area is the SD, of the photoconversion yield for the bacteria enclosed in an area of $15 \times 15 \mu\text{m}^2$. (g) Intensity profile along the z-axis for the area enclosed in the dotted lines of panel a-to-c. All experiments have been repeated twice for each of the conditions.

The reviewer acknowledges the referenced work from the Sauer lab that has been chosen by the authors as selected benchmark, which though differs in the experimental context (in particular, in vitro in Waldchen et al, 2016 vs in vivo in this study). The Sauer lab’s work emphasises that their reported results strongly suggest that examination for phototoxic effects should not be limited to phenotypic assessment immediately after imaging as the authors opted to present in Figure R2. To significantly strengthen the presented work and increase the appeal of the presented work for the in vivo imaging community, the reviewer strongly recommends performing a direct assessment of cell division upon photoconversion. The reviewer thanks the authors for sharing the power calculations. While the total dose of light an experimenter puts into the cell/tissue is important, what is far more important for the well-being is the instantaneous power delivered to the cell/tissue. The authors are therefore encouraged

to also include in their comparison the instantaneous peak intensity values for each modality that can faithfully reflect the diverse power values when using pulsed versus continuous wave illumination.

We agree with the reviewer on the importance of providing a full spectrum of information about phototoxicity, therefore we updated the table where we collected the value of irradiance limit gather from literature references as well as our experimental parameters. We now report the average power at the back aperture as well as the peak power for the two pulsed laser sources used to photoconvert.

λ^{PC}	Energy at saturation for miRFP	$\langle P \rangle^*$	P_{peak}^*	Irradiance limit
UV/Vis CW: 405 nm	$\sim 5 \text{ J/cm}^2$	$\sim 2 \text{ mW}$	-	$\sim 50 \text{ J/cm}^2$
NIR 550 ps: 775 nm	200 J/cm^2	$\sim 15 \text{ mW}$	$\sim 680 \text{ mW}$	$< 10 \text{ mW}^{**}$
NIR 120 fs: 670 – 800 nm	4 J/cm^2	$\sim 4 \text{ mW}$	$\sim 500 \text{ W}$	$\sim 1 \text{ kJ/cm}^2$ or $7 - 10 \text{ mW}^{**}$

* Average power, $\langle P \rangle$, and peak power, P_{peak} , at the back aperture of the objective.

** Average power at the sample plane

We also extended the discussion on phototoxicity reporting with more detail the work of Hopt et al, in which the photodamage induced by two-photon excitation is extensively investigated. The experimental conditions reported in the studies of Hopt et al and in the therein referenced studies are similar to our study (same pulse width, repetition rate, and wavelength), therefore it represents a valuable resource in judging the applicability of miRFP photoconversion *in vivo*.

From the supplementary text:

“c. NIR pulsed light with a 120 ps pulse width and 80 MHz repetition rate. This laser source is commonly used for two-photon imaging and the power density required by miRFP photoconversion is comparable to the values reported in two-photon in vivo application. At 700 nm we used around 4 mW at the back aperture with a dwell time of 3.16 μs and the reported values for two-photon imaging is between 1 and 50 mW⁴. More specifically, different systematic studies on photodamage for non-linear excitation identify $\sim 7-10 \text{ mW}$ as the limit average laser power on the specimen before critical cell death^{5,6}. The compared studies rely on different assays to quantify damage: cell membrane integrity, cloning efficiency, viability measurements in different systems, and reactive oxygen species, and they all converge to the same limit. It is important to note that these earlier studies on phototoxicity always account for repetitive scanning (that would correspond to a timelapse acquisition, ~ 10 frames), while the photoconversion in our study is generally performed only for the number of times required to photoconvert the region of interest. Another aspect that often makes a comparison between different studies difficult is the dwell time used in the imaging, from a few μs to tens of μs .”

In addition to the literature references, we experimentally tested the effect on cell viability for the described blue-shift photoconversion mechanism and how it compares to the commonly used red-shift photoconversion mechanism, of which we considered Dendra2 as our standard.

*We monitored the growth of miRFP720-expressing *E. coli* colonies over a period of around 100 min when illuminated or not by a photoconversion wavelength: either 775 nm (550 ps, 40 MHz) at the saturation level required by miRFP720 photoconversion, or 405 nm (CW) at the saturation level required by Dendra2 (Supplementary Figure N10a–c). The ratio of growth for the NIR-photoconverted and non-photoconverted area is close to 1, and similar to the ratio of growth obtained for the UV/Violet photoconversion (Supplementary Figure N10d).*

We also monitored HeLa cells in a cell division assay over 7–11 h. Here, similarly, the probability of cell division for HeLa cells does not show any change upon NIR illumination, neither is any increase of cell death observed (Supplementary Figure N10e–g). The probability of division in and out of the photoconverted area are both within the range expected if we consider a division time of 22 hours for HeLa cells and no systematic difference is observed.

The data has been integrated in Supplementary Note 5 and as additional panels in Figure 3 of the Main Text. The reference in the Main Text to this section has been highlighted in yellow as well as the related Methods section.

Supplementary Figure N10. Cell viability after photoconversion. (a) *E. coli* expressing miRFP720 have been photoactivated with 775 nm light at saturation level in a region of around 25 μm (dotted square) and (b) subsequently followed in brightfield for around 100 min over a bigger region of 50 μm . This allows to follow at once the growth of bacteria in and out of the photoconverted region, quantifying the increase in area of a bacteria colony (c, top). The same experiment has been repeated for photoconversion with UV/Violet light at a power of saturation for Dendra2 (bottom). The box plot reports the ratio of the colony area in the last time point (~ 96 min) with respect to the first time point in four independent repetitions. The box indicates the 25 to 75% and the line shows the mean. (d) To directly compare the effect of the two wavelengths of photoconversion the ratio between the growth in the photoconverted area over that in the non-photoconverted area is reported, where the color code of each dataset matches between the two box plots. (e) Phototoxicity assay through cell division upon 775 nm illumination. Hela cells transiently expressing H2B-miRFP720 have been photoconverted in a confined area (dashed square) and (f) subsequently followed by monitoring a larger area in brightfield over a time period of 7–11 hours. (g) Each cell enclosed in the field of view has been labelled according to their fate over the observation period: death, division or unperturbed behavior (“live”). The probability of each fate is then reported in the stacked columns, together with the number of cells for each area (photoconverted and non-photoconverted) and the time of observation. To note is that the analysis report on all the cells, transfected and not.

Figure 3

The reviewer believes that the normalisation of achievable photoconversion yields to illumination intensity with other photoconversion modalities in b) and c) is problematic when put in direct comparison to miRFP. The reader will get the impression that the achievable conversion yields are comparable to miRFP. The authors are advised to include the photoconversion yield raw values for all cases to ensure a fair comparison (see comments for Figure R1). Can the authors confirm that the 775nm light in 3c is based on pulsed illumination for miRFP and continuous wave illumination for primed conversion of Dendra2? The reviewer has a hard time to understand e) based on the limited figure legend. Can the authors explain how/where the 405nm light was used.

Following the suggestion of the reviewer we have now changed the axis of Figure 3, panel b and c, to report the photoconversion yield of both PCFPs. In order to preserve the main information that we wanted to highlight with this quantification, i.e. the kinetics of photoconversion, we split the information of each PCFP on the two y-axes.

Furthermore, we updated the figure legend to more clearly report on the presented experiments. The 775 nm light used for this experiment is pulsed, and the same laser source is used for both the photoconversion of miRFP713 or miRFP720 and Dendra2 (700 ps, 40 MHz). In the specific case of primed conversion, the 775 nm pulsed illumination is combined with a pulsed 488 nm light of 50 - 70 ps in width and 78 MHz. Following the original publication of Dempsey et al. Nat Meth 2015 this should not preclude the primed conversion ability:

“Using a tunable pulsed laser source to generate converting photons, we found a wide spectral range (~700–850 nm) in the NIR range that—in combination with 488-nm illumination—efficiently photoconverted Dendra2”

Finally, we also corrected a mistake on the figure legend, where 405 nm was indicated instead of 775 nm.

Figure 3. Multiplexed photoconversion strategies for green-to-red and NIR-to-far-red PCFPs. (a) Emission spectra for Dendra2 and miRFP713 before (solid) and after (dashed) photoconversion. (b, c) Kinetics of photoconversion upon various illumination for miRFP713 (dark grey) and Dendra2 (blue). (b) The fold of conversion for the two proteins is compared over the same energy range of 405 nm illumination (CW). (c) The energy dependence of the conversion yield for miRFP with 775 nm (700 ps, 40 MHz) illumination is compared to the primed conversion for Dendra2 with fixed 488 nm (CW, 0.13 kW/cm²) and increasing 775 nm (700 ps, 40 MHz) illumination. The comparisons in (b, c) have been performed on a layer of *E. coli* expressing miRFP713 or Dendra2. Each data point is the mean and SD for bacteria enclosed in a field of view of $\sim 15 \times 15 \mu\text{m}^2$ (approximately 100 bacteria). Representative example of spatially uncoupled (d) and coupled (e) photoconversion. (d) Using photoconversion wavelengths and illumination conditions unique for the PCFPs, like pulsed 775 nm for miRFP and low energy of 405 nm light for the green-to-red PCFPs, two different spatial locations (within the dotted square) can be selectively photoconverted in the field of view. (e) The spatially localized illumination with a combination of high 775 nm and 488 nm light (within the squared dotted area) photoconverts both PCFPs in the same area. Scale bar, 10 μm .

Reviewer #3

Remarks to the Author

Report to 393188 revised version

General: The revised version of Pennacchietti et al (393188) originally submitted in 11/2022 has been improved to a good extent. The length of the rebuttal letter (31 pages), however, points to the flaws and partly imprecision of the original submission. In fact, reading through the reports of my three reviewer colleagues yields the same aspects in each report that make the manuscript disputable whether the paper is of sufficient interest for the wide readership of NCOMM.

Thank you for your opinion. The revision process and comments of the reviewers have contributed to strengthen the work, clarifying and adding important information to the original paper. We would like to note that two of the three other reviewers (#2 and #4) have no further comments and support the publication of the revised manuscript. Reviewer #1 also appreciates our revision and has a few minor comments, which we have now addressed in this second revision.

The responses of the authors regarding the improvement of the fluorescence quantum yield of these novel compounds do not level off the overall relatively low efficiency (as this is a fact, it remains a challenge to find arguments for the employment of these compounds in optogenetic applications).

The degree of blue-shifted photoconversion of the near-infrared fluorescent proteins, termed miRFPs, is sufficient for spatiotemporal tracking of both the proteins tagged with them as well as cells expressing them, upon photoconversion, as experimentally proven in Figure 3, 4, and 5.

We are sorry but we are confused by the second part of this statement. We did not intend to and do not show any optogenetic experiments or applications in our manuscript. The blue-shift photoconversion of miRFPs is used in fluorescence microscopy and fluorescence imaging applications, not opsin or non-opsin optogenetics.

A second field of critical discussion is the excitation of the Soret band of these compounds (exhibiting a fairly low oscillator strength), where the reviewers' reports indicated the small shift of these bands when the two states, lit and dark, are compared.

As reported in the paragraph where we introduce the use of 405 nm illumination (line 154):

"... The Soret absorption band can be used to drive photoconversion in the UV/Vis spectral region ..."

the Soret band is not used as excitation light but rather as photoconversion light. Furthermore, given the one-way nature of the photoconversion (as observed in Supplementary Fig. S9 and S22) it is not expected that excitation with 405 nm or nearby wavelengths would trigger a backward reaction. In other words, the two species are excited and detected in the NIR and far-red spectral ranges respectively, and we do not use UV light to selectively excite one species or the other.

What remains is the peculiarity of these compounds that undergo a blue shift instead of the commonly found red shift of absorption in this class of compounds. Yet, to the understanding of this reviewer, an advantage in applications of these compounds does not become clear.

Yes, the discovered by us photoinduced blue-shift in both excitation and emission spectra is indeed unique and to the best of our knowledge is only observed in these NIR FPs of the miRFP family engineered from bacterial phytochromes.

We here demonstrate the advantage of this photoconversion mechanism in photolabeling studies.

Overall, this reviewer is not convinced that the obtained results and the potential applications are advantageous over those from already introduced proteins, and thus this manuscript may not reach the broad interest of the NCOMMS readership.

Yet, I will leave this decision to the editorial board.

We are sorry to hear this statement. The advantages of NIR FPs such as miRFPs, are their ability to utilize the far-red and NIR spectral ranges for protein tagging, cell labelling, and in vivo imaging, all of which were shown in our manuscript. Moreover, having the miRFP-tagged proteins as well as miRFP-expressing cells or tissues available to be instantly photolabeled by a blue-shift-inducing illumination allows us to move from static to dynamic visualization, namely to perform tracking of these photolabeled biological objects in space and in time, which was not possible – in this far-red/NIR spectral range – before our studies. We demonstrate the spatiotemporal applications of these blue-shift photolabeled proteins and cells in several sections throughout the manuscript, for examples in Fig. 4, 5 and 6.

Rebuttal to individual points raised by this reviewer:

The underlying photochemistry (spectral parameters, quantitative description etc.) has been explained in greater detail yet, see above some of the ,general‘ comments). Presentation of the specimen (abbreviation clarity, structural presentation of the compound(s) has been improved and makes the manuscript more readily understandable to the readership.

We greatly appreciate your opinion on the improvement of our structural and photochemistry presentation and explanation of the miRFP series of NIR FPs engineered from bacterial phytochromes.

The term ,bathy‘ phytochromes is more a definition than a clear fact, so this point can be considered as sufficiently discussed in the rebuttal letter (although these compounds show a photochemistry identified for ,bathy‘ phys).

Comment on low S/N ratio is sufficiently well explained.

Thank you.

Determining the quantum yield of conversion (originally 1.93/94): the doubt that this method is reliable, seems to be not sufficiently well discussed in the rebuttal letter. There are a number of parameters to keep in mind, amongst other the finding that many of these proteins show heterogeneous absorption bands – easy to identify if the fluorescence excitation wavelength is scanned through the envelope of the absorption band. As has been reported positioning the excitation beam on the high energy shoulder, in the center, or on the low energy shoulder in some proteins yields excitation bands that vary in the wavelength of the maximal emission. Also, as has already been outlined in the report to the original submission, quantum yields of fluorescence for the two states most probably differs, and one may consider a partial spectral overlap of both states yito a ,contamination‘. Still, this reviewer is convinced to go straight and measure spectrophotometrically the conversion. Ther employment of the here described method may serve as a relative measure in between these proteins.

We appreciate your detailed explanation of the advantages and disadvantages of the method we used to measure the photoconversion yield and will consider the obtained yield values as the relative measures between the photoconverted species of miRFPs. We would also like to specify that we did not use the term “quantum yield”, but rather yield of photoconversion, and have clearly specified how we derived this ratiometric parameter.

Short comments: (i) Question why the D-ring does not convert: please clarify the manuscript better by highlighting the strong H-bonds.

In this second revision, we have added the explanation of the formed strong H-bond with the biliverdin D-ring and engineered into the biliverdin immediate environment histidine residue in the miRFP proteins.

(iv) The authors are correct here, there is no emphasis in their manuscript to claim an exception. These proteins ,seem to follow the rule‘. However, this sentence: ,However, in miRFPs the

chromophore is stabilized in the Pr state, and its protonation may cause its hypsochromic shift observed experimentally in this paper‘ seems to be illogic. This reviewer understands that the Pr state is already protonated. Then, there should be no further protonation causing a blue shift, and, in fact, the result that protonation in GFPs (of various variation) causes a blue shift does not hold here. The phy-community seems to be more convinced that the conjugation length, i.e., the extent

In this second revision, we have removed our statement regarding the protonation of the biliverdin chromophore in the photoconverted blue-shifted miRFPs species as one of the possible explanations for the cause of their hypsochromic shift. The changes to the revised manuscript are highlighted in yellow in the discussion section.

Revisions for the manuscript NCOMMS-22-41203A:

Blue-shift photoconversion of near-infrared fluorescent proteins for labeling and tracking in living cells and organisms

Francesca Pennacchietti, Jonatan Alvelid, Rodrigo A. Morales, Martina Damenti, Dirk Ollech, Olena S. Oliinyk, Daria M. Shcherbakova, Eduardo J. Villablanca, Vladislav V. Verkhusha, and Ilaria Testa

We thank all the Reviewers for acknowledging the improvement of the revised manuscript.

In this letter we further address the remaining concerns raised by Reviewer #1 and #3 regarding switching induce photo-toxicity checks and further clarification on switching kinetics. Below we provide a point-by-point response. The comments of the Reviewers are written in blue and our responses in black. The changes as compared to the last version of the manuscript are highlighted in yellow.

Reviewer comments Reviewer #1

The reviewer acknowledges that the authors have now opted to specify in some detail the limitations of the previous approaches. The reviewer notices that the authors listed ‘slow kinetics’ as a limiting factor for reference [18]. The reviewer has carefully studied this work and cannot find any study that refers to a process characterised by slow kinetics. Can the authors specify what they mean, as this limitation is mentioned in the rebuttal and in the main manuscript and can be easily misinterpreted in their chosen word context.

The kinetics we are referring to is the dependency of the photoconversion yield to the light dose (at the wavelength of photoconversion), as described among other in Figure 3b-c. We did not intend “slow” as a limitation but rather as a description of the photoconversion mechanism driven by the primed conversion mechanism itself, therefore we have rephrased those sections of the main text (highlighted in yellow) to avoid any misleading interpretations.

The kinetics of photoconversion directly influence the applicability range of a photoconversion mechanism, defining its temporal boundaries. For example, if we consider a photolabeling experiment where the main aim is tracking, the speed with which a photoconverted species is formed (at a sufficient signal-to-noise ratio) will represent the temporal cutoff to the accessible dynamics of the investigated system. This parameter is therefore more or less influential depending on the biological application: while it is crucial for dynamics happening in the millisecond/second time scale (such as the presented organelle dynamics) it is less relevant for slower dynamics that can extend over several minutes, such as cell migration in tissue.

The dependency of the photoconversion yield to the light dose is described as overall efficiency. The authors indeed used this description in their original submission which is also used by the authors who originally reported this mechanism (see under 5.3 in Mohr&Pantazis, Chem. Eur. J. 2018) where they state that primed conversion ‘[...] does not match the efficiency of traditional 405nm photoconversion’. The use of the word ‘kinetics’ is misleading, as it implies that the mechanism of the electron transfer upon primed conversion is known (here: transition times) which has not been reported yet. The authors are advised to revert to their original description.

The non-linear process of photoconversion using a picosecond 775nm laser source – albeit not presented in an in vivo context - is convincing. It is though intriguing that the 405nm photoconversion is not visibly evident below the focal plane at 15µm depth as indicated in panel f). Can the authors comment on it? Further, the authors are encouraged to also include the photoconversion yield for experimental panel (c) to ensure an objective comparison with d) and e).

We thank the reviewer for the positive feedback on the experimental demonstration of the axial confinement for miRFP photoconversion, Following the Reviewer's comment we updated Supplementary Figure S8 to include the photoconversion yield vs power also for the third condition tested, 775 nm with 700 ps pulse width and 40 MHz repetition frequency (i.e. the STED beam).

For the 405 nm in Supplementary Figure S8c, the axial profile is affected by the scattering of the 405 nm illumination, which contributes to a higher level of photoconversion in the first layer of the bacterial pellet with respect to the 775 nm illumination. Those two aspects (the linear response to photoconversion through 405 nm light and scattering by the first layer) are convoluted in the intensity profile observed in the plot. Nevertheless, from the line profile a longer tail can be observed, extending beyond the plane of photoconversion to a greater extent with respect to the profile obtained when using either of the two NIR 775 nm illumination sources.

Supplementary Figure S8. Optical sectioning for different photoconversion wavelengths. *E. coli* expressing miRFP713 illuminated with different photoconversion-inducing illumination sources: 775 nm (120 fs, 80 MHz) (a), 405 nm (CW) (b), and 775 nm (700 ps, 40 MHz) (c). (a–c) The photoconverted area is at 15 μm from the cover glass, and in the images the blue-shifted channel after photoconversion is reported for the xy plane at $z = 15 \mu\text{m}$ (up) and the xz projection (down). (d–e) Photoconversion yield for the blue-shifted channel at increasing laser power of the photoconverting inducing illumination at 775 nm (120 fs, 80 MHz) (d), 775 nm (700 ps, 40 MHz) (e) or 405 nm (CW) (f). The vertical dotted lines specify the energy of the experiment in panels (a–b). It is important to note that these images have been acquired on a scanning system, therefore the effective energy per pixel is convoluted with the scanning. Each reported data point is the mean, and the shaded area is the SD, of the photoconversion yield for the bacteria enclosed in an area of $15 \times 15 \mu\text{m}^2$. (g) Intensity profile along the z-axis for the area enclosed in the dotted lines of panel a-to-c. All experiments have been repeated twice for each of the conditions.

Thank you for the data addition. The reviewer notes that the photoconversion counts are on the very low end of detection for figure b which might result in the impression of a confined photoconversion using 405nm in particular for the area below the focal plane. The authors are encouraged to consider increasing the counts for proper display of the tail.

The reviewer acknowledges the referenced work from the Sauer lab that has been chosen by the authors as selected benchmark, which though differs in the experimental context (in particular, in vitro in Waldchen et al, 2016 vs in vivo in this study). The Sauer lab’s work emphasises that their reported results strongly suggest that examination for phototoxic effects should not be limited to phenotypic assessment immediately after imaging as the authors opted to present in Figure R2. To significantly strengthen the presented work and increase the appeal of the presented work for the in vivo imaging community, the reviewer strongly recommends performing a direct assessment of cell division upon photoconversion. The reviewer thanks the authors for sharing the power calculations. While the total dose of light an experimenter puts into the cell/tissue is important, what is far more important for the well-being is the instantaneous power delivered to the cell/tissue. The authors are therefore encouraged to also include in their comparison the instantaneous peak intensity values for each modality that can faithfully reflect the diverse power values when using pulsed versus continuous wave illumination.

We agree with the reviewer on the importance of providing a full spectrum of information about phototoxicity, therefore we updated the table where we collected the value of irradiance limit gather from literature references as well as our experimental parameters. We now report the average power at the back aperture as well as the peak power for the two pulsed laser sources used to photoconvert.

λ^{PC}	Energy at saturation for miRFP	$\langle P \rangle^*$	P_{peak}^*	Irradiance limit
UV/Vis CW: 405 nm	~ 5 J/cm ²	~ 2 mW	-	~50 J/cm ²
NIR 550 ps: 775 nm	200 J/cm ²	~15 mW	~ 680 mW	< 10 mW **
NIR 120 fs: 670 – 800 nm	4 J/cm ²	~ 4 mW	~500 W	~1 kJ/cm ² or 7 – 10 mW**

* Average power. $\langle P \rangle$. and peak power, P_{peak} , at the back aperture of the objective.

** Average power at the sample plane

We also extended the discussion on phototoxicity reporting with more detail the work of Hopt et al, in which the photodamage induced by two-photon excitation is extensively investigated. The experimental conditions reported in the studies of Hopt et al and in the therein referenced studies are similar to our study (same pulse width, repetition rate, and wavelength), therefore it represents a valuable resource in judging the applicability of miRFP photoconversion *in vivo*.

From the supplementary text:

“c. NIR pulsed light with a 120 ps pulse width and 80 MHz repetition rate. This laser source is commonly used for two-photon imaging and the power density required by miRFP photoconversion is comparable to the values reported in two-photon in vivo application. At 700 nm we used around 4 mW at the back aperture with a dwell time of 3.16 μ s and the reported values for two-photon imaging is between 1 and 50 mW⁴. More specifically, different systematic studies on photodamage for non-linear excitation identify ~ 7–10 mW as the limit average laser power on the specimen before critical cell death^{5,6}. The compared studies rely on different assays to quantify damage: cell membrane integrity, cloning efficiency, viability measurements in different systems, and reactive oxygen species, and they all

converge to the same limit. It is important to note that these earlier studies on phototoxicity always account for repetitive scanning (that would correspond to a timelapse acquisition, ~ 10 frames), while the photoconversion in our study is generally performed only for the number of times required to photoconvert the region of interest. Another aspect that often makes a comparison between different studies difficult is the dwell time used in the imaging, from a few μs^5 to tens of μs^6 .

In addition to the literature references, we experimentally tested the effect on cell viability for the described blue-shift photoconversion mechanism and how it compares to the commonly used red-shift photoconversion mechanism, of which we considered Dendra2 as our standard

We monitored the growth of miRFP720-expressing *E. coli* colonies over a period of around 100 min when illuminated or not by a photoconversion wavelength: either 775 nm (550 ps, 40 MHz) at the saturation level required by miRFP720 photoconversion, or 405 nm (CW) at the saturation level required by Dendra2 (Supplementary Figure N10a–c). The ratio of growth for the NIR-photoconverted and non-photoconverted area is close to 1, and similar to the ratio of growth obtained for the UV/Violet-photoconversion (Supplementary Figure N10d)

We also monitored HeLa cells in a cell division assay over 7–11 h. Here, similarly, the probability of cell division for HeLa cells does not show any change upon NIR illumination, neither is any increase of cell death observed (Supplementary Figure N10e–g). The probability of division in and out of the photoconverted area are both within the range expected if we consider a division time of 22 hours for HeLa cells and no systematic difference is observed

The data has been integrated in Supplementary Note 5 and as additional panels in Figure 3 of the Main Text. The reference in the Main Text to this section has been highlighted in yellow as well as the related Methods section.

Supplementary Figure N10. Cell viability after photoconversion. (a) *E. coli* expressing miRFP720 have been photoactivated with 775 nm light at saturation level in a region of around 25 μm (dotted square) and (b) subsequently followed in brightfield for around 100 min over a bigger region of 50 μm . This allows to follow at once the growth of bacteria in and out of the photoconverted region, quantifying the increase in area of a bacteria colony (c, top). The same experiment has been repeated for photoconversion with UV/Violet light at a power of saturation for Dendra2 (bottom). The box plot reports the ratio of the colony area in the last time point (~ 96 min) with respect to the first time point in four independent repetitions. The box indicates the 25 to 75% and the line shows the mean. (d) To directly compare the effect of the two wavelengths of photoconversion the ratio between the growth in the photoconverted area over that in the non-photoconverted area is reported, where the color code of each dataset matches between the two box plots. (e) Phototoxicity assay through cell division upon 775 nm illumination. Hela cells transiently expressing H2B-miRFP720 have been photoconverted in a confined area (dashed square) and (f) subsequently followed by monitoring a larger area in brightfield over a time period of 7–11 hours. (g) Each cell enclosed in the field of view has been labelled according to their fate over the observation period: death, division or unperturbed behavior (“live”). The probability of each fate is then reported in the stacked columns, together with the number of cells for each area (photoconverted and non-photoconverted) and the time of observation. To note is that the analysis report on all the cells, transfected and not.

The reviewer thanks the authors for including the peak illumination intensities. The authors might consider denoting the intensities as W/cm² or W/m². Further, the peak illumination intensity has not been completed for the peak illumination intensity. The authors might include a note that peak and average is the same under this condition.

The reviewer is concerned that a straightforward control experiment that is critical for demonstrating the validity of the in vivo data has not been performed. Given the new nature of this approach, the reviewer feels strongly to demonstrate photoconverted nuclei can undergo cell division in vivo. This experiment is easy to perform and will assure the developmental biology community of the value of using the suggested approach for in vivo experiments.

Figure 3

The reviewer believes that the normalisation of achievable photoconversion yields to illumination intensity with other photoconversion modalities in b) and c) is problematic when put in direct comparison to miRFP. The reader will get the impression that the achievable conversion yields are comparable to miRFP. The authors are advised to include the photoconversion yield raw values for all cases to ensure a fair comparison (see comments for Figure R1). Can the authors confirm that the 775nm light in 3c is based on pulsed illumination for miRFP and continuous wave illumination for primed conversion of Dendra2? The reviewer has a hard time to understand e) based on the limited figure legend. Can the authors explain how/where the 405nm light was used.

Following the suggestion of the reviewer we have now changed the axis of Figure 3, panel b and c, to report the photoconversion yield of both PCFPs. In order to preserve the main information that we wanted to highlight with this quantification, i.e. the kinetics of photoconversion, we split the information of each PCFP on the two y-axes.

Furthermore, we updated the figure legend to more clearly report on the presented experiments. The 775 nm light used for this experiment is pulsed, and the same laser source is used for both the photoconversion of miRFP713 or miRFP720 and Dendra2 (700 ps, 40 MHz). In the specific case of primed conversion, the 775 nm pulsed illumination is combined with a pulsed 488 nm light of 50 - 70

ps in width and 78 MHz. Following the original publication of Dempsey et al. Nat Meth 2015 this should not preclude the primed conversion ability:

“Using a tunable pulsed laser source to generate converting photons, we found a wide spectral range (~700–850 nm) in the NIR range that—in combination with 488-nm illumination—efficiently photoconverted Dendra2”

Finally, we also corrected a mistake on the figure legend, where 405 nm was indicated instead of 775 nm.

Figure 3. Multiplexed photoconversion strategies for green-to-red and NIR-to-far-red PCFPs.

(a) Emission spectra for Dendra2 and miRFP713 before (solid) and after (dashed) photoconversion. (b, c) Kinetics of photoconversion upon various illumination for miRFP713 (dark grey) and Dendra2 (blue). (b) The fold of conversion for the two proteins is compared over the same energy range of 405 nm illumination (CW). (c) The energy dependence of the conversion yield for miRFP with 775 nm (700 ps, 40 MHz) illumination is compared to the primed conversion for Dendra2 with fixed 488 nm (CW, 0.13 kW/cm²) and increasing 775 nm (700 ps, 40 MHz) illumination. The comparisons in (b, c) have been performed on a layer of *E. coli* expressing miRFP713 or Dendra2. Each data point is the mean and SD for bacteria enclosed in a field of view of ~ 15 × 15 μm² (approximately 100 bacteria). Representative example of spatially uncoupled (d) and coupled (e) photoconversion. (d) Using photoconversion wavelengths and illumination conditions unique for the PCFPs, like pulsed 775 nm for miRFP and low energy of 405 nm light for the green-to-red PCFPs, two different spatial locations (within the dotted square) can be selectively photoconverted in the field of view. (e) The spatially localized illumination with a combination of high 775 nm and 488 nm light (within the squared dotted area) photoconverts both PCFPs in the same area. Scale bar, 10 μm.

The reviewer thanks the authors for including more information for Figure 3. The authors have opted to perform primed conversion with a pulsed converting beam laser which indeed has been reported to be feasible, yet the overall photoconversion yield is inferior to using a converting CW laser (up to 10 times lower, see Dempsey et al., Nat Meth 2015). The authors are requested to use CW laser for optimal settings or include a comment to account for this selection in the legend.

Reviewer #3

Remarks to the Author

Report to 393188 revised version

General: The revised version of Pennacchiotti et al (393188) originally submitted in 11/2022 has been improved to a good extent. The length of the rebuttal letter (31 pages), however, points to the flaws and partly imprecision of the original submission. In fact, reading through the reports of my three reviewer colleagues yields the same aspects in each report that make the manuscript disputable whether the paper is of sufficient interest for the wide readership of NCOMM.

Thank you for your opinion. The revision process and comments of the reviewers have contributed to strengthen the work, clarifying and adding important information to the original paper. We would like to note that two of the three other reviewers (#2 and #4) have no further comments and support the publication of the revised manuscript. Reviewer #1 also appreciates our revision and has a few minor comments, which we have now addressed in this second revision.

The responses of the authors regarding the improvement of the fluorescence quantum yield of these novel compounds do not level off the overall relatively low efficiency (as this is a fact, it remains a challenge to find arguments for the employment of these compounds in optogenetic applications).

The degree of blue-shifted photoconversion of the near-infrared fluorescent proteins, termed miRFPs, is sufficient for spatiotemporal tracking of both the proteins tagged with them as well as cells expressing them, upon photoconversion, as experimentally proven in Figure 3, 4, and 5.

We are sorry but we are confused by the second part of this statement. We did not intend to and do not show any optogenetic experiments or applications in our manuscript. The blue-shift photoconversion of miRFPs is used in fluorescence microscopy and fluorescence imaging applications, not opsin or nonopsin optogenetics.

A second field of critical discussion is the excitation of the Soret band of these compounds (exhibiting a fairly low oscillator strength), where the reviewers' reports indicated the small shift of these bands when the two states, lit and dark, are compared.

As reported in the paragraph where we introduce the use of 405 nm illumination (line 154):

"... The Soret absorption band can be used to drive photoconversion in the UV/Vis spectral region ..." the Soret band is not used as excitation light but rather as photoconversion light. Furthermore, given the one-way nature of the photoconversion (as observed in Supplementary Fig. S9 and S22) it is not expected that excitation with 405 nm or nearby wavelengths would trigger a backward reaction. In other words, the two species are excited and detected in the NIR and far-red spectral ranges respectively, and we do not use UV light to selectively excite one species or the other.

What remains is the peculiarity of these compounds that undergo a blue shift instead of the commonly found red shift of absorption in this class of compounds. Yet, to the understanding of this reviewer, an advantage in applications of these compounds does not become clear.

Yes, the discovered by us photoinduced blue-shift in both excitation and emission spectra is indeed unique and to the best of our knowledge is only observed in these NIR FPs of the miRFP family engineered from bacterial phytochromes.

We here demonstrate the advantage of this photoconversion mechanism in photolabeling studies.

Overall, this reviewer is not convinced that the obtained results and the potential applications are advantageous over those from already introduced proteins, and thus this manuscript may not reach the broad interest of the NCOMMs readership.

Yet, I will leave this decision to the editorial board.

We are sorry to hear this statement. The advantages of NIR FPs such as miRFPs, are their ability to utilize the far-red and NIR spectral ranges for protein tagging, cell labelling, and in vivo imaging, all of which were shown in our manuscript. Moreover, having the miRFP-tagged proteins as well as miRFP-expressing cells or tissues available to be instantly photolabeled by a blue-shift-inducing illumination allows us to move from static to dynamic visualization, namely to perform tracking of these photolabeled biological objects in space and in time, which was not possible – in this far-red/NIR spectral range – before our studies. We demonstrate the spatiotemporal applications of these blue-shift photolabeled proteins and cells in several sections throughout the manuscript, for examples in Fig. 4, 5 and 6.

Rebuttal to individual points raised by this reviewer:

The underlying photochemistry (spectral parameters, quantitative description etc.) has been explained in greater detail yet, see above some of the ‘general’ comments). Presentation of the specimen (abbreviation clarity, structural presentation of the compound(s) has been improved and makes the manuscript more readily understandable to the readership.

We greatly appreciate your opinion on the improvement of our structural and photochemistry presentation and explanation of the miRFP series of NIR FPs engineered from bacterial phytochromes.

The term ‘bathy’ phytochromes is more a definition than a clear fact, so this point can be considered as sufficiently discussed in the rebuttal letter (although these compounds show a photochemistry identified for ‘bathy’ phys).

Comment on low S/N ratio is sufficiently well explained.

Thank you.

Determining the quantum yield of conversion (originally 1.93/94): the doubt that this method is reliable, seems to be not sufficiently well discussed in the rebuttal letter. There are a number of parameters to keep in mind, amongst other the finding that many of these proteins show heterogeneous absorption bands – easy to identify if the fluorescence excitation wavelength is scanned through the envelope of the absorption band. As has been reported positioning the excitation beam on the high energy shoulder, in the center, or on the low energy shoulder in some proteins yields excitation bands that vary in the wavelength of the maximal emission. Also, as has already been outlined in the report to the original submission, quantum yields of fluorescence for the two states most probably differs, and one may consider a partial spectral overlap of both states to a ‘contamination’. Still, this reviewer is convinced to go straight and measure spectrophotometrically the conversion. The employment of the here described method may serve as a relative measure in between these proteins.

We appreciate your detailed explanation of the advantages and disadvantages of the method we used to measure the photoconversion yield and will consider the obtained yield values as the relative measures

between the photoconverted species of miRFPs. We would also like to specify that we did not use the term “quantum yield”, but rather yield of photoconversion, and have clearly specified how we derived this ratiometric parameter.

Short comments: (i) Question why the D-ring does not convert: please clarify the manuscript better by highlighting the strong H-bonds.

In this second revision, we have added the explanation of the formed strong H-bond with the biliverdin D-ring and engineered into the biliverdin immediate environment histidine residue in the miRFP proteins.

(iv) The authors are correct here, there is no emphasis in their manuscript to claim an exception. These proteins ,seem to follow the rule‘. However, this sentence: ,However, in miRFPs the chromophore is stabilized in the Pr state, and its protonation may cause its hypsochromic shift observed experimentally in this paper‘ seems to be illogic. This reviewer understands that the Pr state is already protonated. Then, there should be no further protonation causing a blue shift, and, in fact, the result that protonation in GFPs (of various variation) causes a blue shift does not hold here. The phy-community seems to be more convinced that the conjugation length, i.e., the extent

In this second revision, we have removed our statement regarding the protonation of the biliverdin chromophore in the photoconverted blue-shifted miRFPs species as one of the possible explanations for the cause of their hypsochromic shift. The changes to the revised manuscript are highlighted in yellow in the discussion section.

Revisions for the manuscript NCOMMS-22-41203B:**Blue-shift photoconversion of near-infrared fluorescent proteins for labeling and tracking in living cells and organisms**

Francesca Pennacchiotti, Jonatan Alvelid, Rodrigo A. Morales, Martina Damenti, Dirk Ollech, Olena S. Oliinyk, Daria M. Shcherbakova, Eduardo J. Villablanca, Vladislav V. Verkhusha, and Ilaria Testa

We thank the Reviewers for this and the previous feedback that significantly contributed to improve the work and the manuscript.

In this letter we address the last comments of Reviewer #1, providing a point-by-point response to them. The comments of the Reviewers are written in blue and our responses in black.

Reviewer comments**Reviewer #1**

The dependency of the photoconversion yield to the light dose is described as overall efficiency. The authors indeed used this description in their original submission which is also used by the authors who originally reported this mechanism (see under 5.3 in Mohr&Pantazis, Chem. Eur. J. 2018) where they state that primed conversion '[...] does not match the efficiency of traditional 405nm photoconversion'. The use of the word 'kinetics' is misleading, as it implies that the mechanism of the electron transfer upon primed conversion is known (here: transition times) which has not been reported yet. The authors are advised to revert to their original description.

Following the suggestion of the Reviewer we revert to "efficiency" to describe the photoconversion dependency to light intensity.

Thank you for the data addition. The reviewer notes that the photoconversion counts are on the very low end of detection for figure b which might result in the impression of a confined photoconversion using 405nm in particular for the area below the focal plane. The authors are encouraged to consider increasing the counts for proper display of the tail.

The contrast in Figure 1g and Supplementary Note 5 has been updated. Thanks to the comment we realized that the color bar was wrong with respect to the displayed image and it is now corrected accordingly.

The reviewer thanks the authors for including the peak illumination intensities. The authors might consider denoting the intensities as W/cm² or W/m². Further, the peak illumination intensity has not been completed for the peak illumination intensity. The authors might include a note that peak and average is the same under this condition.

We revised the supplementary table in Supplementary Note 5 on phototoxicity by introducing a footnote for the missing 405 nm peak and average power value.

The reviewer is concerned that a straightforward control experiment that is critical for demonstrating the validity of the in vivo data has not been performed. Given the new nature of this approach, the reviewer feels strongly to demonstrate photoconverted nuclei can undergo cell division in vivo. This experiment is easy to perform and will assure the developmental biology community of the value of using the suggested approach for in vivo experiments.

The concern about phototoxicity raised by the Reviewer has our understanding and we did not mean to underestimate it. On the contrary, we wanted to provide a robust comparison among the different illumination conditions explored for miRFP photoconversion rather than a phenomenological one. This was possible only by simplifying the system: in the zebrafish larvae and with the mosaic expression of the H2B-miRFP plasmid under a ubiquitous promoter, there is no control on which cell type will express the protein and in which amount. This implies variability in the measurements, among which: (1) the location of the miRFP-positive cells in the body of the zebrafish will change, directly influencing the effective light dose; (2) the expression level will vary for different cell types, possibly affecting their division cycle; and (3) the non-uniform reactivity of the different tissue to light, for example pigments are extremely sensitive to light.

In accordance with our experimental settings and available conditions, we assessed phototoxicity directly in 3-5 dpf zebrafish larvae, which is more suited for imaging, but its overall cell proliferation is lower than at earlier stage of development. To compensate for the decreased proliferation, we decided to (1) amputate the tail fin of zebrafish larvae at 72 hpf (3dpf, see Morales & Allende, 2019; PMID: 30891030), (2) photoconvert nuclei in the regenerating tissue 24h after amputation (4dpf), and (3) analyze the growing tissue after an additional period of 20 hours. In the following figure we report these two time points for amputated larvae (gray line in figure 1). It is important to note that between the photoconversion and imaging 20h after, the larvae were kept mounted in 0.5 % agarose at 28 °C, which partially affected the normal growth and shape of the regenerating tail fin. Despite this, we observed a general growth of the amputated tail fin, together with an expansion of the photoconverted signal in the growing tail fin, which supports the non-toxic and non-disruptive effect of the photoconversion light dose to the zebrafish development.

Overall, the presented experiments on zebrafish are to be considered a proof of principle of not only expressing the protein in a multicellular organism (also extensively proven in Matlashov et al. 2020 and in the recent Zhang et al. 2023, doi:10.1038/s41592-023-01975-z), but also successfully photoconverting it in a multicellular organism. We are aware that with this work we are not fully

Figure 1. Phototoxicity in vivo. Zebrafish larvae at 4 to 5 dpf expressing miRFP720-H2B. The photoconverted channel is in magenta, while the non-photoconverted is in green. The same tail region was imaged immediately after photoconversion (a) and after 20 hours (b). The transmitted image is also presented for the second time point. Around 40 nuclei have been photoconverted in the indicated region by illumination with 775 nm light (40 MHz, 550 ps). For each nucleus, a region of $15 \times 15 \mu\text{m}^2$ have been photoconverted at $21 \text{ MW}/\text{cm}^2$ in three consecutive planes at $1 \mu\text{m}$ axial distance. The presented images are projections of $\sim 50 \mu\text{m}$ in depth, of images collected at axial distances of $5 \mu\text{m}$.

covering all the possible controls on phototoxicity, and we have modified the text to clarify the aim of our experiments. In particular, we propose to rephrase the *in vivo* section in the Results section to better present the experiment as a proof of concept:

“miRFP NIR-to-far-red photoconversion: in vivo

We used zebrafish as a proof-of-principle to test the applicability of our method in a multicellular organism. We injected in 1-cell stage zebrafish zygotes a Tol2-based plasmid carrying the nuclear histone H2B-miRFP720 fusion protein under the expression of a ubiquitous promoter (ubb promoter), and we successfully observed labelled nuclei in Zebrafish larvae (Fig. 6). We tested the photoconversion in living day 4 post-fertilization zebrafish larvae. We focused on the aorta-gonad-mesonephros area and selectively photoconverted a single nucleus of the tissue (Fig. 6a). The axial confinement could be confirmed on a volumetric recording (Fig. 6b), and the signal could be followed over an extended period of 2.5 h under live-cell imaging conditions (Fig. 6a and Supplementary Fig. 19). Single nuclei were photoconverted without damaging the tissue and adjacent blood vessels (Supplementary Fig 20).”

Furthermore, in Supplementary Note 3 great attention has been devoted to collecting literature references on phototoxicity, where the response of different systems to similar and higher doses is reported. Based on them, we would like to stress again that the light doses of miRFP photoconversion used are in the same order of magnitude as two-photon and STED microscopy, both of which are techniques currently used for *in vivo* imaging.

The reviewer thanks the authors for including more information for Figure 3. The authors have opted to perform primed conversion with a pulsed converting beam laser which indeed has been reported to be feasible, yet the overall photoconversion yield is inferior to using a converting CW laser (up to 10 times lower, see Dempsey et al., Nat Meth 2015). The authors are requested to use CW laser for optimal settings or include a comment to account for this selection in the legend.

To comply with the clarification asked by the Reviewer we specify the type of illumination used for primed conversion and how it compares with its reported application in the Material and Methods section.

“The pulsed 775 nm light in combination with 488 nm light was used for primed conversion of the green-to-red variants. Although less efficient than illumination with continuous wave light, this configuration allows the exploration and integration of the mechanism in the commercial microscope used.”

The choice of pulsed illumination rather than continuous wave has been dictated by the instrumentation at our availability, and no CW laser of the wavelength and characteristics requested is at our disposal. In particular, the experiments in Figure 3 panel a-e have been performed on a Leica SP8 as reported in the Material and Methods. All the laser lines of interest for the performed experiment are pulsed and with characteristics that, as acknowledged by the Reviewer, are stated in the figure legend.

The exploration and implementation of primed conversion is contingent on the comprehensive characterization and implementation of the miRFP photoconversion. For this process, 405 nm and far-red pulsed illumination have been found to be the most efficient wavelengths, therefore those are the wavelengths at which we validated crosstalk effects as well as alternative schemes for multiplexing. We based our implementation of primed conversion on the original paper of Dempsey et al. where both illumination conditions (pulsed and CW) are explored. The paper and other seminal works on the detail and use of primed conversion are clearly cited in the manuscript.